# REVISITING MULTIVARIATE TIME SERIES FORECASTING WITH MISSING VALUES

## ABSTRACT

Missing values are common in real-world time series, and multivariate time series forecasting with missing values (MTSF-M) has become a crucial area of research for ensuring reliable predictions. To address the challenge of missing data, current approaches have developed an imputation-then-prediction framework that uses imputation modules to fill in missing values, followed by forecasting on the imputed data. However, this framework overlooks a critical issue: there is no ground truth for the missing values, making the imputation process susceptible to errors that can degrade prediction accuracy. In this paper, we conduct a systematic empirical study and reveal that imputation without direct supervision can corrupt the underlying data distribution and actively degrade prediction accuracy. To address this, we propose a paradigm shift that moves away from imputation and directly predicts from the partially observed time series. We introduce **C**onsistency-**R**egularized **I**nformation **B**ottleneck (CRIB), a novel framework built on the Information Bottleneck principle. CRIB combines a unified-variate attention mechanism with a consistency regularization scheme to learn robust representations that filter out noise introduced by missing values while preserving essential predictive signals. Comprehensive experiments on four real-world datasets demonstrate the effectiveness of CRIB, which predicts accurately even under high missing rates. Our code is available in https://anonymous.4open.science/r/CRIB-F660.

## 1 INTRODUCTION

Multivariate time series forecasting (MTSF), which aims to predict future values of multiple variates based on historical observations, plays an important role in many domains, such as traffic flow forecasting (Shang et al., 2022; Yu et al., 2017; Bai et al., 2020), financial analysis (Schaffer et al., 2021; Zivot & Wang, 2006; Hu et al., 2025b;a), and weather prediction (Zheng et al., 2015; Wu et al., 2021; Tan et al., 2022). However, due to uncontrollable factors such as data collection difficulties and transmission failures (Li et al., 2023; Marisca et al., 2022; Cini et al., 2021; Zhang et al., 2025a), real-world multivariate time series data is often partially observed, with missing values scattered throughout the series. These missing values inevitably introduce noise, leading to distribution shifts and disrupting the variate correlations. MTSF models (Cao et al., 2020; Liu et al., 2022; Ekambaram et al., 2023; Hu et al., 2025e), which typically rely on complete data, are highly sensitive to such shifts and correlation destruction, thus failing to make accurate predictions (Zhou et al., 2023; Hu et al., 2025c). This has driven increasing interest in multivariate time series forecasting with missing values (MTSF-M) (Cao et al., 2018; Zuo et al., 2023; Tang et al., 2020), where the objective is to generate accurate and robust forecasts despite the presence of incomplete data.

To mitigate the impact of missing values, recent MTSF-M research (Yu et al., 2025; Peng et al., 2025) has focused on enhancing observed data by imputing missing values to improve prediction performance. One common approach is the two-stage framework, where an imputation module (Wu et al., 2022; Cao et al., 2018; Du et al., 2023) first fills in the missing values, and a forecasting model then predicts future values based on the imputed data (Peng et al., 2025; Chen et al., 2023; Wu et al., 2015). Moreover, to reduce error accumulation between these two stages of two separate models, some studies have proposed an end-to-end framework (Yu et al., 2024; 2025) that imputes missing values progressively during encoding and performs forecasting using the imputed representations. Overall, these methods generally follow an imputation-then-prediction paradigm, aiming

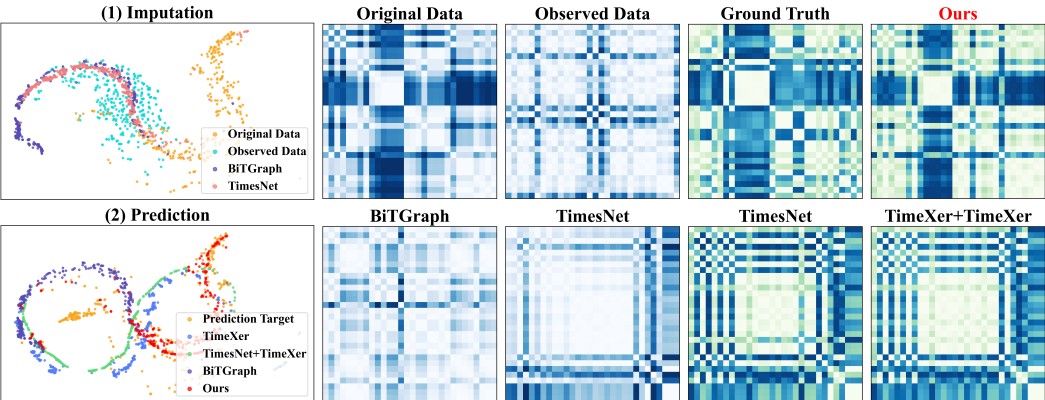

Figure 1: Analysis of the imputation-then-prediction paradigm on PEMS-BAY (40% missing rate). **(a)** t-SNE visualizations show that current imputation modules cannot recover the original data distribution and their forecasts mismatch with the prediction target, while our direct-prediction method aligns better with the target. **(b, c)** Correlation maps reveal that imputation fails to recover true variate correlations, whereas our method preserves underlying correlations more effectively.

to improve forecasting accuracy by mitigating the negative effects of missing values compared to directly applying forecasting models to incomplete data.

However, current MTSF-M methods ignore a critical limitation in real-world applications: **there is no ground truth for missing values**. In such scenarios, the imputation module of the current MTSF-M methods would lack reliable guidance, which means the imputed values and reconstructed correlations cannot be guaranteed to be accurate with only the final prediction guidance. As a result, noise would propagate into the prediction stage and degrade forecasting performance, particularly when the missing rate is high. To investigate this issue, we conduct an empirical analysis of representative imputation-then-prediction methods, where original and observed data denote the complete and partially observed data, respectively. This includes the two-stage framework combining Times-Net (Wu et al., 2022) for imputation and TimeXer (Wang et al., 2024b) for forecasting, as well as the end-to-end framework BiTGraph (Chen et al., 2023). Fig. 1 illustrates the empirical results, where panel (a) visualizes the distributions of imputed and predicted values, and panels (b) and (c) present the correlations among variates. Our findings highlight two key phenomena:

❶ *Improper imputation can corrupt the observed data.* Current MTSF-M frameworks commonly employ imputation modules to recover missing values. However, as shown in Fig. 1 (a-1, b), without enough direct supervision, imputed values deviate considerably from the distribution of the original complete data, and the underlying correlations among variates are not correctly reconstructed. The deterioration of both the data distribution and variate correlations suggests that imputation with only prediction guidance can degrade the observed data rather than repair it.

❷ *Flawed imputation, in turn, leads to poor prediction performance.* Errors from the imputation stage inevitably propagate into forecasting. As shown in Fig. 1 (a-2, c), the predictions exhibit large deviations from the prediction targets. Notably, even a model TimeXer applied directly to incomplete observed data outperforms a more complex framework that combines TimesNet for imputation with TimeXer for prediction. These findings indicate that a flawed imputation stage can actively harm, rather than enhance, the forecasting capabilities of a model.

Based on these two observations, we ask a fundamental question: *Is it possible to predict directly from partially observed time series, avoiding the pitfalls of imputation while maintaining high accuracy?* To answer this, we propose **C**onsistency-**R**egularized **I**nformation **B**ottleneck (CRIB), a novel framework that predicts directly from partially observed data, bypassing the issues associated with imputation. CRIB is built on the Information Bottleneck (IB) principle, which enables it to learn a compressed representation that filters noise from missing values while preserving essential predictive signals. To achieve this, it employs a unified-variate attention mechanism to capture complex correlations from the sparse input and is trained with a consistency regularization scheme to enhance robustness, especially under high missing rates.

Our main contributions can be summarized as follows:

- **Empirical analysis**: We perform a systematic empirical analysis of the dominant imputation-then-prediction paradigm for MTSF-M. We reveal that, guided only by a prediction objective, imputation modules can corrupt the observed data distribution and degrade prediction performance.

- **Method**: We propose a novel direct-prediction method, CRIB, which removes the imputation completely. CRIB is an IB-based method that integrates a unified-variate attention mechanism and consistency regularization to get refined representations, effectively balancing the tradeoff between filtering out noise and preserving task-relevant signals.

- **Experiments**: We conduct comprehensive experiments on four real-world benchmarks and show that CRIB significantly outperforms existing state-of-the-art methods by an average of 18%, especially under high missing rates. Our results validate the superiority of the proposed direct-prediction approach over the imputation-then-prediction paradigm.

## 2 PRELIMINARIES

**Notations & Problem Formulation**  In MTSF-M tasks, the historical time series is denoted as $X = \{x_i^{1:T} \mid i = 1, \cdots, N\} \in \mathbb{R}^{N \times T}$, where $T$ is the number of time steps and $N$ is the number of variates. The goal is to predict the future $S$ time steps $Y = \{x_i^{T+1:T+S} \mid i = 1, \cdots, N\} \in \mathbb{R}^{N \times S}$. Missingness is represented by a binary mask $M \in \{0, 1\}^{N \times T}$, where $X^{\mathrm{o}} = \{X^{i,j} | M^{i,j} = 1\}$ are observed values and $X^{\mathrm{m}} = \{X^{i,j} | M^{i,j} = 0\}$ are missing values. We denote $Z \in \mathbb{R}^{N \times D}$ as the intermediate representations of input, where $D$ is the dimension of the representation.

**Information Bottleneck for MTSF-M**  IB theory (Tishby & Zaslavsky, 2015; Voloshynovskiy et al., 2019) provides an information-theoretic framework for learning compact and informative representations. Given the partially observed input $X^{\mathrm{o}}$ and the prediction target $Y$, the goal is to learn a latent representation $Z$ that is maximally compressive with respect to $X^{\mathrm{o}}$ while remaining maximally informative about $Y$. This trade-off in CRIB is formalized by the following objective:

$$\min_{\theta} [I_{\theta}(Z; X^{\mathrm{o}}) - \beta \cdot I_{\theta}(Y; Z)]. \tag{1}$$

Here, $\theta$ represents the learnable parameters of our proposed CRIB. $I(Z; X^{\mathrm{o}})$ and $I(Y; Z)$ are the mutual information terms measuring compactness and informativeness, respectively. The Lagrange multiplier $\beta \in \mathbb{R}^{+}$ controls the balance between these two terms (Tishby et al., 2000). Furthermore, under standard assumptions in the IB literature (Alemi et al., 2016; Chalk et al., 2016; Ma et al., 2023), the joint distribution of the variables can be factorized as:

$$p(X^{\mathrm{o}}, Y, Z) = p(Z|X^{\mathrm{o}}, Y)p(Y|X^{\mathrm{o}})p(X^{\mathrm{o}}) = p(Z|X^{\mathrm{o}})p(Y|X^{\mathrm{o}})p(X^{\mathrm{o}}), \tag{2}$$

namely, there is a Markov chain $Y \leftrightarrow X^{\mathrm{o}} \leftrightarrow Z$, indicating that the representations $Z$ is learned only from $X^{\mathrm{o}}$ without direct access to the target $Y$.

## 3 METHODOLOGY

As illustrated in Fig. 2, our proposed model, CRIB, bypasses the problematic imputation stage by performing forecasts directly on the partially observed data. The architecture is composed of several key stages, each designed to address the challenges of learning from partially observed data. First, to handle the raw, sparse input, we introduce a Patching Embedding layer that employs a Temporal Convolutional Network (TCN) (Bai et al., 2018) to learn robust local feature representations from available data points. Second, to capture the complex global correlations that are disrupted by missingness, a Unified-Variate Attention mechanism models correlations across all patches simultaneously. Third, to ensure the model learns features that are stable and invariant to different missingness, especially under high missing rates, we introduce a Consistency Regularization scheme based on data augmentation. The entire learning process is guided by the IB principle, which provides a theoretical foundation for learning a representation that is maximally compressive against noise while being sufficiently informative for the forecasting task.

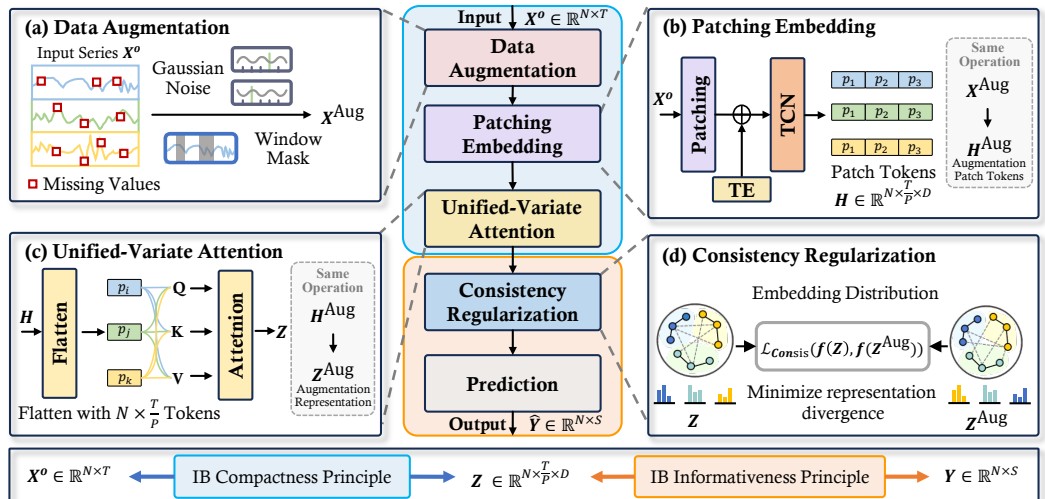

Figure 2: Overall framework of CRIB. (a) Data Augmentation creates a more challenging view of the partially observed data $X^{\text{o}}$ by generating an augmented version $X^{\text{Aug}}$. (b) The Patching Embedding layer converts the $X^{\text{o}}$ and $X^{\text{Aug}}$ into robust patch-level feature representations $H$ and $H^{\text{Aug}}$. (c) The Unified-Variate Attention mechanism models the global correlations between all the patches within $H$ and $H^{\text{Aug}}$ to produce refined representations $Z$ and $Z^{\text{Aug}}$. (d) Consistency Regularization aligns the representations from the original $Z$ and the augmented views $Z^{\text{Aug}}$. The entire process is guided by the IB principles of compactness and informativeness to produce the final forecast $\widehat{Y}$.

## 3.1 PATCHING EMBEDDING

To effectively enhance the semantic information that is not available in the partially observed, point-level time series $X^{\text{o}} \in \mathbb{R}^{N \times T}$, we first transform the input into a sequence of more meaningful patch-level representations (Nie et al., 2022). The series is partitioned into non-overlapping patches $\widehat{X} = \{\widehat{x}_i^{1:T/P} \mid i = 1, \cdots, N\} \in \mathbb{R}^{N \times (T/P) \times P}$ of length $P$. We choose $P$ such that the total length $T$ is evenly divisible. Consequently, this patching strategy reduces the sequence length from $T$ to $T/P$, thus remarkably lowering the memory and computational cost of attention calculation.

Next, to enable the following unified-variate attention mechanism to capture the temporal directionality of each variate $x_i^{1:T}$, we adopt the temporal encoding strategy inspired by vanilla transformer (Vaswani, 2017) as follows:

$$\text{TE}(t, m) = \begin{cases} \sin(t/10000^{2m/P}) & \text{if } m = 2k, \\ \cos(t/10000^{2m/P}) & \text{if } m = 2k+1, \end{cases} \tag{3}$$

where $m$ represents the $m$-th dimension of the feature. These temporal embeddings are added to the input patches to provide temporal information. Each patch, now containing a mix of observed values and temporal embeddings, is then processed by a TCN. It utilizes its efficient dilated convolution structure to transform sparse patches with missing values into dense feature representations $H \in \mathbb{R}^{N \times (T/P) \times D}$ that capture local temporal correlations.

## 3.2 UNIFIED-VARIATE ATTENTION

To model the complex, non-local correlations disrupted by missing data, we introduce a unified attention mechanism. Instead of using separate modules for inter- and intra-variate correlations among all the variates, our approach treats all patch representations uniformly. We first flatten the patch representations $H$ into a sequence $\widehat{H} \in \mathbb{R}^{(N \times T/P) \times D}$ with $N \times T/P$ tokens. A standard self-attention mechanism is then applied to this flattened sequence:

$$Z = \text{Attention}(Q, K, V) = \text{Softmax}(\frac{QK^{\top}}{\sqrt{D}})V, \tag{4}$$

where $Q, K, V \in \mathbb{R}^{(N \times T/P) \times D}$ are the linear projections of tokens $\widehat{H}$, and $\top$ denotes the matrix transpose. This allows the model to learn all possible correlations—both within a single variate's timeline (intra-variate) and across different variates (inter-variate)—without imposing strong, pre-defined structural biases. Such flexibility is particularly advantageous for sparse data, as it permits the model to rely on the most informative available signals, regardless of their origin. Unlike previous methods (Yi et al., 2024; Wang et al., 2024a) that employ strategies to reduce the memory and time costs of attention calculations, often at the expense of attention mechanism performance, we accelerate attention computation by patching time series. This can reduce the number of temporal tokens from $T$ to $T/P$, lowering the memory and computational cost of attention calculation by a factor of $P^2$, while enhancing the semantic-level information of the data.

### 3.3 FINAL PREDICTION

In CRIB, we implement the predictor using a simple Multi-Layer Perceptron (MLP) as follows:

$$\widehat{Y} = \text{Predictor}(Z) = \text{MLP}(Z) \in \mathbb{R}^{N \times S}, \tag{5}$$

where $S$ is the prediction length and $\text{MLP}(\cdot)$ denotes a simple two-layer fully connected network with a ReLU activation function applied between the layers. We deliberately employ a simple linear predictor to demonstrate that the forecasting performance of CRIB stems from the high-quality, robust representations $Z$ learned by our IB-guided attention mechanism, rather than employing a complex and powerful predictor (Liu et al., 2023; Zeng et al., 2023).

### 3.4 INFORMATION BOTTLENECK GUIDANCE

To enhance the quality of the learned representations $Z$ and improve forecasting accuracy, we propose an IB-based guidance. This guidance aims to balance compactness (filtering out irrelevant information) with informativeness (preserving relevant task-specific signals), allowing CRIB to focus on the most significant factors for accurate forecasting. In this section, we present how the compactness and informativeness principles are formulated and implemented in our framework. Full derivations are detailed in Appendix B.

#### 3.4.1 COMPACTNESS PRINCIPLE

The compactness principle, which aims to minimize the mutual information $I_\theta(Z; X^\text{o})$, forces the learned representation $Z$ to be a minimal sufficient statistic of the input. In our context, this encourages the model to discard non-essential information, which critically includes the noise introduced by the arbitrary locations of missing values. Following the variational inference (Voloshynovskiy et al., 2019), we derive a equivalent form of the compactness term in Eq. 1 as follows:

$$I_\theta(Z; X^\text{o}) = \mathbb{E}_{p(x^\text{o}, z)}[\log \frac{p(x^\text{o}, z)}{p(z) \cdot p(x^\text{o})}] = \mathbb{E}_{p(x^\text{o})}[D_{KL}(p(z|x^\text{o})||q(z))] - D_{KL}[p(z) \, || \, q(z)]. \tag{6}$$

Because of difficulty in posterior calculation and the non-negative property of Kullback-Leibler (KL) divergence, we use $p_\theta(z|x^\text{o})$ to approximate the true posterior distribution $p(z|x^\text{o})$ and bound Eq. 6:

$$I_\theta(Z; X^\text{o}) \leq \mathbb{E}_{p(x^\text{o})} D_{KL}[p_\theta(z|x^\text{o})||q(z)] \overset{\text{def}}{=} \mathcal{L}_{\text{Comp}}, \tag{7}$$

where we set isotropic Gaussian as the prior distribution of refined representations $Z$, i.e., $p(Z) = \mathcal{N}(0, I)$. Therefore, representations $Z$ are produced through a multivariate Gaussian distribution as:

$$p_\theta(Z|X^\text{o}) = \mathcal{N}(\mu_\theta(X^\text{o}), \text{diag}(\delta_\theta(X^\text{o}))), \tag{8}$$

where $\mu_\theta(\cdot)$ and $\sigma_\theta(\cdot)$ are designed as neural networks with parameter $\theta$. For training, we use the standard reparameterization trick (Kingma, 2013), $Z = \mu_\theta(X^\text{o}) + \sigma_\theta(X^\text{o}) \odot \epsilon$, which makes the objective in Eq. 7 differentiable without the need for stochastic estimation as follows:

$$\mathcal{L}_{\text{Comp}} = \frac{1}{2} \sum_{j=1}^{D} \left( 1 + \log \left( \sigma_\theta^{(j)}(X^\text{o}) \right)^2 - \left( \mu_\theta^{(j)}(X^\text{o}) \right)^2 - \left( \sigma_\theta^{(j)}(X^\text{o}) \right)^2 \right). \tag{9}$$

Here, $\mu_\theta^{(j)}(X^\text{o})$ and $\sigma_\theta^{(j)}(X^\text{o})$ denote the $j$-th element of the mean and standard deviation vectors.

### 3.4.2 INFORMATIVENESS PRINCIPLE

To balance the compactness objective, the informativeness principle ensures that the representation $Z$ preserves sufficient information for the forecasting task. To derive a tractable lower bound for the informativeness term, we follow the framework in (Voloshynovskiy et al., 1912) and Eq. 2, and assume that time series data follow a Gaussian distribution with fixed variance ($\sigma^2 I$), i.e., $q_\theta(y|z) = \mathcal{N}(\widehat{y}, \sigma^2 I)$ (Choi & Lee, 2023). The derivation proceeds as follows:

$$
\begin{aligned}
I_\theta(Y; Z) &= \mathbb{E}_{p(z,y)}[\log \frac{p(y|z)}{p(y)}] = \mathbb{E}_{p(z,y)}[\log \frac{q_\theta(y|z)}{p(y)}] + \mathbb{E}_{p(z,y)}[\log \frac{p(y|z)}{q_\theta(y|z)}], \\
&\geq \mathbb{E}_{p(z,y)}[\log q_\theta(y|z)] = -\mathbb{E}_{p(z,y)} \left[ \frac{1}{2\sigma^2} \|y - \widehat{y}\|^2 + \frac{T}{2} \log(2\pi\sigma^2) \right], \\
&\propto -\mathbb{E}_{p(z,y)} \left[ \|y - \widehat{y}\|^2 \right] \stackrel{\text{def}}{=} -\mathcal{L}_{\text{Pred}},
\end{aligned}
\tag{10}
$$

thus encouraging the model to extract task-relevant information from intermediate representations.

### 3.5 CONSISTENCY REGULARIZATION

While the IB framework encourages learning a compact representation, high missing rates can still lead to unstable training as shown in Appendix F.2, where the model overfits to the specific variate in a given time window (Choi & Lee, 2023). To mitigate this and enhance robustness, we introduce a consistency regularization scheme (Bachman et al., 2014; Laine & Aila, 2016). The core intuition is that the model's prediction should be invariant to the missingness. We achieve this by creating an augmented, more challenging view of the input, e.g, introducing additional noise to partially observed data. By enforcing that the representations learned from the observed and augmented views remain consistent, we regularize the model to handle missing values while stabilizing the refined representations instead of focusing excessively on a limited subset of observed data and neglecting crucial task-relevant variate correlations.

**Data Augmentation** Specifically, we generate $X^{\text{Aug}} \in \mathbb{R}^{N \times T}$ by applying two augmentations (Wen et al., 2020): (1) Random Masking, where we randomly select an additional 10% of the observed time points and set them to zero to simulate a more severe missingness scenario; and (2) Gaussian Noise, where we add noise $\epsilon \in \mathcal{N}(0, I)$ to all observed points to simulate sensor noise, enhancing the model's robustness to minor fluctuations in the input..

**Consistency Regularization** Then, through the same forward process as $X^{\text{o}}$, we can get their refined representations $Z^{\text{Aug}}$. The refined representations of observed and augmented data are regularized via the following consistency regularization loss function:

$$
\mathcal{L}_{\text{Consis}} = \frac{1}{N \times T/P} \sum_{i=1}^{N \times T/P} \|z_i - z_i^{\text{Aug}}\|^2,
\tag{11}
$$

where $N \times T/P$ is the number of the flattened tokens. By aligning the representations of the observed and augmented data, the model is encouraged to learn stable representations, thus enhancing robustness in scenarios with high missing rates. Furthermore, this consistency regularization can be seamlessly integrated into the overall optimization objective, complementing the IB theory to ensure that the refined representations retain essential task-relevant information while filtering out irrelevant noise from the missing values.

### 3.6 MODEL LEARNING

We have proposed a consistency-regularized method CRIB, which can complete MTSF-M tasks based on the IB theory. Overall, we optimize our model based on the following objective by combining all the introduced loss functions:

$$
\min_\theta \left[ \alpha \cdot (\mathcal{L}_{\text{Comp}}^\theta + \beta \cdot \mathcal{L}_{\text{Pred}}^\theta) + \gamma \cdot \mathcal{L}_{\text{Consis}} \right],
\tag{12}
$$

where $\alpha, \beta, \gamma \in \mathbb{R}$ are the preset balancing coefficients. This entire guidance helps CRIB extract the most important task-relevant information from the partially observed time series data while filtering out irrelevant noise introduced by missing values.

## 4 EXPERIMENT

In this section, extensive experiments on four real-world time series forecasting datasets are conducted to illustrate the effectiveness of our proposed CRIB. More experiments are in Appendix F.

### 4.1 EXPERIMENT SETTINGS

**Datasets.** We evaluate our model on four MTSF datasets: PEMS-BAY (Li et al., 2017), Metr-LA (Li et al., 2017), ETTh1 (Zhou et al., 2021), and Electricity (Wu et al., 2021). The key statistics and information of these datasets are summarized in Appendix C. To assess the model's effectiveness and robustness in handling missing values, we introduce synthetic missingness by randomly removing data points at varying missing rates of 20%, 40%, 60%, and 70% with three different missing patterns. During the experiments, we normalized the data to facilitate better model fitting.

**Baselines.** We chose 12 representative models for performance comparison. (1) Representative MTSF-M methods: BRITS (Cao et al., 2018), SAITS (Du et al., 2023), SPIN (Marisca et al., 2022), GRIN (Cini et al., 2021), and BiTGraph (Chen et al., 2023). (2) Transformer-based MTSF methods: iTransformer (Liu et al., 2023), PatchTST (Nie et al., 2022), and PAttn (Tan et al., 2024). (3) MLP-based and RNN-based MTSF methods: DLinear (Zeng et al., 2023), WPMixer (Murad et al., 2025), TimeXer (Wang et al., 2024b), and SegRNN (Lin et al., 2023).

Since the last two kinds of methods are not designed for MTSF-M tasks, we also study their variants by combining them with the current SOTA time series imputation method **TimesNet** (Wu et al., 2022) to build a two-stage framework, where TimesNet imputes and they predict. To simulate a practical scenario where the ground truth for missing values is unavailable during inference, Times-Net is trained on each dataset with a 10% missing rate and then imputes the observed data with 20%, 40%, 60%, and 70% missing rates. The original models and the variants are denoted as **Original** and **Imputed**, respectively. More baseline details are in Appendix D.

**Implementation Details.** We use Adam optimizer (Kingma, 2014) to learn the parameters of all models with $10^{-3}$ learning rate. The unified-variate attention of CRIB is configured with 2 layers and 4 heads, while the predictor is implemented as a simple 2-layer MLP. Both historical and future time window sizes are set to 24 for all methods, following the setting of BiTGraph (Chen et al., 2023). The patch length is set to 8, so every time series in a time window is patched into three tokens. The entire dataset is divided into training, validation, and testing sets with ratios of 60%, 20%, and 20%. Hyperparameters of all baselines are consistent with their original papers.

**Metrics.** In our experiments, we use Mean Absolute Error (MAE) and Mean Squared Error (MSE) to evaluate the forecasting performance of different methods.

### 4.2 MAIN RESULTS

Table 1: Performance comparison on four datasets with a point missing pattern (average MAE and MSE across 20% to 70% missing rate). Best is **Bold** and second-best is Underlined.

| Data | Metric | BiTGraph | BRITS | GRIN | SAITS | SPIN | SegRNN | | WPMixer | | iTransformer | | PatchTST | | DLinear | | TimeXer | | PAttn | | Ours | IMP |
|---|---|---|---|---|---|---|---|---|---|---|---|---|---|---|---|---|---|---|---|---|---|---|
| | | Original | Original | Original | Original | Original | Original | Imputed | Original | Imputed | Original | Imputed | Original | Imputed | Original | Imputed | Original | Imputed | Original | Imputed | Original | |
| **PEMS-BAY** | MAE | 0.413 | 0.366 | 0.350 | OOM | 0.402 | 0.120 | 0.178 | 0.155 | 0.201 | 0.107 | 0.125 | 0.129 | 0.139 | 0.156 | 0.148 | 0.125 | 0.135 | 0.110 | 0.148 | **0.093** | 13% |
| | MSE | 0.788 | 0.705 | 0.623 | OOM | 0.649 | 0.067 | 0.203 | 0.082 | 0.140 | 0.055 | 0.072 | 0.060 | 0.086 | 0.087 | 0.081 | 0.051 | 0.073 | 0.061 | 0.091 | **0.043** | 15% |
| **Metr-LA** | MAE | 0.445 | 0.366 | 0.389 | 0.451 | 0.625 | 0.318 | 0.314 | 0.356 | 0.342 | 0.273 | 0.290 | 0.313 | 0.306 | 0.399 | 0.366 | 0.321 | 0.298 | 0.302 | 0.294 | **0.262** | 4% |
| | MSE | 0.760 | 0.611 | 0.653 | 0.721 | 0.965 | 0.345 | 0.360 | 0.356 | 0.385 | 0.317 | 0.330 | 0.320 | 0.349 | 0.373 | 0.362 | 0.313 | 0.333 | 0.337 | 0.345 | **0.301** | 4% |
| **ETTh1** | MAE | 0.337 | 0.357 | 0.356 | 0.372 | 0.437 | 0.356 | 0.425 | 0.340 | 0.399 | 0.342 | 0.419 | 0.324 | 0.386 | 0.402 | 0.598 | 0.314 | 0.347 | 0.341 | 0.432 | **0.256** | 18% |
| | MSE | 0.387 | 0.421 | 0.400 | 0.457 | 0.468 | 0.479 | 0.477 | 0.432 | 0.417 | 0.408 | 0.473 | 0.385 | 0.435 | 0.560 | 0.682 | 0.377 | 0.370 | 0.416 | 0.470 | **0.269** | 27% |
| **Electricity** | MAE | 0.036 | 0.035 | 0.034 | 0.053 | 0.136 | 0.078 | 0.255 | 0.049 | 0.218 | 0.034 | 0.130 | 0.036 | 0.105 | 0.074 | 0.210 | 0.029 | 0.083 | 0.042 | 0.152 | **0.026** | 10% |
| | MSE | 0.113 | 0.059 | 0.061 | 0.266 | 0.358 | 1.010 | 1.286 | 0.172 | 0.286 | 0.054 | 0.547 | 0.092 | 0.379 | 0.404 | 2.000 | 0.064 | 0.100 | 0.115 | 0.864 | **0.044** | 18% |

The average performance comparisons between baselines and CRIB across four datasets are presented in Tab. 1, with full results in Appendix E and more missing patterns performance comparison in Fig. 3 and Appendix F.3. We denote out-of-memory and improvement as OOM and IMP, respectively. Based on these results, we summarize our observations (**Obs.**) as follows:

**Obs. ❶: CRIB demonstrates superior performance improvement in MTSF-M tasks.** As shown in Tabs. 1 and 4, Fig. 3, and Appendix F.3, CRIB achieves the lowest MAE and MSE across all

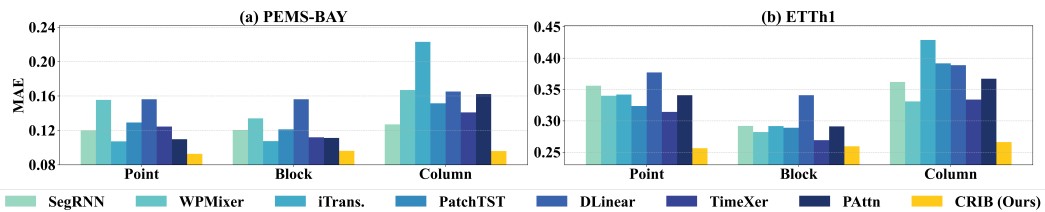

Figure 3: Average MAE on PEMS-BAY and ETTh1 with point, block, and column missing patterns.

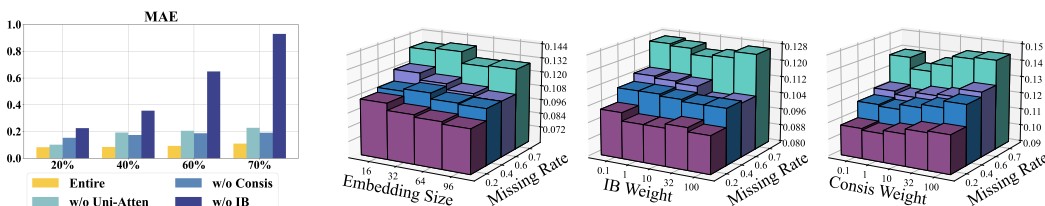

(a) Ablation Experiment Results          (b) Sensitivity Study Results

Figure 4: Ablation and Sensitivity experiment results on PEMS-BAY dataset of CRIB.

4 datasets and 3 missing patterns, with substantial improvements. Specifically, CRIB reduces the MAE by over 18% on ETTh1 and over 13% on PEMS-BAY compared to the strongest baseline. We attribute this improvement to our model's design, which integrates patch embedding, unified-variate attention, and consistency regularization under the IB principle, thus enabling CRIB to effectively filter noise from incomplete data while preserving essential predictive signals.

**Obs. ❷: Modern MTSF models have surpassed specialized models, and applying imputation to them is often detrimental.** Our experiments show that recent MTSF models (e.g., PatchTST), when applied directly to partially observed data, consistently outperform methods designed specifically for missing values (e.g., BiTGraph). Moreover, we find that applying an explicit imputation step to these modern models is often harmful; their performance on partially observed data is frequently superior to that of their two-stage variants, which use a pre-trained imputer (e.g., TimesNet). For example, PatchTST has an average $0.324$ MAE while its variant has a worse average $0.386$ MAE on the ETTh1 dataset. These phenomena suggest that imputation without direct ground-truth supervision can introduce erroneous values. This, in turn, distorts the underlying data distribution and corrupts variate correlations, ultimately degrading forecasting performance.

## 4.3 ABLATION AND SENSITIVITY STUDY

Table 2: Ablation study of consistency regularization under different missing rates on ETTh1.

| Method | Missing 20% | | Missing 40% | | Missing 60% | | Missing 70% | |
|---|---|---|---|---|---|---|---|---|
| | MAE | MSE | MAE | MSE | MAE | MSE | MAE | MSE |
| w/o Consis | 0.235±0.0022 | 0.264±0.0001 | 0.283±0.0011 | 0.276±0.0003 | 0.339±0.0021 | 0.405±0.0003 | 0.448±0.0020 | 0.574±0.0010 |
| **CRST-IB** | **0.220±0.0001** | **0.171±0.0001** | **0.251±0.0001** | **0.249±0.0001** | **0.267±0.0001** | **0.296±0.0001** | **0.288±0.0001** | **0.361±0.0008** |

We conduct ablation and parameter sensitivity studies to examine the contribution and robustness of each component in CRIB. The experiments are performed on PEMS-BAY dataset with four missing rates. In the **Ablation Study** (Fig. 4 (a)), we design three ablation experiments with configurations as follows: (1) **w/o Uni-Atten**: we replace the unified-variate attention mechanism with the vanilla attention mechanism. (2) **w/o Consis**: we remove the consistency regularization. (3) **w/o IB**: we remove the compactness and informativeness guidance of IB. In the **Sensitivity Study** (Fig. 4 (b)), we vary the weights assigned to the **Embedding Size**, **IB weight:** $\alpha$, and **Consis Weight:** $\gamma$ to study how each impacts model performance. We get observations as follows:

**Obs. ❸: Capturing variate correlations and ensuring consistency are critical for direct forecasting.** Both removing the unified-variate attention module (w/o Uni-Atten) and consistency regularization (w/o Consis) lead to a significant performance drop. This highlights the importance of

modeling inter-variate dependencies to comprehend the true data correlations, especially when values are missing. Moreover, as shown in Tab. 2, consistency regularization is crucial for improving the model's accuracy and stability, evidenced by lower prediction error and variance.

**Obs. ❹: The Information Bottleneck principle is the model's foundational component.** The most severe performance degradation occurs when the IB guidance is removed (w/o IB). The relative stability of the full model and the other variants, contrasted with the sharp decline of the w/o IB variant, confirms that the IB principle is fundamental to our model's ability to filter noise and achieve robust performance from incomplete data.

**Obs. ❺: CRIB is robust to hyperparameter variations, though over-regularization can be detrimental under high missing rates.** As shown in Fig. 4 (b), a larger embedding size generally correlates with better performance. However, the model remains effective even with a small embedding size (e.g., 32), demonstrating its efficiency in terms of computational and memory costs. For the IB and consistency regularization weights, we observe a trade-off. At low missing rates, higher weight values can improve accuracy. However, as the missing rate increases, excessively high weights tend to over-regularize the model, which can hinder its ability to capture complex variate correlations and thus degrade the final forecasting performance.

## 5 RELATED WORK

**Multivariate Time Series Forecasting with Missing Values**   Existing MTSF methods (Liu et al., 2023; Wang et al., 2024b; Hu et al., 2025d), which typically assume complete data, suffer significant performance degradation when applied to partially observed datasets. To address this issue, research on MTSF-M has emerged, focusing mainly on two directions: two-stage frameworks and end-to-end models. Two-stage methods combine imputation models (Cao et al., 2018; Cini et al., 2021; Marisca et al., 2022) with forecasting models (Liu et al., 2023; Wu et al., 2021; Tashiro et al., 2021). However, this decoupled design often leads to error propagation across stages (Chen et al., 2023), reducing overall forecasting accuracy. End-to-end approaches, on the other hand, aim to jointly impute missing values and perform forecasting by interleaving spatial and temporal modules (Yu et al., 2024). Despite their promise, these methods face a key limitation: the lack of ground truth for the missing values. As a result, the imputation process becomes noisy, which negatively impacts prediction performance. To address these limitations, we propose a direct prediction method CRIB, which integrates an IB-based Consistency Regularization to effectively identify relevant signals while filtering out redundant or noisy information, leading to more accurate forecasts.

**Information Bottleneck for Time Series**   The IB principle offers a framework for learning a compressed representation of an input that is maximally informative about a target task (Tishby et al., 2000). In time series, this is often implemented via Variational Autoencoders (VAEs) (Kingma, 2013; Voloshynovskiy et al., 2019). Existing methods like GP-VAE (Fortuin et al., 2020), MTS-IB (Ullmann et al., 2023), and RIB (Xu & Fekri, 2018) use the IB framework to model temporal dynamics. However, these approaches face a key limitation: a direct application of the IB principle can cause the model to concentrate too narrowly on observed features (Choi & Lee, 2023; Zhang et al., 2025b), thereby neglecting the broader variate correlations crucial for forecasting from incomplete data. In contrast to these works, our proposed CRIB applies the IB principle with a unified-attention mechanism and a consistency regularization, which encourages the model to capture stable representations and robust variate correlations even from sparse, incomplete inputs.

## 6 CONCLUSION

In this paper, we analyze the dominant 'imputation-then-prediction' paradigm for MTSF-M tasks. Our empirical analysis reveals a fundamental flaw in this framework: without direct supervision, imputation can corrupt data distribution and degrade, rather than improve, final forecasting accuracy. To address this, we propose a direct prediction paradigm and introduce CRIB, a novel framework designed to learn directly from incomplete data. By leveraging the IB principle with unified-variate attention and consistency regularization, CRIB effectively filters noise while capturing robust predictive signals from partial observations. Extensive experiments validate our method, showing that CRIB achieves a significant 18% improvement and confirms the superiority of direct prediction.

## 7 ETHICS STATEMENT

As our work only focuses on the time series forecasting problem, there is no potential ethical risk.

## 8 REPRODUCIBILITY STATEMENT

In the main text, we have formally defined the model architecture with equations. All the implementation details, including dataset descriptions, metrics, and experiment configurations are provided in the manuscript. Code is available in https://anonymous.4open.science/r/CRIB-F660.

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

## A  THE USE OF LARGE LANGUAGE MODELS (LLMs)

We used Large Language Models (LLMs) as auxiliary tools to assist with the writing process. They were used solely to polish the language and improve readability, with no influence over the research design, experimental implementation or analysis. We conceived and executed all methodological contributions, experiments, and conclusions independently.

## B  FULL DERIVATION

We illustrate the full derivation of the two terms of IB as follows.

**Compactness Principle:**

$$
\begin{aligned}
I_\theta(Z; X^{\mathrm{o}}) &= \mathbb{E}_{p(x^{\mathrm{o}},z)}[\log \frac{p(x^{\mathrm{o}}, z)}{p(z) \cdot p(x^{\mathrm{o}})}], \\
&= \mathbb{E}_{p(x^{\mathrm{o}},z)}[\log \frac{p(z|x^{\mathrm{o}}) \cdot p(x^{\mathrm{o}})}{p(z) \cdot p(x^{\mathrm{o}})}], \\
&= \mathbb{E}_{p(x^{\mathrm{o}},z)}[\log \frac{p(z|x^{\mathrm{o}})}{p(z)}], \\
&= \mathbb{E}_{p(x^{\mathrm{o}},z)}[\log \frac{p(z|x^{\mathrm{o}})}{p(z)} \cdot \frac{q(z)}{q(z)}], \\
&= \mathbb{E}_{p(x^{\mathrm{o}},z)}[\log \frac{p(z|x^{\mathrm{o}})}{q(z)}] - \mathbb{E}_{p(x^{\mathrm{o}},z)}[\log \frac{p(z)}{q(z)}], \\
&= \mathbb{E}_{p(x^{\mathrm{o}},z)}[\log \frac{p(z|x^{\mathrm{o}})}{q(z)}] - D_{KL}[p(z) \,||\, q(z)], \\
&= \mathbb{E}_{p(x^{\mathrm{o}})}[D_{KL}(p(z|x^{\mathrm{o}})||q(z))] - D_{KL}[p(z) \,||\, q(z)], \\
&\leq \mathbb{E}_{p(x^{\mathrm{o}})}[D_{KL}(p(z|x^{\mathrm{o}})||p(z))].
\end{aligned}
\tag{13}
$$

**Informativeness Principle:**

$$
\begin{aligned}
I_\theta(Y; Z) &= \mathbb{E}_{p(z,y)}[\log \frac{p(z, y)}{p(z) \cdot p(y)}], \\
&= \mathbb{E}_{p(z,y)}[\log \frac{p(y|z) \cdot p(z)}{p(y) \cdot p(z)}], \\
&= \mathbb{E}_{p(z,y)}[\log \frac{p(y|z)}{p(y)}], \\
&= \mathbb{E}_{p(z,y)}[\log \frac{p(y|z) \cdot q_\theta(y|z)}{p(y) \cdot q_\theta(y|z)}], \\
&= \mathbb{E}_{p(z,y)}[\log \frac{q_\theta(y|z)}{p(y)}] + \mathbb{E}_{p(z,y)}[\log \frac{p(y|z)}{q_\theta(y|z)}], \\
&= \mathbb{E}_{p(z,y)}[\log \frac{q_\theta(y|z)}{p(y)}] + \iint_{z,y} p(z) \cdot p(y|z) \cdot \log \frac{p(y|z)}{q_\theta(y|z)} \, dz dy, \\
&= \mathbb{E}_{p(z,y)}[\log \frac{q_\theta(y|z)}{p(y)}] + \int_z p(z) \cdot D_{KL}[p(y|z) \,||\, q_\theta(y|z)] \, dz \\
&\geq \mathbb{E}_{p(z,y)}[\log \frac{q_\theta(y|z)}{p(y)}], \\
&= \mathbb{E}_{p(z,y)}[\log q_\theta(y|z)] + H(Y), \\
&\geq \mathbb{E}_{p(z,y)}[\log q_\theta(y|z)].
\end{aligned}
\tag{14}
$$

The inequalities of the upper and lower bound in Eqs. (13) and (14) follow directly from the non-negativity of the KL-divergence and Entropy.

## C  DATASETS

Table 3: Dataset Statistics.

| Statistics | PEMS-BAY | Metr-LA | ETTh1 | Electricity |
|---|---|---|---|---|
| Timesteps (T) | 52116 | 34272 | 17420 | 26304 |
| Variates (N) | 325 | 207 | 7 | 321 |
| Frequency | 5 min | 5 min | 1 h | 1 h |
| Mean Value | 62.62 | 53.72 | 4.58 | 2538.79 |
| Std Value | 9.59 | 20.26 | 6.53 | 15027.57 |

We introduce information about datasets (Yu et al., 2024) as follows:

- **PEMS-BAY** (Li et al., 2017): This is a traffic speed dataset collected by the California Transportation Agencies' Performance Measurement System. It contains data collected by 325 sensors from January 1, 2017, to May 31, 2017. Each time series is sampled at a 5-minute interval, resulting in a total of 52,116 time slices.

- **METR-LA** (Li et al., 2017): This is a traffic speed dataset collected using loop detectors located on the LA County road network. It contains data collected by 207 sensors from March 1, 2012, to June 30, 2012. Each time series is sampled at a 5-minute interval, resulting in a total of 34,272 time slices.

- **ETTh1** (Zhou et al., 2021): This is a dataset used for forecasting tasks, containing data from a power plant. It consists of measurements taken hourly, including features such as power consumption, temperature, and pressure. Each time series is sampled at a 1-hour interval, resulting in a total of 17,420 time slices.

- **Electricity** (Wu et al., 2021): This dataset contains electricity consumption data. Each time series is sampled at a 1-hour interval, resulting in a total of 26,304 time slices.

## D  BASELINES

- **BiTGraph** (Chen et al., 2023): A model that jointly captures temporal correlations and spatial structures using biased Multi-Scale Instance PartialTCN and Biased GCN modules to effectively handle missing patterns in time series forecasting.

- **BRITS** (Cao et al., 2018): A bidirectional RNN model that imputes missing values directly within a recurrent dynamical system, effectively handling correlations, nonlinear dynamics, and general missing data patterns.

- **GRIN** (Cini et al., 2021): A graph neural network architecture designed for multivariate time series imputation, leveraging spatial and temporal message passing to reconstruct missing data.

- **SAITS** (Du et al., 2023): A self-attention-based model for multivariate time series imputation that uses diagonally-masked self-attention blocks to capture temporal and feature correlations.

- **SPIN** (Marisca et al., 2022): An attention-based spatial-temporal model for imputing multivariate time series, which avoids error propagation and does not rely on bidirectional encoding.

- **SegRNN** (Lin et al., 2023): An RNN-based model using segment-wise iterations and parallel multi-step forecasting to reduce recurrence and improve accuracy, speed, and efficiency over Transformer baselines.

- **WPMixer** (Murad et al., 2025): A MLP-based model (Wavelet Patch Mixer), leveraging the benefits of patching, multi-resolution wavelet decomposition, and mixing.

- **iTransformer** (Liu et al., 2023): A restructured Transformer for time series forecasting that captures multivariate correlations via attention on variate tokens, enhancing performance and efficiency across variable lookback windows.

- **PatchTST** (Nie et al., 2022): A Transformer-based model that segments time series into patches with a channel-independent design, enhancing long-term forecasting.

- **DLinear** (Zeng et al., 2023): A model that uses a simple MLP as the predictor to forecast accurately and has achieved great success.

- **TimeXer** (Wang et al., 2024b): A Transformer-based model that employs patch-level and variate-level representations respectively for endogenous and exogenous variables, with an endogenous global token as a bridge in-between.

- **PAttn** (Tan et al., 2024): A simple Transformer-based model combining patching with one-layer attention.

# E  FULL EXPERIMENTS

Table 4: Performance comparison of different models for multivariate time series forecasting with missing values. Missing rate is set at 20%, 40%, 60%, and 70%. The best results are highlighted in **Bold** and the second-best is highlighted in Underline.

| Data | Metric | BiTGraph | BRITS | GRIN | SAITS | SPIN | SegRNN | | WPMixer | | iTransformer | | PatchTST | | DLinear | | TimeXer | | PAttn | | Ours |
|---|---|---|---|---|---|---|---|---|---|---|---|---|---|---|---|---|---|---|---|---|---|---|
| | | Original | Original | Imputed | Original | Imputed | Original | Imputed | Original | Imputed | Original | Imputed | Original | Imputed | Original | Imputed | Original | Imputed | Original | Imputed | Original |
| **PEMS-BAY** | MAE@20% | 0.403 | 0.351 | 0.343 | OOM | 0.218 | 0.114 | 0.231 | 0.122 | 0.249 | 0.097 | 0.153 | 0.107 | 0.158 | 0.145 | 0.163 | 0.097 | 0.146 | 0.109 | 0.173 | **0.083** |
| | MSE@20% | 0.754 | 0.664 | 0.585 | OOM | 0.234 | 0.066 | 0.232 | 0.068 | 0.193 | 0.048 | 0.094 | 0.058 | 0.107 | 0.078 | 0.096 | 0.042 | 0.087 | 0.062 | 0.111 | **0.034** |
| | MAE@40% | 0.411 | 0.360 | 0.346 | OOM | 0.288 | 0.108 | 0.179 | 0.129 | 0.203 | 0.093 | 0.127 | 0.106 | 0.142 | 0.144 | 0.138 | 0.097 | 0.140 | 0.098 | 0.165 | **0.085** |
| | MSE@40% | 0.777 | 0.696 | 0.609 | OOM | 0.360 | 0.054 | 0.185 | 0.059 | 0.135 | 0.043 | 0.074 | 0.047 | 0.087 | 0.074 | 0.069 | 0.037 | 0.073 | 0.048 | 0.100 | **0.035** |
| | MAE@60% | 0.419 | 0.372 | 0.355 | OOM | 0.501 | 0.122 | 0.153 | 0.186 | 0.181 | 0.108 | 0.107 | 0.139 | 0.122 | 0.158 | 0.141 | 0.142 | 0.121 | 0.109 | 0.121 | **0.093** |
| | MSE@60% | 0.806 | 0.720 | 0.647 | OOM | 0.948 | 0.066 | 0.187 | 0.093 | 0.118 | 0.055 | 0.060 | 0.060 | 0.072 | 0.090 | 0.073 | 0.052 | 0.061 | 0.059 | 0.073 | **0.043** |
| | MAE@70% | 0.420 | 0.382 | 0.356 | OOM | 0.601 | 0.136 | 0.148 | 0.185 | 0.173 | 0.131 | 0.114 | 0.165 | 0.134 | 0.178 | 0.152 | 0.162 | 0.134 | 0.123 | 0.131 | **0.110** |
| | MSE@70% | 0.816 | 0.742 | 0.653 | OOM | 1.053 | 0.081 | 0.208 | 0.107 | 0.116 | 0.074 | 0.061 | 0.076 | 0.079 | 0.105 | 0.084 | 0.072 | 0.071 | 0.076 | 0.081 | **0.058** |
| **Metr-LA** | MAE@20% | 0.435 | 0.351 | 0.387 | 0.484 | 0.336 | 0.280 | 0.356 | 0.300 | 0.372 | 0.256 | 0.308 | 0.265 | 0.323 | 0.319 | 0.372 | 0.296 | 0.306 | 0.268 | 0.324 | **0.248** |
| | MSE@20% | 0.760 | 0.596 | 0.638 | 0.743 | 0.576 | 0.319 | 0.406 | 0.320 | 0.434 | 0.309 | 0.347 | 0.313 | 0.374 | 0.327 | 0.371 | 0.303 | 0.345 | 0.323 | 0.370 | **0.271** |
| | MAE@40% | 0.442 | 0.359 | 0.390 | 0.463 | 0.452 | 0.317 | 0.306 | 0.335 | 0.334 | 0.254 | 0.280 | 0.302 | 0.307 | 0.346 | 0.344 | 0.293 | 0.291 | 0.308 | 0.282 | **0.249** |
| | MSE@40% | 0.756 | 0.600 | 0.667 | 0.697 | 0.692 | 0.305 | 0.321 | 0.301 | 0.349 | 0.273 | 0.297 | 0.286 | 0.321 | 0.299 | 0.315 | 0.273 | 0.306 | 0.305 | 0.308 | **0.272** |
| | MAE@60% | 0.449 | 0.371 | 0.386 | 0.434 | 0.856 | 0.324 | 0.293 | 0.377 | 0.327 | 0.274 | 0.280 | 0.341 | 0.296 | 0.426 | 0.360 | 0.327 | 0.295 | 0.309 | 0.277 | **0.265** |
| | MSE@60% | 0.760 | 0.615 | 0.649 | 0.719 | 1.196 | 0.326 | 0.334 | 0.357 | 0.357 | 0.309 | 0.314 | 0.308 | 0.332 | 0.381 | 0.351 | 0.309 | 0.323 | 0.316 | 0.329 | **0.305** |
| | MAE@70% | 0.452 | 0.382 | 0.393 | 0.422 | 0.856 | 0.351 | 0.301 | 0.411 | 0.334 | 0.307 | 0.291 | 0.345 | 0.297 | 0.506 | 0.388 | 0.371 | 0.299 | 0.323 | 0.293 | **0.309** |
| | MSE@70% | 0.765 | 0.632 | 0.658 | 0.723 | 1.397 | 0.429 | 0.380 | 0.446 | 0.401 | 0.378 | 0.361 | 0.372 | 0.370 | 0.485 | 0.411 | 0.369 | 0.359 | 0.405 | 0.371 | **0.356** |
| **ETTh1** | MAE@20% | 0.257 | 0.232 | 0.234 | 0.369 | 0.232 | 0.250 | 0.394 | 0.239 | 0.386 | 0.238 | 0.417 | 0.236 | 0.398 | 0.265 | 0.538 | 0.232 | 0.341 | 0.243 | 0.432 | **0.220** |
| | MSE@20% | 0.307 | 0.378 | 0.282 | 0.457 | 0.191 | 0.217 | 0.331 | 0.201 | 0.323 | 0.204 | 0.374 | 0.201 | 0.382 | 0.235 | 0.508 | 0.199 | 0.296 | 0.215 | 0.387 | **0.171** |
| | MAE@40% | 0.278 | 0.317 | 0.338 | 0.349 | 0.320 | 0.303 | 0.403 | 0.285 | 0.383 | 0.330 | 0.493 | 0.286 | 0.419 | 0.334 | 0.616 | 0.274 | 0.340 | 0.321 | 0.506 | **0.251** |
| | MSE@40% | 0.316 | 0.373 | 0.386 | 0.430 | 0.346 | 0.310 | 0.384 | 0.284 | 0.352 | 0.303 | 0.556 | 0.274 | 0.445 | 0.365 | 0.646 | 0.264 | 0.323 | 0.296 | 0.518 | **0.249** |
| | MAE@60% | 0.394 | 0.432 | 0.399 | 0.380 | 0.598 | 0.394 | 0.445 | 0.380 | 0.404 | 0.364 | 0.382 | 0.356 | 0.355 | 0.443 | 0.631 | 0.343 | 0.346 | 0.368 | 0.399 | **0.267** |
| | MSE@60% | 0.493 | 0.499 | 0.498 | 0.482 | 0.667 | 0.566 | 0.539 | 0.514 | 0.451 | 0.454 | 0.454 | 0.437 | 0.418 | 0.623 | 0.763 | 0.435 | 0.393 | 0.470 | 0.462 | **0.296** |
| | MAE@70% | 0.420 | 0.449 | 0.455 | 0.390 | 0.599 | 0.477 | 0.459 | 0.456 | 0.424 | 0.436 | 0.384 | 0.417 | 0.373 | 0.567 | 0.608 | 0.408 | 0.363 | 0.432 | 0.390 | **0.288** |
| | MSE@70% | 0.432 | 0.436 | 0.433 | 0.461 | 0.669 | 0.824 | 0.653 | 0.728 | 0.541 | 0.671 | 0.507 | 0.630 | 0.497 | 1.016 | 0.812 | 0.608 | 0.468 | 0.683 | 0.514 | **0.361** |
| **Electricity** | MAE@20% | 0.029 | 0.018 | 0.020 | 0.050 | 0.021 | 0.051 | 0.357 | 0.026 | 0.348 | 0.020 | 0.172 | 0.020 | 0.143 | 0.039 | 0.169 | 0.019 | 0.129 | 0.022 | 0.197 | **0.015** |
| | MSE@20% | 0.123 | 0.026 | 0.015 | 0.243 | 0.028 | 0.478 | 0.976 | 0.035 | 0.506 | 0.027 | 0.266 | 0.028 | 0.236 | 0.075 | 0.412 | 0.022 | 0.158 | 0.027 | 0.499 | **0.012** |
| | MAE@40% | 0.028 | 0.030 | 0.030 | 0.051 | 0.031 | 0.066 | 0.281 | 0.038 | 0.232 | 0.031 | 0.153 | 0.029 | 0.120 | 0.058 | 0.194 | 0.024 | 0.089 | 0.042 | 0.188 | **0.023** |
| | MSE@40% | 0.116 | 0.054 | 0.066 | 0.266 | 0.070 | 0.722 | 1.072 | 0.083 | 0.267 | 0.035 | 0.937 | 0.045 | 0.526 | 0.185 | 1.835 | 0.038 | 0.093 | 0.064 | 1.200 | **0.028** |
| | MAE@60% | 0.038 | 0.044 | 0.041 | 0.053 | 0.223 | 0.089 | 0.210 | 0.056 | 0.159 | 0.041 | 0.118 | 0.040 | 0.089 | 0.086 | 0.228 | 0.032 | 0.060 | 0.048 | 0.133 | **0.030** |
| | MSE@60% | 0.109 | 0.054 | 0.059 | 0.258 | 0.667 | 1.185 | 1.412 | 0.197 | 0.184 | 0.065 | 0.700 | 0.110 | 0.496 | 0.465 | 2.348 | 0.062 | 0.073 | 0.139 | 1.248 | **0.047** |
| | MAE@70% | 0.049 | 0.048 | 0.045 | 0.058 | 0.271 | 0.107 | 0.174 | 0.075 | 0.135 | 0.046 | 0.079 | 0.052 | 0.067 | 0.111 | 0.249 | 0.041 | 0.055 | 0.056 | 0.090 | **0.038** |
| | MSE@70% | 0.104 | 0.102 | 0.105 | 0.296 | 0.669 | 1.655 | 1.684 | 0.374 | 0.187 | 0.091 | 0.287 | 0.184 | 0.258 | 0.888 | 2.405 | 0.135 | 0.075 | 0.230 | 0.510 | **0.091** |

# F  EXTRA EXPERIMENTS

## F.1  FORECASTING RESULTS VISUALIZATION

We present a spatial visualization of forecasting results to demonstrate the effectiveness of CRIB under varying missing rates. Fig. 5 shows the final timestamp in the historical time window and the first forecasting timestamp on the PEMS-BAY dataset. At lower missing rates (20% and 40%), by effectively leveraging inter-variate correlations extracted from the data, CRIB accurately predicts the future values. Even at higher missing rates (60% and 70%), CRIB can maintain stable performance and predict the spatial distribution of the PEMS-BAY datasets. These findings underscore CRIB's capability to handle incomplete data and produce reliable predictions.

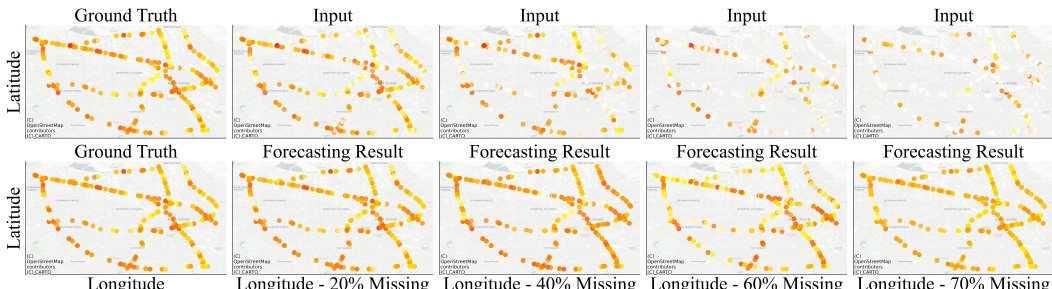

Figure 5: Visualization of the input and forecasting results of CRIB on the PEMS-BAY dataset with missing rates from 20% to 70%.

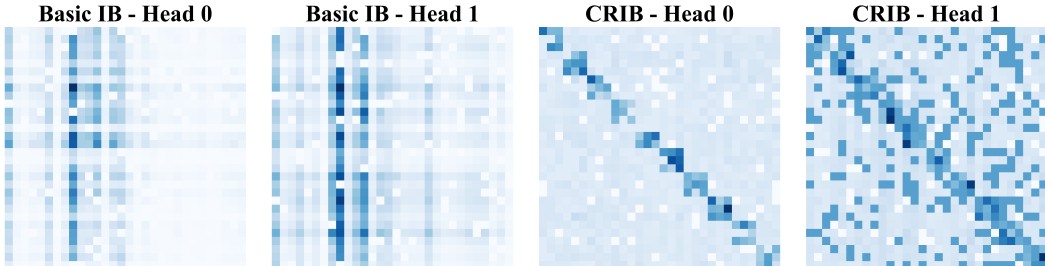

Figure 6: Visualization comparison of attention maps on the Metr-LA dataset with 60% missing values. **Left**: Two attention maps of the direct application of IB on the standard Transformer. **Right**: Two attention maps of CRIB.

## F.2 UNIFIED-VARIATE ATTENTION MAPS VISUALIZATION

In Fig. 6, we compare visualizations of directly applying IB on the Transformer with our proposed CRIB. In the first experiment, a transformer model serves as the predictor. The **left** two figures clearly show that directly applying IB to the model would force the model to focus on a few specific values (straight line attention), thereby neglecting global information. In contrast, the **right** figures reveal that CRIB can not only capture the original intra-variate temporal correlations in one attention head but also effectively uncovers cross-variate correlations in another, rather than relying solely on raw correlations. As a result, the final forecasting performance is improved remarkably by our unified-variate attention mechanism and consistency regularization scheme.

## F.3 EXPERIMENTS ON VARIOUS MISSING PATTERNS

Figures 7 to 14 present the main forecasting results, comparing our proposed model, CRIB, against state-of-the-art baselines. The results clearly show that CRIB consistently achieves the lowest MAE and MSE across all evaluated scenarios. This superiority holds true for both the PEMS-BAY and ETTh1 datasets, under point, block, and column missing patterns, and across a wide range of missing rates from 20% to 70%. Notably, while the performance of most baseline models degrades significantly as the missing rate increases, CRIB maintains its superior performance and stability. This demonstrates the robustness and effectiveness of our direct-prediction approach, validating its superiority over existing methods, especially in challenging high-missing-rate environments.

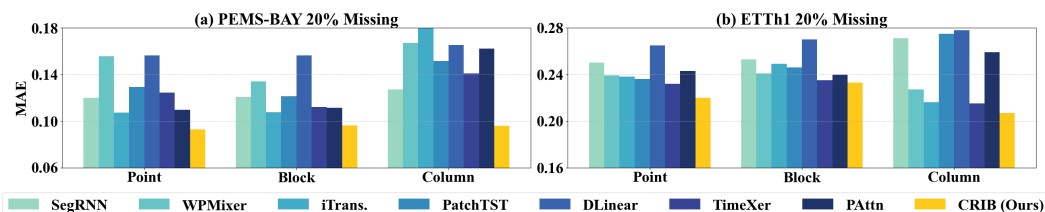

Figure 7: MAE comparison on PEMS-BAY and ETTh1 with point, block, and column missing patterns on 20% missing rate.

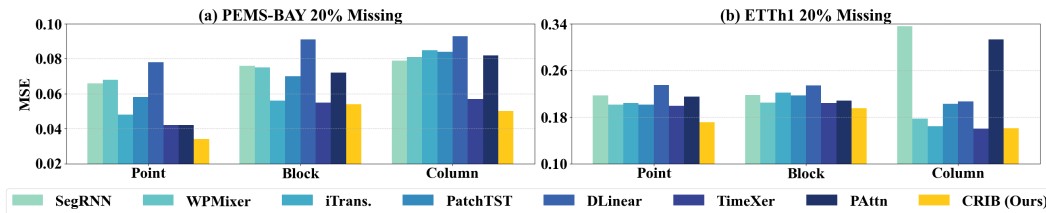

Figure 8: MSE comparison on PEMS-BAY and ETTh1 with point, block, and column missing patterns on 20% missing rate.

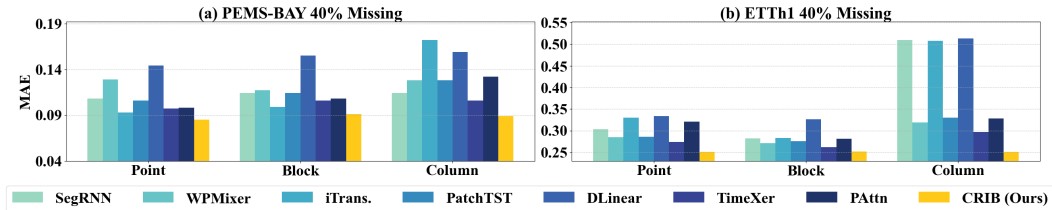

Figure 9: MAE comparison on PEMS-BAY and ETTh1 with point, block, and column missing patterns on 40% missing rate.

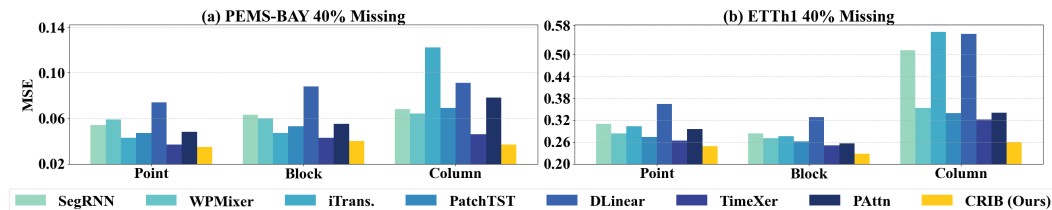

Figure 10: MSE comparison on PEMS-BAY and ETTh1 with point, block, and column missing patterns on 40% missing rate.

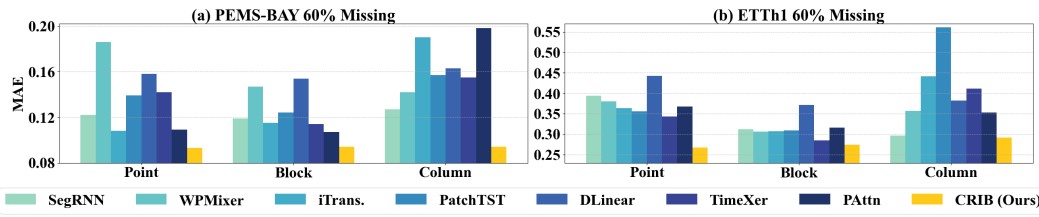

Figure 11: MAE comparison on PEMS-BAY and ETTh1 with point, block, and column missing patterns on 60% missing rate.

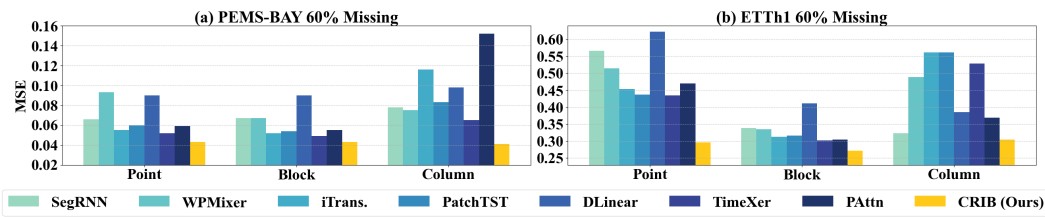

Figure 12: MSE comparison on PEMS-BAY and ETTh1 with point, block, and column missing patterns on 60% missing rate.

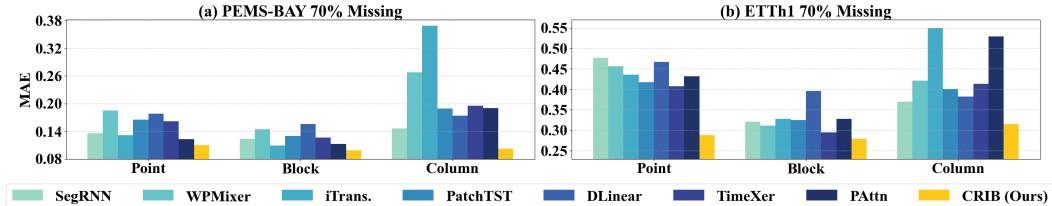

Figure 13: MAE comparison on PEMS-BAY and ETTh1 with point, block, and column missing patterns on 70% missing rate.

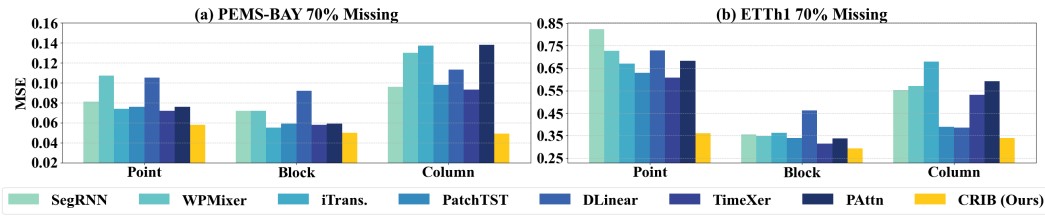

Figure 14: MSE comparison on PEMS-BAY and ETTh1 with point, block, and column missing patterns on 70% missing rate.

# G REBUTTAL RESPONSES

## G.1 MOTIVATION AND CONTRIBUTION

We acknowledge that the limitations of the 'imputation-then-prediction' paradigm have been discussed in prior works, and that direct prediction and Information Bottleneck (IB) frameworks are established in previous literature. However, as detailed in our Introduction and Related Work, existing end-to-end methods still incorporate explicit imputation modules within their framework. Consequently, they structurally remain within the "imputation-then-prediction" paradigm, merely shifting the imputation to a latent or module scale. The distribution shift and correlation destruction shown in Fig. 1 are calculated based on real data and model outputs (t-SNE and Correlation Map), not a Toy Example. This intuitively reveals the risks of unsupervised imputation. Our contribution extends beyond a simple critique to provide a systematic empirical re-evaluation and CRIB.

Table 5: Performance comparison on PEMS-BAY dataset (40% Missing).

| Metric | CRIB | BiTGraph | DLinear | TimesNet + DLinear | TimeXer | TimesNet + TimeXer |
|--------|------|----------|---------|--------------------|---------|--------------------|
| MAE | **0.093** | 0.413 | 0.156 | 0.148 | 0.125 | 0.135 |
| MSE | **0.043** | 0.788 | 0.087 | 0.081 | 0.051 | 0.073 |

❶ **Detrimental Imputation:** We demonstrate that imputation without ground truth is often harmful. As visualized in Fig. 1, methods following the "imputation-then-prediction" paradigm fail to recover the true data distribution and instead reinforce biased patterns from partial observations.

❷ **Performance Degradation:** We provide counter-intuitive evidence that imputation actively harms prediction accuracy. For instance, equipping the predictor TimeXer with the SOTA imputer Times-Net increases the MAE from 0.125 to 0.135 as shown in Tab. 5. It also has no help in understanding the data distribution and variate correlations as demonstrated in Fig. 1.

❸ **Limitations of Vanilla IB:** We further observe that a naive application of the IB is insufficient. Fig. 6 ( Appendix F.2) shows that a direct IB-based Transformer yields degenerate attention maps, biasing the model towards local linearity and neglecting global dependencies. Furthermore, results in 2 indicate that CRIB without the consistency loss exhibits significantly higher variance. We resolve this by integrating IB with Consistency Regularization and Unified-Variate Attention to effectively filter noise introduced by missingness.

## G.2 NEW BASELINES AND DATASETS

**Justification:** We selected the combination of TimesNet (Imputation) and DLinear (Prediction) specifically to demonstrate that even current Time Series Forecasting models fail when applied within the "imputation-then-prediction" framework. To make it more solid, we conducted additional experiments on the combination of TimesNet (Imputation) and TimeXer (Prediction) in Fig. 1, which reveal similar issues, demonstrating that the imputation without ground truth is detrimental.

**New Comparisons:** To ensure a comprehensive and fair evaluation, we selected baselines based on their code availability and their applicability to general MTSF-M scenarios, which typically lack predefined graph structures. Accordingly, we have expanded our experimental validation to include comparisons with **CSDI** (Tashiro et al., 2021), **ImputeFormer** (Nie et al., 2024), **Neural-CDE** (Kidger et al., 2020), and **TimesNet** (Wu et al., 2022) across a broader set of datasets. We exclude GinAR (Yu et al., 2024) and S4M (Peng et al., 2025) solely due to reproducibility issues with their publicly available code.

**Conclusion:** As shown in Tab. 7, extensive experiments on these datasets confirm that CRIB consistently achieves state-of-the-art performance, exhibiting superior robustness across varying missing rates compared to both direct prediction and imputation-based baselines.

Table 6: Statistics of the 10 real-world datasets used in our experiments.

| Statistics | ETTh1 | ETTh2 | ETTm1 | ETTm2 | Electricity | PEMS-BAY | Metr-LA | BeijingAir | Weather | Exchange |
|---|---|---|---|---|---|---|---|---|---|---|
| Time Steps | 17,420 | 17,420 | 69,680 | 69,680 | 26,304 | 52,116 | 34,272 | 36,000 | 52,696 | 7,588 |
| Variates | 7 | 7 | 7 | 7 | 321 | 325 | 207 | 7 | 21 | 8 |

Table 7: Performance comparison on different datasets with varying mask ratios. The best is **Bold**.

| Dataset | Mask | Metric | CRIB (Ours) | SegRNN | WPMixer | iTransformer | PatchTST | DLinear | PAttn | CSDI | NeuralCDE | ImputeFormer | TimesNet |
|---|---|---|---|---|---|---|---|---|---|---|---|---|
| ETTh2 | 0.0 | MAE | **0.094** | 0.096 | 0.095 | 0.098 | 0.096 | 0.098 | 0.096 | 0.113 | 0.197 | 0.101 | 0.097 |
| | | MSE | **0.023** | 0.024 | 0.024 | 0.025 | 0.024 | 0.025 | 0.025 | 0.033 | 0.074 | 0.026 | 0.025 |
| | 0.2 | MAE | **0.107** | 0.115 | 0.116 | 0.115 | 0.109 | 0.142 | 0.116 | 0.255 | 0.244 | 0.122 | 0.121 |
| | | MSE | **0.028** | 0.031 | 0.030 | 0.033 | 0.030 | 0.040 | 0.033 | 0.192 | 0.105 | 0.034 | 0.033 |
| | 0.4 | MAE | **0.131** | 0.148 | 0.147 | 0.298 | 0.215 | 0.181 | 0.289 | 0.404 | 0.276 | 0.140 | 0.153 |
| | | MSE | **0.039** | 0.047 | 0.045 | 0.180 | 0.125 | 0.065 | 0.173 | 0.399 | 0.134 | 0.049 | 0.048 |
| | 0.6 | MAE | **0.154** | 0.183 | 0.202 | 0.270 | 0.216 | 0.253 | 0.243 | 0.618 | 0.347 | 0.173 | 0.209 |
| | | MSE | **0.055** | 0.072 | 0.086 | 0.170 | 0.121 | 0.135 | 0.133 | 0.840 | 0.210 | 0.061 | 0.092 |
| | 0.7 | MAE | **0.172** | 0.205 | 0.218 | 0.218 | 0.186 | 0.325 | 0.196 | 0.798 | 0.413 | 0.174 | 0.246 |
| | | MSE | **0.069** | 0.093 | 0.104 | 0.111 | 0.083 | 0.228 | 0.085 | 1.303 | 0.303 | 0.071 | 0.132 |
| ETTm1 | 0.0 | MAE | **0.2000** | 0.2265 | 0.2405 | 0.2309 | 0.2379 | 0.2772 | 0.2423 | 0.4759 | 0.3331 | 0.2999 | 0.2100 |
| | | MSE | **0.2008** | 0.2302 | 0.2741 | 0.2514 | 0.2648 | 0.3504 | 0.2782 | 0.8854 | 0.4218 | 0.2682 | 0.2018 |
| | 0.2 | MAE | **0.2368** | 0.2616 | 0.2900 | 0.2909 | 0.2779 | 0.3711 | 0.2971 | 0.5198 | 0.3809 | 0.3271 | 0.2584 |
| | | MSE | **0.2586** | 0.3068 | 0.3593 | 0.3751 | 0.3486 | 0.5499 | 0.3864 | 0.9345 | 0.5177 | 0.3282 | 0.2992 |
| | 0.4 | MAE | **0.2734** | 0.3005 | 0.3308 | 0.3593 | 0.3456 | 0.4418 | 0.3592 | 0.5787 | 0.4391 | 0.3571 | 0.3040 |
| | | MSE | **0.3382** | 0.4007 | 0.4645 | 0.4851 | 0.4671 | 0.7412 | 0.4801 | 1.0499 | 0.6643 | 0.4749 | 0.4093 |
| | 0.6 | MAE | **0.3414** | 0.3678 | 0.4171 | 0.4365 | 0.4091 | 0.5551 | 0.4191 | 0.7038 | 0.5457 | 0.4315 | 0.3993 |
| | | MSE | **0.5200** | 0.5960 | 0.7328 | 0.7580 | 0.6769 | 1.1067 | 0.7022 | 1.3801 | 0.9573 | 0.6978 | 0.6525 |
| | 0.7 | MAE | **0.4013** | 0.4266 | 0.4863 | 0.5034 | 0.4743 | 0.6508 | 0.4816 | 0.8287 | 0.6278 | 0.4845 | 0.4828 |
| | | MSE | **0.6956** | 0.8006 | 0.9928 | 1.0663 | 0.9325 | 1.4721 | 0.9837 | 1.7733 | 1.2710 | 0.9035 | 0.8988 |
| ETTm2 | 0.0 | MAE | **0.0746** | 0.0802 | 0.0821 | 0.0781 | 0.0824 | 0.0918 | 0.0828 | 0.1313 | 0.1327 | 0.0813 | 0.0782 |
| | | MSE | **0.0152** | 0.0175 | 0.0182 | 0.0161 | 0.0182 | 0.0217 | 0.0187 | 0.0410 | 0.0363 | 0.0160 | 0.0165 |
| | 0.2 | MAE | **0.0885** | 0.0956 | 0.1059 | 0.1020 | 0.1005 | 0.1405 | 0.1024 | 0.2687 | 0.1760 | 0.0888 | 0.1087 |
| | | MSE | **0.0203** | 0.0231 | 0.0266 | 0.0284 | 0.0251 | 0.0391 | 0.0278 | 0.1976 | 0.0591 | 0.0222 | 0.0260 |
| | 0.4 | MAE | **0.1037** | 0.1159 | 0.1355 | 0.2673 | 0.2041 | 0.1743 | 0.2441 | 0.4131 | 0.1984 | 0.1088 | 0.1417 |
| | | MSE | **0.0258** | 0.0305 | 0.0394 | 0.1688 | 0.1210 | 0.0601 | 0.1490 | 0.4003 | 0.0765 | 0.0267 | 0.0406 |
| | 0.6 | MAE | **0.1266** | 0.1432 | 0.1790 | 0.2651 | 0.2425 | 0.2412 | 0.2398 | 0.6221 | 0.2479 | 0.1360 | 0.1986 |
| | | MSE | **0.0384** | 0.0477 | 0.0685 | 0.1671 | 0.1423 | 0.1215 | 0.1448 | 0.8293 | 0.1194 | 0.0424 | 0.0811 |
| | 0.7 | MAE | **0.1467** | 0.1650 | 0.1899 | 0.1905 | 0.1803 | 0.3104 | 0.1728 | 0.8024 | 0.3130 | 0.1602 | 0.2317 |
| | | MSE | **0.0520** | 0.0619 | 0.0808 | 0.0861 | 0.0779 | 0.2056 | 0.0717 | 1.2968 | 0.1857 | 0.0569 | 0.1142 |
| Weather | 0.0 | MAE | **0.028** | 0.031 | 0.030 | 0.030 | 0.031 | 0.034 | 0.031 | 0.051 | 0.051 | 0.035 | 0.028 |
| | | MSE | **0.016** | 0.021 | 0.019 | 0.018 | 0.019 | 0.023 | 0.020 | 0.042 | 0.029 | 0.020 | 0.016 |
| | 0.2 | MAE | **0.038** | 0.042 | 0.050 | 0.044 | 0.048 | 0.099 | 0.046 | 0.173 | 0.073 | 0.041 | 0.062 |
| | | MSE | **0.025** | 0.028 | 0.025 | 0.031 | 0.027 | 0.045 | 0.032 | 0.220 | 0.036 | 0.028 | 0.033 |
| | 0.4 | MAE | **0.050** | 0.050 | 0.074 | 0.152 | 0.150 | 0.135 | 0.157 | 0.300 | 0.095 | 0.051 | 0.088 |
| | | MSE | **0.033** | 0.037 | 0.036 | 0.167 | 0.156 | 0.078 | 0.140 | 0.500 | 0.060 | 0.033 | 0.047 |
| | 0.6 | MAE | **0.062** | 0.066 | 0.102 | 0.175 | 0.169 | 0.197 | 0.168 | 0.466 | 0.150 | 0.066 | 0.128 |
| | | MSE | **0.057** | 0.061 | 0.062 | 0.223 | 0.169 | 0.178 | 0.163 | 1.099 | 0.150 | 0.064 | 0.089 |
| | 0.7 | MAE | **0.070** | 0.076 | 0.121 | 0.101 | 0.118 | 0.254 | 0.100 | 0.586 | 0.204 | 0.076 | 0.166 |
| | | MSE | **0.075** | 0.081 | 0.084 | 0.138 | 0.128 | 0.318 | 0.120 | 1.707 | 0.284 | 0.083 | 0.154 |
| Exchange | 0.0 | MAE | **0.0184** | 0.0185 | 0.0187 | 0.0190 | 0.0186 | 0.0205 | 0.0187 | 0.0264 | 0.3223 | 0.0344 | 0.0195 |
| | | MSE | **0.0009** | 0.0010 | 0.0010 | 0.0010 | 0.0010 | 0.0011 | 0.0010 | 0.0017 | 0.1724 | 0.0031 | 0.0011 |
| | 0.2 | MAE | **0.0217** | 0.0292 | 0.0324 | 0.0288 | 0.0273 | 0.0795 | 0.0227 | 0.1992 | 0.2969 | 0.0364 | 0.0458 |
| | | MSE | **0.0016** | 0.0031 | 0.0024 | 0.0024 | 0.0019 | 0.0126 | 0.0016 | 0.2089 | 0.1592 | 0.0029 | 0.0041 |
| | 0.4 | MAE | **0.0253** | 0.0629 | 0.0783 | 0.1899 | 0.1416 | 0.1469 | 0.1408 | 0.4358 | 0.3326 | 0.0431 | 0.0786 |
| | | MSE | **0.0015** | 0.0129 | 0.0127 | 0.0770 | 0.0884 | 0.0429 | 0.1047 | 0.5718 | 0.2146 | 0.0034 | 0.0130 |
| | 0.6 | MAE | **0.0363** | 0.1165 | 0.1364 | 0.2439 | 0.1978 | 0.2599 | 0.1283 | 0.7911 | 0.4334 | 0.0608 | 0.1357 |
| | | MSE | **0.0031** | 0.0363 | 0.0405 | 0.1060 | 0.1129 | 0.1394 | 0.0831 | 1.3714 | 0.3814 | 0.0071 | 0.0404 |
| | 0.7 | MAE | **0.0517** | 0.1418 | 0.1836 | 0.2865 | 0.1321 | 0.3778 | 0.0718 | 1.0766 | 0.5412 | 0.0846 | 0.1991 |
| | | MSE | **0.0058** | 0.0490 | 0.0697 | 0.1413 | 0.0581 | 0.3011 | 0.0169 | 2.2850 | 0.5912 | 0.0128 | 0.0881 |
| BejingAir | 0.0 | MAE | **0.2526** | 0.2552 | 0.2571 | 0.2606 | 0.2609 | 0.2707 | 0.2601 | 0.3431 | 0.3167 | 0.2533 | 0.2583 |
| | | MSE | **0.2929** | 0.3026 | 0.3085 | 0.3156 | 0.3008 | 0.3263 | 0.3102 | 0.5141 | 0.3779 | 0.3189 | 0.2939 |
| | 0.2 | MAE | **0.2758** | 0.2849 | 0.2896 | 0.2923 | 0.2866 | 0.3204 | 0.2949 | 0.4035 | 0.3478 | 0.2770 | 0.2922 |
| | | MSE | **0.3317** | 0.3467 | 0.3651 | 0.3655 | 0.3421 | 0.3853 | 0.3603 | 0.6010 | 0.4196 | 0.3496 | 0.3488 |
| | 0.4 | MAE | **0.3106** | 0.3188 | 0.3312 | 0.3545 | 0.3278 | 0.3620 | 0.3480 | 0.4714 | 0.3950 | 0.3297 | 0.3322 |
| | | MSE | **0.4120** | 0.4285 | 0.4601 | 0.4820 | 0.4305 | 0.4760 | 0.4689 | 0.7584 | 0.5474 | 0.4645 | 0.4422 |
| | 0.6 | MAE | **0.3728** | 0.3860 | 0.4011 | 0.4124 | 0.3976 | 0.4362 | 0.4142 | 0.5930 | 0.4750 | 0.4006 | 0.4017 |
| | | MSE | **0.5862** | 0.5939 | 0.6503 | 0.6573 | 0.6458 | 0.6799 | 0.6624 | 1.1378 | 0.7803 | 0.6110 | 0.6275 |
| | 0.7 | MAE | **0.4337** | 0.4412 | 0.4632 | 0.4691 | 0.4613 | 0.5022 | 0.4673 | 0.7057 | 0.5456 | 0.4549 | 0.4789 |
| | | MSE | 0.8010 | **0.7595** | 0.8730 | 0.8874 | 0.8540 | 0.9358 | 0.8745 | 1.5761 | 1.0151 | 0.8327 | 0.9234 |

## G.3 Natural Missingness

We have conducted the experiments on the **AQI** dataset (Yi et al., 2016) with naturally occurring missing data as suggested. To fairly compare 'direct prediction' against the 'imputation-then-prediction' strategy, we designed the experiment as follows:

- **Imputation Model Setup:** We first pre-trained a TimesNet model (Wu et al., 2022) on the AQI training set. To enable learning for imputation, we applied a 10% point missing mask to the observed values during training. We selected a 10% masking rate because our statistical analysis showed that the natural missing rate of the AQI dataset is approximately 10%.

- **Two-Stage Process (AQI_IMP):** We used this pre-trained TimesNet to impute the naturally occurring Not a Number (NaN) values across the entire AQI dataset. We then trained and evaluated the downstream forecasting models on this fully imputed dataset.

- **Direct Prediction (AQI_ORI):** For comparison, we trained and evaluated the models directly on the original AQI dataset containing natural missing values.

- **Evaluation Metric:** To ensure a valid comparison, the MAE and MSE metrics were calculated only on the observed data points (excluding original NaNs from the loss calculation via masking), as the ground truth for the naturally missing parts is unknown.

**Conclusion:** The experimental results are presented in Tab. 8. We observed that for multiple forecasting models, using TimesNet to impute the missing values actually degraded the prediction performance compared to direct prediction. This negative impact empirically corroborates our paper's central claim: in the absence of ground truth supervision, the 'imputation-then-prediction' strategy can introduce noise and corrupt the data distribution, making it suboptimal compared to 'direct prediction' methods like CRIB.

Table 8: Performance comparison on AQI datasets (Original vs. Imputed). The best results are highlighted in **Bold**, and the second-best is highlighted in Underline.

| Dataset | Metric | CRIB | SegRNN | WPMixer | iTransformer | PatchTST | DLinear | PAttn | CSDI | NCDE | ImpFormer | TimesNet |
|---------|--------|------|--------|---------|--------------|----------|---------|-------|------|------|-----------|----------|
| **AQI_ORI** | MAE | **0.555** | 0.604 | 0.624 | 0.608 | 0.627 | 0.598 | 0.621 | 0.858 | 0.798 | 0.795 | 0.648 |
| | MSE | **0.663** | 0.804 | 0.843 | 0.818 | 0.844 | 0.741 | 0.843 | 1.448 | 1.438 | 1.313 | 0.925 |
| **AQI_IMP** | MAE | **0.616** | 0.650 | 0.665 | 0.668 | 0.666 | 0.653 | 0.663 | 0.941 | 0.946 | 0.857 | 0.733 |
| | MSE | **0.844** | 0.966 | 0.986 | 1.012 | 0.993 | 0.893 | 0.995 | 1.775 | 1.928 | 1.543 | 1.206 |

### G.4 TRAINING COST

To ensure fair comparisons, we train all baseline models from scratch using identical dataset splits and experimental protocols, as detailed in Sec. 4.1. We evaluate computational efficiency by reporting the memory footprint and parameter counts of every model on ETTh1 as follows.

**Conclusion:** The results demonstrate that CRIB maintains a computational cost comparable to efficient Transformer baselines (e.g., PatchTST (Nie et al., 2022)) while being more lightweight than complex methods such as CSDI (Tashiro et al., 2021) and TimesNet (Wu et al., 2022).

Table 9: Comparison of model efficiency in terms of parameter count and memory cost. CRIB achieves a balanced trade-off between performance and efficiency.

| Model | Parameters | Memory (MB) |
|-------|-----------|-------------|
| **CRIB (Ours)** | **37,450** | **148.03** |
| DLinear | 1,200 | 18.99 |
| PAttn | 15,640 | 55.13 |
| SegRNN | 8,056 | 30.17 |
| Transformer | 57,063 | 189.38 |
| iTransformer | 39,768 | 154.18 |
| PatchTST | 41,528 | 179.80 |
| TSMixer | 6,837 | 21.16 |
| WPMixer | 44,370 | 50.16 |
| CSDI | 239,649 | 1,269.72 |
| NeuralCDE | 37,767 | 41.40 |
| ImputeFormer | 264,193 | 1,488.22 |
| TimesNet | 863,895 | 1,427.44 |

## G.5 EXTRA ABLATION STUDY

We have done an extra ablation study on three cases of loss of CRIB to prove its effectiveness. The ablation analysis across four datasets confirms the necessity of each component, as removing the Consistency Regularization ($\mathcal{L}_{\text{Consis}}$), Compactness ($\mathcal{L}_{\text{Comp}}$), or Informativeness ($\mathcal{L}_{\text{Pred}}$) objectives consistently leads to performance degradation, validating their collective role in ensuring robust and accurate forecasting in MTSF-M tasks.

Table 10: Ablation study on ETTh1 dataset.

| ETTh1 | 0.2 | | 0.4 | | 0.6 | | 0.7 | |
|---|---|---|---|---|---|---|---|---|
| | MAE | MSE | MAE | MSE | MAE | MSE | MAE | MSE |
| w/o Consis | 0.237 | 0.206 | 0.274 | 0.270 | 0.338 | 0.413 | 0.402 | 0.584 |
| w/o Reg | 0.236 | 0.206 | 0.274 | 0.270 | 0.339 | 0.410 | 0.400 | 0.579 |
| w/o Pred | 0.432 | 0.559 | 0.505 | 0.698 | 0.655 | 1.066 | 0.796 | 1.486 |
| **Entire** | **0.220** | **0.171** | **0.251** | **0.249** | **0.267** | **0.296** | **0.288** | **0.361** |

Table 11: Ablation study on Elec dataset.

| Elec | 0.2 | | 0.4 | | 0.6 | | 0.7 | |
|---|---|---|---|---|---|---|---|---|
| | MAE | MSE | MAE | MSE | MAE | MSE | MAE | MSE |
| w/o Consis | 0.0201 | 0.0196 | 0.0267 | 0.0338 | 0.0346 | 0.0569 | 0.0415 | 0.0931 |
| w/o Reg | 0.0186 | 0.0175 | 0.0244 | 0.0286 | 0.0322 | 0.0544 | 0.0399 | 0.0980 |
| w/o Pred | 0.0881 | 0.5454 | 0.1243 | 0.9424 | 0.1818 | 1.8995 | 0.2283 | 2.9058 |
| **Entire** | **0.0150** | **0.0120** | **0.0230** | **0.0280** | **0.0300** | **0.0470** | **0.0380** | **0.0910** |

Table 12: Ablation study on Metr-LA dataset.

| Metr-LA | 0.2 | | 0.4 | | 0.6 | | 0.7 | |
|---|---|---|---|---|---|---|---|---|
| | MAE | MSE | MAE | MSE | MAE | MSE | MAE | MSE |
| w/o Consis | 0.271 | 0.332 | 0.257 | 0.275 | 0.271 | 0.306 | 0.314 | 0.366 |
| w/o Reg | 0.253 | 0.307 | 0.251 | 0.275 | 0.266 | 0.307 | 0.311 | 0.364 |
| w/o Pred | 0.442 | 0.418 | 0.624 | 0.529 | 0.985 | 1.177 | 1.276 | 1.959 |
| **Entire** | **0.248** | **0.271** | **0.249** | **0.272** | **0.265** | **0.305** | **0.309** | **0.356** |

Table 13: Ablation study on PEMS-BAY dataset.

| PEMS-BAY | 0.2 | | 0.4 | | 0.6 | | 0.7 | |
|---|---|---|---|---|---|---|---|---|
| | MAE | MSE | MAE | MSE | MAE | MSE | MAE | MSE |
| w/o Consis | 0.0959 | 0.0470 | 0.0882 | 0.0389 | 0.0976 | 0.0465 | 0.1296 | 0.0596 |
| w/o Reg | 0.0942 | 0.0453 | 0.0869 | 0.0373 | 0.0964 | 0.0456 | 0.1190 | 0.0591 |
| w/o Pred | 0.4006 | 0.2405 | 0.5618 | 0.3631 | 1.0832 | 1.2727 | 2.1480 | 4.8111 |
| **Entire** | **0.0830** | **0.0340** | **0.0850** | **0.0350** | **0.0930** | **0.0430** | **0.1100** | **0.0580** |

## G.6 EXTRA SENSITIVITY STUDY

We analyze hyperparameter sensitivity in Fig. 4 (b) and conduct additional sensitivity studies. Empirical results indicate that the optimal settings are consistent across different datasets and missing rates. We set the weights for $\mathcal{L}_{\text{Comp}}$, $\mathcal{L}_{\text{Pred}}$, and $\mathcal{L}_{\text{Consis}}$ to $10^{-6}$, 1, and 1 as default, respectively.

Table 14: Sensitivity analysis of $\mathcal{L}_{\text{Pred}}$ weight across all missing rates. (H1: ETTh1, Exch: Exchange, Ill: Illness)

| Weight | 0% Missing | | | 20% Missing | | | 40% Missing | | | 60% Missing | | | 70% Missing | | |
|---|---|---|---|---|---|---|---|---|---|---|---|---|---|---|---|
| | H1 | Exch | Ill | H1 | Exch | Ill | H1 | Exch | Ill | H1 | Exch | Ill | H1 | Exch | Ill |
| 0.1 | 0.201 | 0.0186 | 0.1495 | 0.243 | 0.0332 | 0.2466 | 0.281 | 0.0439 | 0.2741 | 0.348 | 0.0413 | 0.3631 | 0.412 | 0.0373 | 0.4796 |
| 0.5 | 0.198 | 0.0186 | 0.1474 | 0.238 | 0.0301 | 0.2341 | 0.273 | 0.0294 | 0.2828 | 0.340 | 0.0362 | 0.3535 | 0.406 | 0.0423 | 0.4895 |
| 1.0 | 0.198 | 0.0186 | 0.1457 | 0.237 | 0.0287 | 0.2306 | 0.274 | 0.0253 | 0.2573 | 0.337 | 0.0363 | 0.3703 | 0.403 | 0.0517 | 0.4914 |
| 2.0 | 0.199 | 0.0186 | 0.1500 | 0.236 | 0.0261 | 0.2383 | 0.273 | 0.0261 | 0.2660 | 0.339 | 0.0363 | 0.3870 | 0.400 | 0.0475 | 0.4955 |
| 5.0 | 0.200 | 0.0186 | 0.1533 | 0.237 | 0.0246 | 0.2337 | 0.273 | 0.0258 | 0.2709 | 0.338 | 0.0287 | 0.3509 | 0.401 | 0.0320 | 0.4497 |

Table 15: Sensitivity analysis of $\mathcal{L}_{\text{Comp}}$ weight across all missing rates. (H1: ETTh1, Exch: Exchange, Ill: Illness)

| Weight | 0% Missing | | | 20% Missing | | | 40% Missing | | | 60% Missing | | | 70% Missing | | |
|---|---|---|---|---|---|---|---|---|---|---|---|---|---|---|---|
| | H1 | Exch | Ill | H1 | Exch | Ill | H1 | Exch | Ill | H1 | Exch | Ill | H1 | Exch | Ill |
| 10 | 0.214 | 0.0192 | 0.1713 | 0.260 | 0.0509 | 0.2936 | 0.314 | 0.0806 | 0.3263 | 0.417 | 0.2131 | 0.4830 | 0.506 | 0.2851 | 0.6264 |
| 1 | 0.208 | 0.0188 | 0.1702 | 0.251 | 0.0471 | 0.2802 | 0.296 | 0.0835 | 0.2942 | 0.365 | 0.1393 | 0.4377 | 0.437 | 0.2293 | 0.5119 |
| $10^{-2}$ | 0.199 | 0.0186 | 0.1533 | 0.238 | 0.0300 | 0.2536 | 0.274 | 0.0319 | 0.2679 | 0.341 | 0.0435 | 0.3710 | 0.406 | 0.0499 | 0.4588 |
| $10^{-6}$ | 0.198 | 0.0186 | 0.1457 | 0.237 | 0.0287 | 0.2306 | 0.274 | 0.0253 | 0.2573 | 0.337 | 0.0363 | 0.3703 | 0.403 | 0.0517 | 0.4914 |
| $10^{-10}$ | 0.199 | 0.0186 | 0.1458 | 0.237 | 0.0276 | 0.2349 | 0.273 | 0.0260 | 0.2583 | 0.339 | 0.0417 | 0.3625 | 0.403 | 0.0408 | 0.4949 |

Table 16: Sensitivity analysis of $\mathcal{L}_{\text{Consis}}$ weight across all missing rates. (H1: ETTh1, Exch: Exchange, Ill: Illness)

| Weight | 0% Missing | | | 20% Missing | | | 40% Missing | | | 60% Missing | | | 70% Missing | | |
|---|---|---|---|---|---|---|---|---|---|---|---|---|---|---|---|
| | H1 | Exch | Ill | H1 | Exch | Ill | H1 | Exch | Ill | H1 | Exch | Ill | H1 | Exch | Ill |
| 0.1 | 0.201 | 0.0186 | 0.1556 | 0.237 | 0.0231 | 0.2289 | 0.274 | 0.0270 | 0.2527 | 0.340 | 0.0301 | 0.3519 | 0.400 | 0.0341 | 0.4587 |
| 0.5 | 0.199 | 0.0186 | 0.1497 | 0.236 | 0.0260 | 0.2348 | 0.274 | 0.0273 | 0.2582 | 0.339 | 0.0378 | 0.3647 | 0.401 | 0.0368 | 0.4919 |
| 1.0 | 0.198 | 0.0186 | 0.1457 | 0.237 | 0.0287 | 0.2306 | 0.274 | 0.0253 | 0.2573 | 0.337 | 0.0363 | 0.3703 | 0.403 | 0.0517 | 0.4914 |
| 2.0 | 0.199 | 0.0186 | 0.1477 | 0.239 | 0.0306 | 0.2344 | 0.275 | 0.0300 | 0.2801 | 0.340 | 0.0352 | 0.3414 | 0.404 | 0.0427 | 0.5009 |
| 5.0 | 0.200 | 0.0186 | 0.1497 | 0.240 | 0.0312 | 0.2402 | 0.278 | 0.0377 | 0.2808 | 0.344 | 0.0376 | 0.3524 | 0.410 | 0.0357 | 0.5203 |

