# OpenReview forum: "Revisiting Multivariate Time Series Forecasting with Missing Values"
_ICLR.cc/2026/Conference — Submitted to ICLR 2026_

### Official Review · Reviewer_nmeJ · 2025-10-16

**Soundness:** 2
**Presentation:** 3
**Contribution:** 2
**Rating:** 4
**Confidence:** 5

**Summary:**

This paper addresses the problem of multivariate time series forecasting with missing values and proposes CRIB, a direct prediction approach that integrates the Information Bottleneck theory, a unified-variate attention mechanism, and consistency regularization.

**Strengths:**

- S1: The paper conducts an empirical study that analyzes and reveals the limitations of existing “imputation-then-prediction” MTSF-M methods.
- S2: The paper conducts extensive experiments to demonstrate the effectiveness of CRIB, covering four datasets, multiple baselines, and various missing patterns.

**Weaknesses:**

- W1: The novelty of the paper is not clearly articulated, and the motivation behind each technical module is insufficiently explained. It appears that the method mainly combines existing components such as the Information Bottleneck, Unified-Variate Attention, and Consistency Regularization.
- W2：The paper uses an empirical study to analyze and argue that the imputation step in imputation-then-prediction methods is unreasonable. However, the details of the empirical study are not clearly explained.
- W3：The paper lacks comparisons with the latest MTSF-M baselines, such as GinAR.
GinAR: An End-To-End Multivariate Time Series Forecasting Model Suitable for Variable Missing, KDD 2024.
- W4：The paper contains some inconsistencies between the text and figures. For example, $L_{conc}$ in Figure 2(d) is not mentioned in the main text.

**Questions:**

- Q1: How were the distribution and correlation map in Figure 1 illustrated? What are the specific implementation details?
- Q2：Could you clarify the challenges of using direct prediction to address the MTSF-M problem, and explain what specific issues each module of CRIB tackles and what their respective technical novelties are?
- Q3：How does the paper demonstrate that consistency regularization with the IB theory can retain essential task-relevant information while filtering out irrelevant noise caused by missing values?
- Q4：How is the Imputed variant implemented in the experiment? Is there any information leakage between the training of the completion task and the prediction task?
- Q5：How are the three missing patterns, point, block, and column, simulated?

---

> ### Author Response · Authors · 2025-11-23
>
> We sincerely thank Reviewer nmeJ for the thoughtful and constructive reviews of our manuscript! Based on your questions and recommendations, we give point-by-point responses to your comments and describe the revisions we made to address them.
>
> ---
>
> > **W1:** The novelty of the paper is not clearly articulated, and the motivation behind each technical module is insufficiently explained. It appears that the method mainly combines existing components such as the Information Bottleneck, Unified-Variate Attention, and Consistency Regularization.
> >
> > **Q2:** Could you clarify the challenges of using direct prediction to address the MTSF-M problem, and explain what specific issues each module of CRIB tackles and what their respective technical novelties are?
>
>
> **Motivation and Contribution:** Our core contribution is not the IB theory itself, but a systematic re-evaluation of the "imputation-then-prediction" paradigm and the proposal of CRIB to resolve its fundamental limitations:
>
> - **Detrimental Imputation:** We demonstrate that imputation without ground truth is often harmful. As visualized in Figure 1, methods following the "imputation-then-prediction" paradigm fail to recover the true data distribution and instead reinforce biased patterns from partial observations.
>
> - **Performance Degradation:** We provide counter-intuitive evidence that imputation actively harms prediction accuracy. For instance, equipping the predictor TimeXer with the SOTA imputer TimesNet increases the MAE from 0.125 to 0.135.
>
> - **Limitations of Vanilla IB:** We further observe that a naive application of the Information Bottleneck is insufficient. Figure 6 (Appendix F.2) shows that a direct IB-based Transformer yields degenerate attention maps, biasing the model towards local linearity and neglecting global dependencies. CRIB can capture good attention map that shows it captures task-related information. Furthermore, results in Table 2 (Section 4.3) indicate that CRIB without the consistency loss exhibits significantly higher variance. CRIB resolves this by integrating IB with consistency regularization and Unified-Attention to effectively filter noise.
>
> - We also add the qualitative comparison between the **average** final prediction performance on missing rate of 20%, 40%, 60% and 70% of our method CRIB, BiTGraph, DLinear, TimesNet combined with DLinear, TimeXer, and TimesNet combined with TimeXer.
>
>   | PEMS-BAY |   CRIB    | BiTGraph | DLinear | TimesNet + DLinear | TimeXer | TimesNet + TimeXer |
>   | :------- | :-------: | :------: | :-----: | :----------------: | :-----: | :----------------: |
>   | MAE      | **0.093** |  0.413   |  0.156  |       0.148        | `0.125` |       0.135        |
>   | MSE      | **0.043** |  0.788   |  0.087  |       0.081        | `0.051` |       0.073        |
>
> - **The CRIB Solution:** To address these issues, CRIB adopts a "direct prediction" paradigm. By integrating IB with consistency regularization and Unified-Attention, CRIB effectively filters noise inherent in missing values and captures global information, bypassing the error propagation caused by erroneous imputation.
>   - **Patching Embedding:** Solves sparse input problem, extracts local features.
>
>   - **Unified-Variate Attention:** Solves correlation destruction problem, captures dependencies between variates and time steps simultaneously.
>   - **IB Guidance:** Solves noise problem, captures task-related information while filtering redundant information introduced by missing values.
>   - **Consistency Regularization:** Solves robustness problem, ensures model invariance to different missing patterns.

---

> > ### Author Response · Authors · 2025-11-23
> > **Rebuttal by Authors**
> >
> > > **W2:** The paper uses an empirical study to analyze and argue that the imputation step in imputation-then-prediction methods is unreasonable. However, the details of the empirical study are not clearly explained.
> > >
> > > **Q1:** How were the distribution and correlation map in Figure 1 illustrated? What are the specific implementation details?
> >
> >
> >
> > We provide the implementation details for the empirical study presented in Figure 1 to clarify our analysis.
> >
> > - **For the distribution visualization in Figure 1(a)**, we employ t-SNE to map the high-dimensional features into a lower-dimensional space. We compare the distributions of the original data, the observed partial data, and the data imputed by BiTGraph and TimesNet against the representations learned by CRIB.
> > - **For the correlation analysis in Figures 1 (b) and (c)**, we calculate Pearson correlation coefficient matrices to quantify inter-variate dependencies. We evaluate the correlation structures preserved by the ground truth and the observed data, comparing them with those generated by imputation baselines, the direct prediction baseline DLinear, and our proposed model.

---

> ### Author Response · Authors · 2025-11-23
> **Rebuttal by Authors**
>
> > **W3:** The paper lacks comparisons with the latest MTSF-M baselines, such as GinAR. GinAR: An End-To-End Multivariate Time Series Forecasting Model Suitable for Variable Missing, KDD 2024.
>
> To ensure a comprehensive and fair evaluation, we selected baselines based on their code availability and their applicability to general MTSF-M scenarios, which typically lack predefined graph structures (GinAR need this). Accordingly, we have expanded our experimental validation to include comparisons with **CSDI**, **ImputeFormer**, **NeuralCDE**, and **TimesNet** across a broader set of datasets. We exclude S4M solely due to reproducibility issues with their publicly available code.
>
> **Extensive experiments on these datasets confirm that CRIB consistently achieves state-of-the-art performance, exhibiting superior robustness across varying missing rates compared to both direct prediction and imputation-based baselines.**
>
> |DatasetsInformation|ETTh1|ETTh2|ETTm1|ETTm2|Electricity|PEMS-BAY|Metr-LA|BeijingAir|Weather|Exchange|
> |--|:--:|:--:|:--:|:--:|:--:|:--:|:--:|:--:|:--:|:--:|
> |TimeSteps|17420|17420|69680|69680|26304|52116|34272|36000|52696|7588|
> |Variates|7|7|7|7|321|325|207|7|21|8|
>
> |ETTh1|0.0|0.0|0.2|0.2|0.4|0.4|0.6|0.6|0.7|0.7|
> |--|--|--|--|--|--|--|--|--|--|--|
> ||MAE|MSE|MAE|MSE|MAE|MSE|MAE|MSE|MAE|MSE|
> |CRIB|**0.195**|**0.163**|**0.238**|**0.211**|**0.274**|**0.273**|**0.342**|**0.427**|**0.411**|**0.624**|
> |SegRNN|0.209|0.167|0.247|0.215|0.297|0.307|0.380|0.506|0.455|0.737|
> |WPMixer|0.201|0.165|0.243|0.217|0.290|0.302|0.376|0.504|0.458|0.755|
> |iTransformer|0.211|0.172|0.250|0.219|0.338|0.316|0.385|0.494|0.453|0.731|
> |PatchTST|0.206|0.166|0.245|0.219|0.306|0.292|0.367|0.473|0.431|0.693|
> |DLinear|0.204|0.170|0.270|0.245|0.331|0.361|0.443|0.629|0.544|0.927|
> |PAttn|0.199|0.164|0.249|0.227|0.329|0.309|0.375|0.487|0.437|0.718|
> |CSDI|0.217|0.203|0.305|0.325|0.404|0.503|0.567|0.880|0.717|1.311|
> |NeuralCDE|0.291|0.236|0.340|0.315|0.408|0.460|0.522|0.761|0.629|1.101|
> |ImputeFormer|0.213|0.182|0.260|0.265|0.312|0.369|0.399|0.630|0.488|0.938|
> |TimesNet|0.225|0.178|0.256|0.234|0.297|0.302|0.373|0.457|0.443|0.627|
>
>
> |ETTh2|0.0|0.0|0.2|0.2|0.4|0.4|0.6|0.6|0.7|0.7|
> |--|--|--|--|--|--|--|--|--|--|--|
> ||MAE|MSE|MAE|MSE|MAE|MSE|MAE|MSE|MAE|MSE|
> |CRIB|**0.094**|**0.023**|**0.107**|**0.028**|**0.131**|**0.039**|**0.154**|**0.055**|**0.172**|**0.069**|
> |SegRNN|0.096|0.024|0.115|0.031|0.148|0.047|0.183|0.072|0.205|0.093|
> |WPMixer|0.095|0.024|0.116|0.030|0.147|0.045|0.202|0.086|0.218|0.104|
> |iTransformer|0.098|0.025|0.115|0.033|0.298|0.180|0.270|0.170|0.218|0.111|
> |PatchTST|0.096|0.024|0.109|0.030|0.215|0.125|0.216|0.121|0.186|0.083|
> |DLinear|0.098|0.025|0.142|0.040|0.181|0.065|0.253|0.135|0.325|0.228|
> |PAttn|0.096|0.025|0.116|0.033|0.289|0.173|0.243|0.133|0.196|0.085|
> |CSDI|0.113|0.033|0.255|0.192|0.404|0.399|0.618|0.840|0.798|1.303|
> |NeuralCDE|0.197|0.074|0.244|0.105|0.276|0.134|0.347|0.210|0.413|0.303|
> |ImputeFormer|0.101|0.026|0.122|0.034|0.140|0.049|0.173|0.061|0.174|0.071|
> |TimesNet|0.097|0.025|0.121|0.033|0.153|0.048|0.209|0.092|0.246|0.132|
>
>
> |Weather|0.0|0.0|0.2|0.2|0.4|0.4|0.6|0.6|0.7|0.7|
> |--|--|--|--|--|--|--|--|--|--|--|
> ||MAE|MSE|MAE|MSE|MAE|MSE|MAE|MSE|MAE|MSE|
> |CRIB|**0.028**|**0.016**|**0.038**|**0.025**|**0.050**|**0.033**|**0.062**|**0.057**|**0.070**|**0.075**|
> |SegRNN|0.031|0.021|0.042|0.028|0.050|0.037|0.066|0.061|0.076|0.081|
> |WPMixer|0.030|0.019|0.050|0.025|0.074|0.036|0.102|0.062|0.121|0.084|
> |iTransformer|0.030|0.018|0.044|0.031|0.152|0.167|0.175|0.223|0.101|0.138|
> |PatchTST|0.031|0.019|0.048|0.027|0.150|0.156|0.169|0.169|0.118|0.128|
> |DLinear|0.034|0.023|0.099|0.045|0.135|0.078|0.197|0.178|0.254|0.318|
> |PAttn|0.031|0.020|0.046|0.032|0.157|0.140|0.168|0.163|0.100|0.120|
> |CSDI|0.051|0.042|0.173|0.220|0.300|0.500|0.466|1.099|0.586|1.707|
> |NeuralCDE|0.051|0.029|0.073|0.036|0.095|0.060|0.150|0.150|0.204|0.284|
> |ImputeFormer|0.035|0.020|0.041|0.028|0.051|0.033|0.066|0.064|0.076|0.083|
> |TimesNet|0.028|0.016|0.062|0.033|0.088|0.047|0.128|0.089|0.166|0.154|

---

> ### Author Response · Authors · 2025-11-23
> **Rebuttal by Authors**
>
> |Exchange|0.0|0.0|0.2|0.2|0.4|0.4|0.6|0.6|0.7|0.7|
> |--|--|--|--|--|--|--|--|--|--|--|
> ||MAE|MSE|MAE|MSE|MAE|MSE|MAE|MSE|MAE|MSE|
> |CRIB|**0.0184**|**0.0009**|**0.0217**|**0.0016**|**0.0253**|**0.0015**|**0.0363**|**0.0031**|**0.0517**|**0.0058**|
> |SegRNN|0.0185|0.0010|0.0292|0.0031|0.0629|0.0129|0.1165|0.0363|0.1418|0.0490|
> |WPMixer|0.0187|0.0010|0.0324|0.0024|0.0783|0.0127|0.1364|0.0405|0.1836|0.0697|
> |iTransformer|0.0190|0.0010|0.0288|0.0024|0.1899|0.0770|0.2439|0.1060|0.2865|0.1413|
> |PatchTST|0.0186|0.0010|0.0273|0.0019|0.1416|0.0884|0.1978|0.1129|0.1321|0.0581|
> |DLinear|0.0205|0.0011|0.0795|0.0126|0.1469|0.0429|0.2599|0.1394|0.3778|0.3011|
> |PAttn|0.0187|0.0010|0.0227|0.0016|0.1408|0.1047|0.1283|0.0831|0.0718|0.0169|
> |CSDI|0.0264|0.0017|0.1992|0.2089|0.4358|0.5718|0.7911|1.3714|1.0766|2.2850|
> |NeuralCDE|0.3223|0.1724|0.2969|0.1592|0.3326|0.2146|0.4334|0.3814|0.5412|0.5912|
> |ImputeFormer|0.0344|0.0031|0.0364|0.0029|0.0431|0.0034|0.0608|0.0071|0.0846|0.0128|
> |TimesNet|0.0195|0.0011|0.0458|0.0041|0.0786|0.0130|0.1357|0.0404|0.1991|0.0881|
>
>
> |ETTm1|0.0|0.0|0.2|0.2|0.4|0.4|0.6|0.6|0.7|0.7|
> |--|--|--|--|--|--|--|--|--|--|--|
> ||MAE|MSE|MAE|MSE|MAE|MSE|MAE|MSE|MAE|MSE|
> |CRIB|**0.2000**|**0.2008**|**0.2368**|**0.2586**|**0.2734**|**0.3382**|**0.3414**|**0.5200**|**0.4013**|**0.6956**|
> |SegRNN|0.2265|0.2302|0.2616|0.3068|0.3005|0.4007|0.3678|0.5960|0.4266|0.8006|
> |WPMixer|0.2405|0.2741|0.2900|0.3593|0.3308|0.4645|0.4171|0.7328|0.4863|0.9928|
> |iTransformer|0.2309|0.2514|0.2909|0.3751|0.3593|0.4851|0.4365|0.7580|0.5034|1.0663|
> |PatchTST|0.2379|0.2648|0.2779|0.3486|0.3456|0.4671|0.4091|0.6769|0.4743|0.9325|
> |DLinear|0.2772|0.3504|0.3711|0.5499|0.4418|0.7412|0.5551|1.1067|0.6508|1.4721|
> |PAttn|0.2423|0.2782|0.2971|0.3864|0.3592|0.4801|0.4191|0.7022|0.4816|0.9837|
> |CSDI|0.4759|0.8854|0.5198|0.9345|0.5787|1.0499|0.7038|1.3801|0.8287|1.7733|
> |NeuralCDE|0.3331|0.4218|0.3809|0.5177|0.4391|0.6643|0.5457|0.9573|0.6278|1.2710|
> |ImputeFormer|0.2999|0.2682|0.3271|0.3282|0.3571|0.4749|0.4315|0.6978|0.4845|0.9035|
> |TimesNet|0.2100|0.2018|0.2584|0.2992|0.3040|0.4093|0.3993|0.6525|0.4828|0.8988|
>
>
> |ETTm2|0.0|0.0|0.2|0.2|0.4|0.4|0.6|0.6|0.7|0.7|
> |--|--|--|--|--|--|--|--|--|--|--|
> ||MAE|MSE|MAE|MSE|MAE|MSE|MAE|MSE|MAE|MSE|
> |CRIB|**0.0746**|**0.0152**|**0.0885**|**0.0203**|**0.1037**|**0.0258**|**0.1266**|**0.0384**|**0.1467**|**0.0520**|
> |SegRNN|0.0802|0.0175|0.0956|0.0231|0.1159|0.0305|0.1432|0.0477|0.1650|0.0619|
> |WPMixer|0.0821|0.0182|0.1059|0.0266|0.1355|0.0394|0.1790|0.0685|0.1899|0.0808|
> |iTransformer|0.0781|0.0161|0.1020|0.0284|0.2673|0.1688|0.2651|0.1671|0.1905|0.0861|
> |PatchTST|0.0824|0.0182|0.1005|0.0251|0.2041|0.1210|0.2425|0.1423|0.1803|0.0779|
> |DLinear|0.0918|0.0217|0.1405|0.0391|0.1743|0.0601|0.2412|0.1215|0.3104|0.2056|
> |PAttn|0.0828|0.0187|0.1024|0.0278|0.2441|0.1490|0.2398|0.1448|0.1728|0.0717|
> |CSDI|0.1313|0.0410|0.2687|0.1976|0.4131|0.4003|0.6221|0.8293|0.8024|1.2968|
> |NeuralCDE|0.1327|0.0363|0.1760|0.0591|0.1984|0.0765|0.2479|0.1194|0.3130|0.1857|
> |ImputeFormer|0.0813|0.0160|0.0888|0.0222|0.1088|0.0267|0.1360|0.0424|0.1602|0.0569|
> |TimesNet|0.0782|0.0165|0.1087|0.0260|0.1417|0.0406|0.1986|0.0811|0.2317|0.1142|
>
> |BejingAir|0.0|0.0|0.2|0.2|0.4|0.4|0.6|0.6|0.7|0.7|
> |--|--|--|--|--|--|--|--|--|--|--|
> ||MAE|MSE|MAE|MSE|MAE|MSE|MAE|MSE|MAE|MSE|
> |CRIB|**0.2526**|**0.2929**|**0.2758**|**0.3317**|**0.3106**|**0.4120**|**0.3728**|**0.5862**|**0.4337**|0.8010|
> |SegRNN|0.2552|0.3026|0.2849|0.3467|0.3188|0.4285|0.3860|0.5939|0.4412|**0.7595**|
> |WPMixer|0.2571|0.3085|0.2896|0.3651|0.3312|0.4601|0.4011|0.6503|0.4632|0.8730|
> |iTransformer|0.2606|0.3156|0.2923|0.3655|0.3545|0.4820|0.4124|0.6573|0.4691|0.8874|
> |PatchTST|0.2609|0.3008|0.2866|0.3421|0.3278|0.4305|0.3976|0.6458|0.4613|0.8540|
> |DLinear|0.2707|0.3263|0.3204|0.3853|0.3620|0.4760|0.4362|0.6799|0.5022|0.9358|
> |PAttn|0.2601|0.3102|0.2949|0.3603|0.3480|0.4689|0.4142|0.6624|0.4673|0.8745|
> |CSDI|0.3431|0.5141|0.4035|0.6010|0.4714|0.7584|0.5930|1.1378|0.7057|1.5761|
> |NeuralCDE|0.3167|0.3779|0.3478|0.4196|0.3950|0.5474|0.4750|0.7803|0.5456|1.0151|
> |ImputeFormer|0.2533|0.3189|0.2770|0.3496|0.3297|0.4645|0.4006|0.6110|0.4549|0.8327|
> |TimesNet|0.2583|0.2939|0.2922|0.3488|0.3322|0.4422|0.4017|0.6275|0.4789|0.9234|

---

> ### Author Response · Authors · 2025-11-23
> **Rebuttal by Authors**
>
> > **W4:** The paper contains some inconsistencies between the text and figures. For example, in Figure 2(d) $\mathcal{L}_{con}$ is not mentioned in the main text.
>
>
> We appreciate the detailed observation regarding the notation. We clarify that the term $\mathcal{L}\_\text{con}$ appearing in Figure 2(d) corresponds directly to the Consistency Regularization loss defined as $\mathcal{L}\_\text{Consis}$ in Section 3.5. To ensure terminological consistency throughout the paper, we will correct the label in Figure 2(d) to $\mathcal{L}\_\text{Consis}$ in the revised manuscript.
>
>
>
>
> > **Q3:** How does the paper demonstrate that consistency regularization with the IB theory can retain essential task-relevant information while filtering out irrelevant noise caused by missing values?
>
>
> We validate the effectiveness of integrating Consistency Regularization with the Information Bottleneck through three complementary lines of evidence.
>
> - Qualitatively, Figure 1 demonstrates that the embedding distribution learned by CRIB closely aligns with the ground truth, accurately preserving essential inter-variate correlations while suppressing distribution shifts caused by missing values.
>
> - Structurally, the attention visualizations in Figure 6 (Appendix F.2) confirm that our model successfully captures critical intra- and inter-variate dependencies, avoiding the degenerate local bias observed in standard baselines.
>
> - Quantitatively, the ablation study in Figure 4(b) indicates that removing the Information Bottleneck leads to a significant performance drop, confirming its necessity for noise filtering, while removing Consistency Regularization degrades robustness (Table 2 in Section 4.3), particularly under high missing rates. We have further conducted comprehensive ablation experiments on the individual loss components to rigorously verify the distinct contribution of each module to the final performance.
>
>   **The ablation analysis across four datasets confirms the necessity of each component, as removing the Consistency, Regularization, or Prediction objectives consistently leads to performance degradation, validating their collective role in ensuring robust and accurate forecasting.**
>
> |ETTh1|0.2|0.2|0.4|0.4|0.6|0.6|0.7|0.7|
> |--|:--:|:--:|:--:|:--:|:--:|:--:|:--:|:--:|
> ||MAE|MSE|MAE|MSE|MAE|MSE|MAE|MSE|
> |w/oConsis|0.237|0.206|0.274|0.270|0.338|0.413|0.402|0.584|
> |w/oReg|0.236|0.206|0.274|0.270|0.339|0.410|0.400|0.579|
> |w/oPred|0.432|0.559|0.505|0.698|0.655|1.066|0.796|1.486|
> |Entire|0.220|0.171|0.251|0.249|0.267|0.296|0.288|0.361|
>
> |Elec|0.2|0.2|0.4|0.4|0.6|0.6|0.7|0.7|
> |--|:--:|:--:|:--:|:--:|:--:|:--:|:--:|:--:|
> ||MAE|MSE|MAE|MSE|MAE|MSE|MAE|MSE|
> |w/oConsis|0.0201|0.0196|0.0267|0.0338|0.0346|0.0569|0.0415|0.0931|
> |w/oReg|0.0186|0.0175|0.0244|0.0286|0.0322|0.0544|0.0399|0.0980|
> |w/oPred|0.0881|0.5454|0.1243|0.9424|0.1818|1.8995|0.2283|2.9058|
> |Entire|0.0150|0.0120|0.0230|0.0280|0.0300|0.0470|0.0380|0.0910|
>
> |Metr|0.2|0.2|0.4|0.4|0.6|0.6|0.7|0.7|
> |--|:--:|:--:|:--:|:--:|:--:|:--:|:--:|:--:|
> ||MAE|MSE|MAE|MSE|MAE|MSE|MAE|MSE|
> |w/oConsis|0.271|0.332|0.257|0.275|0.271|0.306|0.314|0.366|
> |w/oReg|0.253|0.307|0.251|0.275|0.266|0.307|0.311|0.364|
> |w/oPred|0.442|0.418|0.624|0.529|0.985|1.177|1.276|1.959|
> |Entire|0.248|0.271|0.249|0.272|0.265|0.305|0.309|0.356|
>
> |PEMS|0.2|0.2|0.4|0.4|0.6|0.6|0.7|0.7|
> |--|:--:|:--:|:--:|:--:|:--:|:--:|:--:|:--:|
> ||MAE|MSE|MAE|MSE|MAE|MSE|MAE|MSE|
> |w/oConsis|0.0959|0.0470|0.0882|0.0389|0.0976|0.0465|0.1296|0.0596|
> |w/oReg|0.0942|0.0453|0.0869|0.0373|0.0964|0.0456|0.1190|0.0591|
> |w/oPred|0.4006|0.2405|0.5618|0.3631|1.0832|1.2727|2.1480|4.8111|
> |Entire|0.0830|0.0340|0.0850|0.0350|0.0930|0.0430|0.1100|0.0580|

---

> > ### Author Response · Authors · 2025-11-23
> > **Rebuttal by Authors**
> >
> > > **Q4:** How is the Imputed variant implemented in the experiment? Is there any information leakage between the training of the completion task and the prediction task?
> >
> >
> >
> > No leakage. As stated in Section 4.1, TimesNet is trained only on **train set** with 10% missing rate (simulating limited training data) and then used for imputation on the **test set** with different missing rates (0%-70%). This is strictly separated.
> >
> >
> >
> >
> > > **Q5:** How are the three missing patterns, point, block, and column, simulated?
> >
> >
> > - **Point:** Randomly masking independent points.
> > - **Block:** Masking continuous time segments.
> > - **Column:** Masking the time series of entire variates.

---

> > > ### Comment · Reviewer_nmeJ · 2025-11-24
> > >
> > > Thank you for your responses. I still have some questions:
> > >
> > > 1. Regarding the response to W2/Q1, for Figure 1(a), which layer's representation did you use to plot the t-SNE? Was it the representation Z mentioned in the paper? For Figure 1(b) and (c), what did you use to calculate the variable correlation? Was it the original data or the representations? Also, what were the specific settings you used for plotting? I think a more detailed answer or revision of the manuscript is needed.
> > >
> > > 2. Regarding the response to W3, I was surprised to find that the experimental results of CSDI and ImputeFormer are exactly the same on each dataset. Could you please explain why?
> > >
> > > 3. Regarding the response to Q3, what are the specific settings for your w/o Reg and w/o Pred experiments?
> > >
> > > 4. Regarding the response to Q4, in Table 1, does "original" refer to evaluating the prediction performance of the model using historical sequences with missing values ​​as input, and "imputed" refer to pre-training TimesNet as an imputation model on the training set, then using it to impute the historical sequences in the test set before using the prediction model to predict future sequences? Is my understanding correct?
> > >
> > > 5. Furthermore, why didn't you test on datasets with naturally occurring missing data, such as AQI? This would better reflect real-world situations compared to your simulated missing data.

---

> > > > ### Author Response · Authors · 2025-11-24
> > > > **Response to nmeJ's questions**
> > > >
> > > > Thanks for your time and these detailed questions.
> > > >
> > > > 1. Response to Figure 1 details (t-SNE and Correlation):
> > > >
> > > >    - **For Figure 1(a) (t-SNE):** We visualized the **embeddings from the final layer** of all models. For CRIB, this corresponds strictly to the representation $Z$. For other baselines, we extracted the latent representations immediately preceding the final projection layer.
> > > >
> > > >    - **For Figure 1(b) and (c) (Correlation):** We calculated the variable correlations using the **final prediction results ($\hat{Y}$)**, not the intermediate representations.
> > > >
> > > >    - Settings: The plotting settings are consistent with the main comparison experiments (PEMS-BAY dataset with 40% missing rate).
> > > >
> > > >    We are currently revising the manuscript to include these specific details. Since we are adding a significant number of new experiments (including the new baselines requested), we kindly ask for a little more time to finalize the updated PDF.
> > > >
> > > >
> > > > 2. Response to CSDI and ImputeFormer results:
> > > >
> > > >    We sincerely appreciate your keen eye in identifying this anomaly. We implemented CSDI and ImputeFormer by integrating the official code from the **PyPOTS** library into our training pipeline to ensure a consistent experimental environment. Upon investigating your concern, **we discovered a bug in the underlying validation code within the PyPOTS library implementation we used.**
> > > >
> > > >    We have fixed this issue and are currently re-running all experiments for both CSDI and ImputeFormer. We will update the corrected results as soon as they are finished.
> > > >
> > > >
> > > > 3. Response to Ablation Settings (w/o Reg and w/o Pred):
> > > >
> > > >    For the ablation studies, we implemented these variants by directly excluding the specific loss terms from the final objective function (Eq. 12):
> > > >
> > > >    - **w/o Reg (Compactness):** We set the weight of $\mathcal{L}_\text{Consis}$ as 0 to remove this loss  from the calculation.
> > > >
> > > >    - **w/o Pred (Informativeness):** We set the weight of $\mathcal{L}_\text{Pred}$ as 0  to remove this loss  from the guidance.
> > > >
> > > >
> > > > 4. Response to Table 1 ("Original" vs "Imputed"):
> > > >
> > > >    Your understanding is largely correct. To clarify the exact pipeline:
> > > >
> > > >    - **Data Setup:** We possess the complete ground truth for all datasets; the missingness is artificially generated (masked) for simulation purposes.
> > > >
> > > >    - **'Original' Setting:** We input the historical sequence with missing values directly into the model. The model is trained to predict the **complete ground truth** of the future sequence.
> > > >
> > > >    - **'Imputed' Setting:** This is a two-stage process. First, we pre-train a **TimesNet** model on the training set with a 10% missing rate to act as an imputer. Second, we use this frozen, pre-trained TimesNet to fill the missing values in the training (20%-70% missing), validation, and test sets. Finally, the prediction model is trained on this *imputed* data to predict the **complete ground truth** of the future sequence.
> > > >
> > > >
> > > > 5. Response to Natural Missing Data (AQI):
> > > >
> > > >    Thank you for the suggestion regarding the AQI dataset. Our current dataset selection follows the standard benchmarks provided by the **Time-Series-Library (thuml/Time-Series-Library)** to ensure fair comparison with existing literature. However, we agree that datasets with naturally occurring missing data like AQI better reflect real-world challenges.
> > > >
> > > >    If time permits, we intend to conduct experiments on the AQI dataset. However, we have a question regarding the experimental setup: Should we still apply artificial masking to the observed data? We are concerned that doing so might not differ significantly from our current experiments on standard datasets.
> > > >
> > > >    Alternatively, would you prefer us to conduct a single set of experiments where we perform predictions and calculate the loss directly on the available observed data in the AQI dataset, without introducing additional artificial missingness?

---

> > > > > ### Comment · Reviewer_nmeJ · 2025-11-25
> > > > >
> > > > > Regarding the third question: Based on your description of **w/o Reg**, what is the difference between this and **w/o consistency**? I feel like the settings for both are the same.
> > > > >
> > > > > Regarding the fifth question, since you claim that the impute-then-predict two-stage strategy is suboptimal, then comparing **direct prediction on datasets containing naturally missing data** with **predicting after imputing the original missing data** would be more convincing than experiments that only simulate missing data on complete datasets.

---

> > > > > > ### Author Response · Authors · 2025-11-25
> > > > > > **Response to Reviewer nmeJ**
> > > > > >
> > > > > > Thanks for your time and quick responses.
> > > > > >
> > > > > > **Question 1**
> > > > > >
> > > > > > We apologize for the confusion caused by the abbreviation "Reg" in our previous response. In our paper, we employ three distinct loss terms:
> > > > > >
> > > > > > - **$\mathcal{L}_\text{Comp}$ (Compactness):** The Information Bottleneck regularizer.
> > > > > >
> > > > > > - **$\mathcal{L}_\text{Pred}$ (Informativeness):** The prediction loss.
> > > > > >
> > > > > > - **$\mathcal{L}_\text{Consis}$ (Consistency)**: The consistency regularization between views.
> > > > > >
> > > > > > To clarify the difference between the two ablation settings you mentioned:
> > > > > >
> > > > > > - **w/o Reg:** This refers to removing the **Compactness Loss ($\mathcal{L}_{\text{Comp}}$)**. In the context of the Information Bottleneck principle, this term acts as a regularizer that forces the representation $Z$ to be compressed, thereby filtering out irrelevant information and noise from the input $X^\text{o}$.
> > > > > > - **w/o Consistency:** This refers to removing the **Consistency Loss ($\mathcal{L}_{\text{Consis}}$)**. This term enforces the model to learn invariant representations by minimizing the divergence between the original observed data and the augmented view.
> > > > > >
> > > > > >
> > > > > >
> > > > > > **Question 2**
> > > > > >
> > > > > > We fully agree with your insight. Comparing "direct prediction" against "imputation-then-prediction" on datasets with **naturally occurring missing data** (where the imputation model lacks ground truth supervision) would indeed provide the most convincing evidence for our claim regarding the suboptimality of the two-stage strategy.
> > > > > >
> > > > > > We are eager to conduct this experiment to strengthen our manuscript. However, after checking standard benchmarks (including the *Time-Series-Library* and *PyPOTS*), we found multiple versions of air quality datasets (e.g., PM2.5, Beijing Air Quality, etc.) and are unsure which specific "AQI" dataset you are referring to. **Could you kindly provide a reference or a link to the specific AQI dataset you have in mind?**

---

> > > > > > > ### Comment · Reviewer_nmeJ · 2025-11-26
> > > > > > >
> > > > > > > You can use the PM2.5 dataset from pypots, or the PM2.5 dataset from CSDI[1].
> > > > > > >
> > > > > > > [1] CSDI: Conditional Score-based Diffusion Models for Probabilistic Time Series Imputation

---

> > > > > > > > ### Author Response · Authors · 2025-11-26
> > > > > > > > **Response to Reviewer nmeJ**
> > > > > > > >
> > > > > > > > Thanks for your help and time.
> > > > > > > >
> > > > > > > > The results of ImputeFormer and CSDI are updated. We will update the PDF and do the experiment on PM2.5 soon.

---

> > > > > > > > ### Author Response · Authors · 2025-11-28
> > > > > > > > **Response to the experiments on naturally missing data (AQI)**
> > > > > > > >
> > > > > > > > We have conducted the experiments on the **AQI dataset** with naturally occurring missing data as suggested. To fairly compare 'direct prediction' against the 'imputation-then-prediction' strategy, we designed the experiment as follows:
> > > > > > > >
> > > > > > > > 1. **Imputation Model Setup:** We first pre-trained a **TimesNet** model on the AQI training set. To enable learning for imputation, we applied a **10% point missing mask** to the observed values during training. We selected a 10% masking rate because our statistical analysis showed that the natural missing rate of the AQI dataset is approximately 10%.
> > > > > > > > 2. **Two-Stage Process (AQI_IMP):** We used this pre-trained TimesNet to impute the naturally occurring **NaN values** across the entire AQI dataset. We then trained and evaluated the downstream forecasting models on this fully imputed dataset.
> > > > > > > > 3. **Direct Prediction (AQI_ORI):** For comparison, we trained and evaluated the models directly on the original AQI dataset containing natural missing values.
> > > > > > > > 4. **Evaluation Metric:** To ensure a valid comparison, the MAE and MSE metrics were calculated **only on the observed data points** (excluding original NaNs from the loss calculation via masking), as the ground truth for the naturally missing parts is unknown.
> > > > > > > >
> > > > > > > > **Results:** The experimental results are presented in the table below. We observed that for multiple forecasting models, **using TimesNet to impute the missing values actually degraded the prediction performance** compared to direct prediction. This negative impact empirically corroborates our paper's central claim: in the absence of ground truth supervision, the 'imputation-then-prediction' strategy can introduce noise and corrupt the data distribution, making it suboptimal compared to direct prediction methods like CRIB.
> > > > > > > >
> > > > > > > > | AQI_ORI  |   CRIB    | SegRNN | WPMixer | iTransformer | PatchTST | DLinear | PAttn | CSDI  | NeuralCDE | ImputeFormer | TimesNet |
> > > > > > > > | -------- | :-------: | :----: | :-----: | :----------: | :------: | :-----: | :---: | :---: | :-------: | :----------: | :------: |
> > > > > > > > | Test_MAE | **0.555** | 0.604  |  0.624  |    0.608     |  0.627   |  0.598  | 0.621 | 0.858 |   0.798   |    0.795     |  0.648   |
> > > > > > > > | Test_MSE | **0.663** | 0.804  |  0.843  |    0.818     |  0.844   |  0.741  | 0.843 | 1.448 |   1.438   |    1.313     |  0.925   |
> > > > > > > >
> > > > > > > > | AQI_IMP |   CRIB    | SegRNN | WPMixer | iTransformer | PatchTST | DLinear | PAttn | CSDI  | NeuralCDE | ImputeFormer | TimesNet |
> > > > > > > > | ----------- | :-------: | :----: | :-----: | :----------: | :------: | :-----: | :---: | :---: | :-------: | :----------: | :------: |
> > > > > > > > | Test_MAE    | **0.616** | 0.650  |  0.665  |    0.668     |  0.666   |  0.653  | 0.663 | 0.941 |   0.946   |    0.857     |  0.733   |
> > > > > > > > | Test_MSE    | **0.844** | 0.966  |  0.986  |    1.012     |  0.993   |  0.893  | 0.995 | 1.775 |   1.928   |    1.543     |  1.206   |

---

### Official Review · Reviewer_2Cic · 2025-10-26

**Soundness:** 2
**Presentation:** 2
**Contribution:** 2
**Rating:** 4
**Confidence:** 5

**Summary:**

To address the challenges posed by missing data, the authors introduce the CRIB. CRIB integrates a unified variable attention mechanism with a consistency regularization scheme. Comprehensive experiments conducted on four real-world datasets demonstrate the effectiveness of CRIB, which achieves accurate predictions even under high missing rates.

**Strengths:**

1. The article is well-written.
2. Prediction with missing values ​​is a significant issue.
3. The existing experiments demonstrate the model's excellent performance.

**Weaknesses:**

1. The motivation has already been explored. To my knowledge, the limitations of the "impute-then-predict" paradigm have been repeatedly discussed [1-3]. Additionally, CSDI also addresses the lack of authenticity in missing values during training and proposes a self-supervised learning strategy. Therefore, the authors should present a more compelling motivation. Furthermore, some imputation works that do not explicitly emphasize their applicability to prediction tasks (e.g., TimeNet, TimeMixer++) and some prediction works capable of directly handling missing values should be considered as competitors in this paper and discussed and compared in detail. Otherwise, this approach may lead to an overestimation of the paper's contributions.

2. The "impute-then-predict" approach is not the mainstream framework adhered to by existing methods. Some datasets naturally contain missing values, and many prediction works do not perform an additional imputation step—they directly make predictions. The "impute-then-predict" paradigm generally involves simulating missing data through random masking, which is more akin to a self-supervised learning approach.

3. The paper selects a relatively weak-performing model (DLinear) (which only consists of a few fully connected layers) and a heterogeneous model (TimesNet) to construct the "impute-then-predict" paradigm and uses them as the primary targets of criticism. The rationale for this motivation is insufficient. This experimental setup appears more like a "fragile" control group constructed to highlight the advantages of the proposed method, rather than a fair evaluation of the state-of-the-art or most commonly used paradigms in the real world. A more convincing analysis should directly compare with specialized, powerful imputation models (e.g., CSDI, ImputeFormer) or advanced prediction models capable of directly handling missing values (e.g., appropriate variants of D2STGNN, TimesNet), and delve into the potential limitations of these methods.

4. Furthermore, the authors' first contribution claims: "We perform a systematic empirical analysis of the dominant imputation-then-prediction paradigm for MTSF-M. We reveal that, guided only by a prediction objective, imputation modules can corrupt the observed data distribution and degrade prediction performance" seems exaggerated. The authors use a toy example to support their research, which is not systematic or concrete. Reliable findings should be based on either theoretical foundations or extensive experiments.

5. What challenges would other models face if performing direct prediction? Regarding these challenges, what specific strategies has your model implemented?

6. Although the authors introduce the information bottleneck theory, what is the utility of the lower bound proven in Section 3.4.2? In my view, if it's just an isolated value, it's difficult to grasp its meaning. For instance, the authors should introduce the lower/upper bounds of other methods for comparison to illustrate the significance of this bound.

7. Several imputation models are missing, such as PriSTI, CSDI, and ImputeFormer. Additionally, some spatiotemporal forecasting models are missing, e.g., D2STGNN,STID,PatchSTG. Some time series models are also missing, such as DUET, CycleNet, and TimesNet. Some commonly used time series data sets, such as Weather, Traffic, etc., need to be considered.

8. Furthermore, I recommend denormalizing all prediction results; this is standard practice for PeMS and Metr-LA datasets.

9. I suggest adding MAPE as a third metric.

10. The work appears to be a combination of mature components. The authors' temporal analysis model is a standard Transformer, and the theory is derived from the well-established mutual information/variational inference theory. Therefore, it seems more like an engineering effort of combination rather than an original contribution.

11. Did the authors evaluate the performance of an ablated variant without the Consistency Regularization term? This is necessary to demonstrate that the performance gain indeed stems from this module.

12. The authors' loss function involves three hyperparameters, which could clearly lead to optimization difficulties. Therefore, a detailed sensitivity analysis is required. Figure 4 alone is insufficient.

13. The code cannot be run directly. It would be better if you can add more detailed instructions for using the code.

Reference:

[1] Tashiro, Yusuke, et al. "Csdi: Conditional score-based diffusion models for probabilistic time series imputation." Advances in neural information processing systems 34 (2021): 24804-24816.

[2] Peng, J., Yang, M., Zhang, Q., & Li, X. (2025). S4M: S4 for multivariate time series forecasting with Missing values. arXiv preprint arXiv:2503.00900.

[3] Chen, X., Li, X., Liu, B., & Li, Z. (2023). Biased temporal convolution graph network for time series forecasting with missing values. In The Twelfth International Conference on Learning Representations.

**Questions:**

W1-W13.

---

> ### Author Response · Authors · 2025-11-23
>
> We sincerely thank Reviewer 2Cic for the thoughtful and constructive reviews of our manuscript! Based on your questions and recommendations, we give point-by-point responses to your comments and describe the revisions we made to address them.
>
> ---
>
> > **W1:** The motivation has already been explored. To my knowledge, the limitations of the "impute-then-predict" paradigm have been repeatedly discussed [1-3]. Additionally, CSDI also addresses the lack of authenticity in missing values during training and proposes a self-supervised learning strategy. Therefore, the authors should present a more compelling motivation. Furthermore, some imputation works that do not explicitly emphasize their applicability to prediction tasks (e.g., TimeNet, TimeMixer++) and some prediction works capable of directly handling missing values should be considered as competitors in this paper and discussed and compared in detail. Otherwise, this approach may lead to an overestimation of the pper's contributions.
> >
> > **W3:** The paper selects a relatively weak-performing model (DLinear) (which only consists of a few fully connected layers) and a heterogeneous model (TimesNet) to construct the "impute-then-predict" paradigm and uses them as the primary targets of criticism. The rationale for this motivation is insufficient. This experimental setup appears more like a "fragile" control group constructed to highlight the advantages of the proposed method, rather than a fair evaluation of the state-of-the-art or most commonly used paradigms in the real world. A more convincing analysis should directly compare with specialized, powerful imputation models (e.g., CSDI, ImputeFormer) or advanced prediction models capable of directly handling missing values (e.g., appropriate variants of D2STGNN, TimesNet), and delve into the potential limitations of these methods.
> >
> > **W4:** Furthermore, the authors' first contribution claims: "We perform a systematic empirical analysis of the dominant imputation-then-prediction paradigm for MTSF-M. We reveal that, guided only by a prediction objective, imputation modules can corrupt the observed data distribution and degrade prediction performance" seems exaggerated. The authors use a toy example to support their research, which is not systematic or concrete. Reliable findings should be based on either theoretical foundations or extensive experiments.
> >
> > **W6:** Several imputation models are missing, such as PriSTI, CSDI, and ImputeFormer. Additionally, some spatiotemporal forecasting models are missing, e.g., D2STGNN,STID,PatchSTG. Some time series models are also missing, such as DUET, CycleNet, and TimesNet. Some commonly used time series data sets, such as Weather, Traffic, etc., need to be considered.
> >
> > **W7:** Several imputation models are missing, such as PriSTI, CSDI, and ImputeFormer. Additionally, some spatiotemporal forecasting models are missing, e.g., D2STGNN,STID,PatchSTG. Some time series models are also missing, such as DUET, CycleNet, and TimesNet. Some commonly used time series data sets, such as Weather, Traffic, etc., need to be considered.
> >
> > **W10**: The work appears to be a combination of mature components. The authors' temporal analysis model is a standard Transformer, and the theory is derived from the well-established mutual information/variational inference theory. Therefore, it seems more like an engineering effort of combination rather than an original contribution.
>
>
> We appreciate the reviewer regarding the motivation and baseline selection. We address these concerns as follows:
>
> 1. **Motivation and Contribution:**  We acknowledge that the limitations of the 'imputation-then-prediction' paradigm have been discussed in prior works, and that direct prediction and Information Bottleneck (IB) frameworks are established in the literature. **However**, as detailed in our Introduction and Related Work, existing end-to-end methods still incorporate explicit imputation modules within their framework. **Consequently, they structurally remain within the "imputation-then-prediction" paradigm, merely shifting the imputation to a latent or module scale. The distribution shift and correlation destruction shown in Figure 1 are calculated based on real data and model outputs (t-SNE and Correlation Map), not a Toy Example. This intuitively reveals the risks of unsupervised imputation.** Our contribution extends beyond a simple critique to provide a systematic empirical re-evaluation and a specific architectural solution, CRIB.

---

> ### Author Response · Authors · 2025-11-23
> **Rebuttal by Authors**
>
> - **Detrimental Imputation:** We demonstrate that imputation without ground truth is often harmful. As visualized in Figure 1, methods following the "imputation-then-prediction" paradigm fail to recover the true data distribution and instead reinforce biased patterns from partial observations.
>
>    - **Performance Degradation:** We provide counter-intuitive evidence that imputation actively harms prediction accuracy. For instance, equipping the predictor TimeXer with the SOTA imputer TimesNet increases the MAE from 0.125 to 0.135.
>
>    - **Limitations of Vanilla IB:** We further observe that a naive application of the Information Bottleneck is insufficient. Figure 6 (Appendix F.2) shows that a direct IB-based Transformer yields degenerate attention maps, biasing the model towards local linearity and neglecting global dependencies. Furthermore, results in Table 2 (Section 4.3) indicate that CRIB without the consistency loss exhibits significantly higher variance. CRIB resolves this by integrating IB with consistency regularization and Unified-Attention to effectively filter noise.
>
>
>    - We also add the qualitative comparison between the **average** final prediction performance on missing rate of 20%, 40%, 60% and 70% of our method CRIB, BiTGraph, DLinear, TimesNet combined with DLinear, TimeXer, and TimesNet combined with TimeXer.
>
>      | PEMS-BAY |   CRIB    | BiTGraph | DLinear | TimesNet + DLinear | TimeXer | TimesNet + TimeXer |
>      | :------- | :-------: | :------: | :-----: | :----------------: | :-----: | :----------------: |
>      | MAE      | **0.093** |  0.413   |  0.156  |       0.148        | `0.125` |       0.135        |
>      | MSE      | **0.043** |  0.788   |  0.087  |       0.081        | `0.051` |       0.073        |
>
>
> **2. Baseline Selection**
>
> - **Justification:** We selected the combination of TimesNet and DLinear specifically to demonstrate that even current Time Series Forecasting models fail when applied within the "imputation-then-prediction" framework. We also conduct additional experiments by combining TimesNet and TimeXer, which reveal similar issues, demonstrating that the imputation without ground truth can often be detrimental.
> - **New Comparisons:** To ensure a comprehensive and fair evaluation, we selected baselines based on their code availability and their applicability to general MTSF-M scenarios, which typically lack predefined graph structures. Accordingly, we have expanded our experimental validation to include comparisons with **CSDI**, **ImputeFormer**, **NeuralCDE**, and **TimesNet** across a broader set of datasets. We exclude GinAR and S4M solely due to reproducibility issues with their publicly available code.
>
> **Extensive experiments on these datasets confirm that CRIB consistently achieves state-of-the-art performance, exhibiting superior robustness across varying missing rates compared to both direct prediction and imputation-based baselines.**
>
> |DatasetsInformation|ETTh1|ETTh2|ETTm1|ETTm2|Electricity|PEMS-BAY|Metr-LA|BeijingAir|Weather|Exchange|
> |--|:--:|:--:|:--:|:--:|:--:|:--:|:--:|:--:|:--:|:--:|
> |TimeSteps|17420|17420|69680|69680|26304|52116|34272|36000|52696|7588|
> |Variates|7|7|7|7|321|325|207|7|21|8|
>
>
> |ETTh1|0.0|0.0|0.2|0.2|0.4|0.4|0.6|0.6|0.7|0.7|
> |--|--|--|--|--|--|--|--|--|--|--|
> ||MAE|MSE|MAE|MSE|MAE|MSE|MAE|MSE|MAE|MSE|
> |CRIB|**0.195**|**0.163**|**0.238**|**0.211**|**0.274**|**0.273**|**0.342**|**0.427**|**0.411**|**0.624**|
> |SegRNN|0.209|0.167|0.247|0.215|0.297|0.307|0.380|0.506|0.455|0.737|
> |WPMixer|0.201|0.165|0.243|0.217|0.290|0.302|0.376|0.504|0.458|0.755|
> |iTransformer|0.211|0.172|0.250|0.219|0.338|0.316|0.385|0.494|0.453|0.731|
> |PatchTST|0.206|0.166|0.245|0.219|0.306|0.292|0.367|0.473|0.431|0.693|
> |DLinear|0.204|0.170|0.270|0.245|0.331|0.361|0.443|0.629|0.544|0.927|
> |PAttn|0.199|0.164|0.249|0.227|0.329|0.309|0.375|0.487|0.437|0.718|
> |CSDI|0.217|0.203|0.305|0.325|0.404|0.503|0.567|0.880|0.717|1.311|
> |NeuralCDE|0.291|0.236|0.340|0.315|0.408|0.460|0.522|0.761|0.629|1.101|
> |ImputeFormer|0.213|0.182|0.260|0.265|0.312|0.369|0.399|0.630|0.488|0.938|
> |TimesNet|0.225|0.178|0.256|0.234|0.297|0.302|0.373|0.457|0.443|0.627|

---

> ### Author Response · Authors · 2025-11-23
> **Rebuttal by Authors**
>
> |ETTh2|0.0|0.0|0.2|0.2|0.4|0.4|0.6|0.6|0.7|0.7|
> |--|--|--|--|--|--|--|--|--|--|--|
> ||MAE|MSE|MAE|MSE|MAE|MSE|MAE|MSE|MAE|MSE|
> |CRIB|**0.094**|**0.023**|**0.107**|**0.028**|**0.131**|**0.039**|**0.154**|**0.055**|**0.172**|**0.069**|
> |SegRNN|0.096|0.024|0.115|0.031|0.148|0.047|0.183|0.072|0.205|0.093|
> |WPMixer|0.095|0.024|0.116|0.030|0.147|0.045|0.202|0.086|0.218|0.104|
> |iTransformer|0.098|0.025|0.115|0.033|0.298|0.180|0.270|0.170|0.218|0.111|
> |PatchTST|0.096|0.024|0.109|0.030|0.215|0.125|0.216|0.121|0.186|0.083|
> |DLinear|0.098|0.025|0.142|0.040|0.181|0.065|0.253|0.135|0.325|0.228|
> |PAttn|0.096|0.025|0.116|0.033|0.289|0.173|0.243|0.133|0.196|0.085|
> |CSDI|0.113|0.033|0.255|0.192|0.404|0.399|0.618|0.840|0.798|1.303|
> |NeuralCDE|0.197|0.074|0.244|0.105|0.276|0.134|0.347|0.210|0.413|0.303|
> |ImputeFormer|0.101|0.026|0.122|0.034|0.140|0.049|0.173|0.061|0.174|0.071|
> |TimesNet|0.097|0.025|0.121|0.033|0.153|0.048|0.209|0.092|0.246|0.132|
>
>
> |Weather|0.0|0.0|0.2|0.2|0.4|0.4|0.6|0.6|0.7|0.7|
> |--|--|--|--|--|--|--|--|--|--|--|
> ||MAE|MSE|MAE|MSE|MAE|MSE|MAE|MSE|MAE|MSE|
> |CRIB|**0.028**|**0.016**|**0.038**|**0.025**|**0.050**|**0.033**|**0.062**|**0.057**|**0.070**|**0.075**|
> |SegRNN|0.031|0.021|0.042|0.028|0.050|0.037|0.066|0.061|0.076|0.081|
> |WPMixer|0.030|0.019|0.050|0.025|0.074|0.036|0.102|0.062|0.121|0.084|
> |iTransformer|0.030|0.018|0.044|0.031|0.152|0.167|0.175|0.223|0.101|0.138|
> |PatchTST|0.031|0.019|0.048|0.027|0.150|0.156|0.169|0.169|0.118|0.128|
> |DLinear|0.034|0.023|0.099|0.045|0.135|0.078|0.197|0.178|0.254|0.318|
> |PAttn|0.031|0.020|0.046|0.032|0.157|0.140|0.168|0.163|0.100|0.120|
> |CSDI|0.051|0.042|0.173|0.220|0.300|0.500|0.466|1.099|0.586|1.707|
> |NeuralCDE|0.051|0.029|0.073|0.036|0.095|0.060|0.150|0.150|0.204|0.284|
> |ImputeFormer|0.035|0.020|0.041|0.028|0.051|0.033|0.066|0.064|0.076|0.083|
> |TimesNet|0.028|0.016|0.062|0.033|0.088|0.047|0.128|0.089|0.166|0.154|
>
>
>
> |Exchange|0.0|0.0|0.2|0.2|0.4|0.4|0.6|0.6|0.7|0.7|
> |--|--|--|--|--|--|--|--|--|--|--|
> ||MAE|MSE|MAE|MSE|MAE|MSE|MAE|MSE|MAE|MSE|
> |CRIB|**0.0184**|**0.0009**|**0.0217**|**0.0016**|**0.0253**|**0.0015**|**0.0363**|**0.0031**|**0.0517**|**0.0058**|
> |SegRNN|0.0185|0.0010|0.0292|0.0031|0.0629|0.0129|0.1165|0.0363|0.1418|0.0490|
> |WPMixer|0.0187|0.0010|0.0324|0.0024|0.0783|0.0127|0.1364|0.0405|0.1836|0.0697|
> |iTransformer|0.0190|0.0010|0.0288|0.0024|0.1899|0.0770|0.2439|0.1060|0.2865|0.1413|
> |PatchTST|0.0186|0.0010|0.0273|0.0019|0.1416|0.0884|0.1978|0.1129|0.1321|0.0581|
> |DLinear|0.0205|0.0011|0.0795|0.0126|0.1469|0.0429|0.2599|0.1394|0.3778|0.3011|
> |PAttn|0.0187|0.0010|0.0227|0.0016|0.1408|0.1047|0.1283|0.0831|0.0718|0.0169|
> |CSDI|0.0264|0.0017|0.1992|0.2089|0.4358|0.5718|0.7911|1.3714|1.0766|2.2850|
> |NeuralCDE|0.3223|0.1724|0.2969|0.1592|0.3326|0.2146|0.4334|0.3814|0.5412|0.5912|
> |ImputeFormer|0.0344|0.0031|0.0364|0.0029|0.0431|0.0034|0.0608|0.0071|0.0846|0.0128|
> |TimesNet|0.0195|0.0011|0.0458|0.0041|0.0786|0.0130|0.1357|0.0404|0.1991|0.0881|
>
>
> |ETTm1|0.0|0.0|0.2|0.2|0.4|0.4|0.6|0.6|0.7|0.7|
> |--|--|--|--|--|--|--|--|--|--|--|
> ||MAE|MSE|MAE|MSE|MAE|MSE|MAE|MSE|MAE|MSE|
> |CRIB|**0.2000**|**0.2008**|**0.2368**|**0.2586**|**0.2734**|**0.3382**|**0.3414**|**0.5200**|**0.4013**|**0.6956**|
> |SegRNN|0.2265|0.2302|0.2616|0.3068|0.3005|0.4007|0.3678|0.5960|0.4266|0.8006|
> |WPMixer|0.2405|0.2741|0.2900|0.3593|0.3308|0.4645|0.4171|0.7328|0.4863|0.9928|
> |iTransformer|0.2309|0.2514|0.2909|0.3751|0.3593|0.4851|0.4365|0.7580|0.5034|1.0663|
> |PatchTST|0.2379|0.2648|0.2779|0.3486|0.3456|0.4671|0.4091|0.6769|0.4743|0.9325|
> |DLinear|0.2772|0.3504|0.3711|0.5499|0.4418|0.7412|0.5551|1.1067|0.6508|1.4721|
> |PAttn|0.2423|0.2782|0.2971|0.3864|0.3592|0.4801|0.4191|0.7022|0.4816|0.9837|
> |CSDI|0.4759|0.8854|0.5198|0.9345|0.5787|1.0499|0.7038|1.3801|0.8287|1.7733|
> |NeuralCDE|0.3331|0.4218|0.3809|0.5177|0.4391|0.6643|0.5457|0.9573|0.6278|1.2710|
> |ImputeFormer|0.2999|0.2682|0.3271|0.3282|0.3571|0.4749|0.4315|0.6978|0.4845|0.9035|
> |TimesNet|0.2100|0.2018|0.2584|0.2992|0.3040|0.4093|0.3993|0.6525|0.4828|0.8988|

---

> ### Author Response · Authors · 2025-11-23
> **Rebuttal by Authors**
>
> |ETTm2|0.0|0.0|0.2|0.2|0.4|0.4|0.6|0.6|0.7|0.7|
> |--|--|--|--|--|--|--|--|--|--|--|
> ||MAE|MSE|MAE|MSE|MAE|MSE|MAE|MSE|MAE|MSE|
> |CRIB|**0.0746**|**0.0152**|**0.0885**|**0.0203**|**0.1037**|**0.0258**|**0.1266**|**0.0384**|**0.1467**|**0.0520**|
> |SegRNN|0.0802|0.0175|0.0956|0.0231|0.1159|0.0305|0.1432|0.0477|0.1650|0.0619|
> |WPMixer|0.0821|0.0182|0.1059|0.0266|0.1355|0.0394|0.1790|0.0685|0.1899|0.0808|
> |iTransformer|0.0781|0.0161|0.1020|0.0284|0.2673|0.1688|0.2651|0.1671|0.1905|0.0861|
> |PatchTST|0.0824|0.0182|0.1005|0.0251|0.2041|0.1210|0.2425|0.1423|0.1803|0.0779|
> |DLinear|0.0918|0.0217|0.1405|0.0391|0.1743|0.0601|0.2412|0.1215|0.3104|0.2056|
> |PAttn|0.0828|0.0187|0.1024|0.0278|0.2441|0.1490|0.2398|0.1448|0.1728|0.0717|
> |CSDI|0.1313|0.0410|0.2687|0.1976|0.4131|0.4003|0.6221|0.8293|0.8024|1.2968|
> |NeuralCDE|0.1327|0.0363|0.1760|0.0591|0.1984|0.0765|0.2479|0.1194|0.3130|0.1857|
> |ImputeFormer|0.0813|0.0160|0.0888|0.0222|0.1088|0.0267|0.1360|0.0424|0.1602|0.0569|
> |TimesNet|0.0782|0.0165|0.1087|0.0260|0.1417|0.0406|0.1986|0.0811|0.2317|0.1142|
>
> |BejingAir|0.0|0.0|0.2|0.2|0.4|0.4|0.6|0.6|0.7|0.7|
> |--|--|--|--|--|--|--|--|--|--|--|
> ||MAE|MSE|MAE|MSE|MAE|MSE|MAE|MSE|MAE|MSE|
> |CRIB|**0.2526**|**0.2929**|**0.2758**|**0.3317**|**0.3106**|**0.4120**|**0.3728**|**0.5862**|**0.4337**|0.8010|
> |SegRNN|0.2552|0.3026|0.2849|0.3467|0.3188|0.4285|0.3860|0.5939|0.4412|**0.7595**|
> |WPMixer|0.2571|0.3085|0.2896|0.3651|0.3312|0.4601|0.4011|0.6503|0.4632|0.8730|
> |iTransformer|0.2606|0.3156|0.2923|0.3655|0.3545|0.4820|0.4124|0.6573|0.4691|0.8874|
> |PatchTST|0.2609|0.3008|0.2866|0.3421|0.3278|0.4305|0.3976|0.6458|0.4613|0.8540|
> |DLinear|0.2707|0.3263|0.3204|0.3853|0.3620|0.4760|0.4362|0.6799|0.5022|0.9358|
> |PAttn|0.2601|0.3102|0.2949|0.3603|0.3480|0.4689|0.4142|0.6624|0.4673|0.8745|
> |CSDI|0.3431|0.5141|0.4035|0.6010|0.4714|0.7584|0.5930|1.1378|0.7057|1.5761|
> |NeuralCDE|0.3167|0.3779|0.3478|0.4196|0.3950|0.5474|0.4750|0.7803|0.5456|1.0151|
> |ImputeFormer|0.2533|0.3189|0.2770|0.3496|0.3297|0.4645|0.4006|0.6110|0.4549|0.8327|
> |TimesNet|0.2583|0.2939|0.2922|0.3488|0.3322|0.4422|0.4017|0.6275|0.4789|0.9234|

---

> ### Author Response · Authors · 2025-11-23
> **Rebuttal by Authors**
>
> > **W2:** The "imputation-then-prediction" approach is not the mainstream framework adhered to by existing methods. Some datasets naturally contain missing values, and many prediction works do not perform an additional imputation step—they directly make predictions. The "imputation-then-prediction" paradigm generally involves simulating missing data through random masking, which is more akin to a self-supervised learning approach.
>
>
> We clarify the distinction between standard forecasting scenarios and the specific problem setting addressed in this work. **While existing direct prediction methods effectively handle datasets with sporadic missing values, they are not designed for the high-missingness regimes (e.g., 40%, 60%, 70%) that we investigate.** In such scenarios, standard methods often fail, leading practitioners to resort to the "imputation-then-prediction" strategy, which necessitates the critical evaluation presented in this paper.
>
> Furthermore, regarding the masking strategy, we emphasize that our random masking is not merely a self-supervised pre-training objective. **We explicitly apply it during the testing phase to simulate real-world deployment environments where data availability is severely compromised.** This ensures a rigorous evaluation of model robustness under conditions of significant information loss, distinct from standard self-supervised learning benchmarks.
>
>
>
> > **W5:** What challenges would other models face if performing direct prediction? Regarding these challenges, what specific strategies has your model implemented?
>
>
> We identify a critical challenge for existing direct prediction models (e.g., DLinear): they struggle to learn a robust mapping from highly sparse inputs—often filled with mean values—to complete targets, leading to overfitting on noise introduced by missing patterns. **This limitation is empirically demonstrated in Figure 1**, where the representative baseline DLinear exhibits **a significant distribution shift** compared to the ground truth (Figure 1a) and **fails to capture essential inter-variate correlations** (Figure 1c).
>
> **CRIB specifically addresses these issues through two core mechanisms: the $\mathcal{L}\_\text{Comp}$, which explicitly filters out task-irrelevant noise, and Consistency Regularization $\mathcal{L}\_\text{Consis}$, which enforces model robustness against varying missingness patterns.** By integrating these strategies, our method successfully recovers accurate data distributions and preserves complex variable dependencies, effectively overcoming the noise sensitivity that hinders standard direct prediction approaches, as shown in Figure 1.

---

> > ### Author Response · Authors · 2025-11-23
> > **Rebuttal by Authors**
> >
> > > **W6:** Although the authors introduce the information bottleneck theory, what is the utility of the lower bound proven in Section 3.4.2? In my view, if it's just an isolated value, it's difficult to grasp its meaning. For instance, the authors should introduce the lower/upper bounds of other methods for comparison to illustrate the significance of this bound.
> > >
> > > **W11:** Did the authors evaluate the performance of an ablated variant without the Consistency Regularization term? This is necessary to demonstrate that the performance gain indeed stems from this module.
> >
> >
> > The derivation in Section 3.4.2 proves that maximizing the Informativeness lower bound is equivalent to minimizing prediction error (MSE), providing a theoretical basis for directly using $\mathcal{L}_\text{Pred}$ in the loss function, ensuring the rationality of the optimization objective and helping model capture task relative information.
> >
> > We have done extra **ablation study on three loss of CRIB** to prove its effectiveness. **The ablation analysis across four datasets confirms the necessity of each component, as removing the Consistency, Regularization, or Prediction objectives consistently leads to performance degradation, validating their collective role in ensuring robust and accurate forecasting.**
> >
> > |ETTh1|0.2|0.2|0.4|0.4|0.6|0.6|0.7|0.7|
> > |--|:--:|:--:|:--:|:--:|:--:|:--:|:--:|:--:|
> > ||MAE|MSE|MAE|MSE|MAE|MSE|MAE|MSE|
> > |w/oConsis|0.237|0.206|0.274|0.270|0.338|0.413|0.402|0.584|
> > |w/oReg|0.236|0.206|0.274|0.270|0.339|0.410|0.400|0.579|
> > |w/oPred|0.432|0.559|0.505|0.698|0.655|1.066|0.796|1.486|
> > |Entire|0.220|0.171|0.251|0.249|0.267|0.296|0.288|0.361|
> >
> > |Elec|0.2|0.2|0.4|0.4|0.6|0.6|0.7|0.7|
> > |--|:--:|:--:|:--:|:--:|:--:|:--:|:--:|:--:|
> > ||MAE|MSE|MAE|MSE|MAE|MSE|MAE|MSE|
> > |w/oConsis|0.0201|0.0196|0.0267|0.0338|0.0346|0.0569|0.0415|0.0931|
> > |w/oReg|0.0186|0.0175|0.0244|0.0286|0.0322|0.0544|0.0399|0.0980|
> > |w/oPred|0.0881|0.5454|0.1243|0.9424|0.1818|1.8995|0.2283|2.9058|
> > |Entire|0.0150|0.0120|0.0230|0.0280|0.0300|0.0470|0.0380|0.0910|
> >
> > |Metr|0.2|0.2|0.4|0.4|0.6|0.6|0.7|0.7|
> > |--|:--:|:--:|:--:|:--:|:--:|:--:|:--:|:--:|
> > ||MAE|MSE|MAE|MSE|MAE|MSE|MAE|MSE|
> > |w/oConsis|0.271|0.332|0.257|0.275|0.271|0.306|0.314|0.366|
> > |w/oReg|0.253|0.307|0.251|0.275|0.266|0.307|0.311|0.364|
> > |w/oPred|0.442|0.418|0.624|0.529|0.985|1.177|1.276|1.959|
> > |Entire|0.248|0.271|0.249|0.272|0.265|0.305|0.309|0.356|
> >
> > |PEMS|0.2|0.2|0.4|0.4|0.6|0.6|0.7|0.7|
> > |--|:--:|:--:|:--:|:--:|:--:|:--:|:--:|:--:|
> > ||MAE|MSE|MAE|MSE|MAE|MSE|MAE|MSE|
> > |w/oConsis|0.0959|0.0470|0.0882|0.0389|0.0976|0.0465|0.1296|0.0596|
> > |w/oReg|0.0942|0.0453|0.0869|0.0373|0.0964|0.0456|0.1190|0.0591|
> > |w/oPred|0.4006|0.2405|0.5618|0.3631|1.0832|1.2727|2.1480|4.8111|
> > |Entire|0.0830|0.0340|0.0850|0.0350|0.0930|0.0430|0.1100|0.0580|

---

> ### Author Response · Authors · 2025-11-23
> **Rebuttal by Authors**
>
> > **W8:** Furthermore, I recommend denormalizing all prediction results; this is standard practice for PeMS and Metr-LA datasets.
> >
> > **W9:** I suggest adding MAPE as a third metric.
>
>
> We acknowledge the value of the reviewer's suggestions regarding the evaluation protocol. We agree that denormalizing results for traffic datasets and including MAPE as a metric are standard practices that enhance comparability. Due to the strict time constraints of the current phase, we are unable to re-run the full suite of experiments with these settings immediately. However, we commit to incorporating these metrics and post-processing steps in the final version of the manuscript.
>
>
>
>
> > **W12:** The authors' loss function involves three hyperparameters, which could clearly lead to optimization difficulties. Therefore, a detailed sensitivity analysis is required. Figure 4 alone is insufficient.
>
> We analyze hyperparameter sensitivity in Figure 4(b) and conduct additional experiments. **Empirical results indicate that the optimal settings are consistent across different datasets and missing rates.** In our default configuration, we set the weights for $\mathcal{L}\_\text{comp}$, $\mathcal{L}\_\text{pred}$, and $\mathcal{L}\_\text{consis}$ to $10^{-6}$, $1$, and $1$, respectively.
>
>
> |$\mathcal{L}_\text{Pred}$Weight-0%MissingRate|ETTh1|Exchange|Illness|
> |--|:--:|:--:|:--:|
> |0.1|0.201|0.01858|0.1495|
> |0.5|0.198|0.01858|0.1474|
> |1|0.198|0.01859|0.1457|
> |2|0.199|0.01859|0.1500|
> |5|0.200|0.01862|0.1533|
>
>
> |$\mathcal{L}_\text{Pred}$Weight-20%MissingRate|ETTh1|Exchange|Illness|
> |--|:--:|:--:|:--:|
> |0.1|0.243|0.03319|0.2466|
> |0.5|0.238|0.03005|0.2341|
> |1|0.237|0.02873|0.2306|
> |2|0.236|0.02614|0.2383|
> |5|0.237|0.02457|0.2337|
>
>
> |$\mathcal{L}_\text{Pred}$Weight-40%MissingRate|ETTh1|Exchange|Illness|
> |--|:--:|:--:|:--:|
> |0.1|0.281|0.04386|0.2741|
> |0.5|0.273|0.02943|0.2828|
> |1|0.274|0.02528|0.2573|
> |2|0.273|0.02614|0.2660|
> |5|0.273|0.02577|0.2709|
>
>
> |$\mathcal{L}_\text{Pred}$Weight-60%MissingRate|ETTh1|Exchange|Illness|
> |--|:--:|:--:|:--:|
> |0.1|0.348|0.04130|0.3631|
> |0.5|0.340|0.03621|0.3535|
> |1|0.337|0.03634|0.3703|
> |2|0.339|0.03634|0.3870|
> |5|0.338|0.02874|0.3509|
>
>
> |$\mathcal{L}_\text{Pred}$Weight-70%MissingRate|ETTh1|Exchange|Illness|
> |--|:--:|:--:|:--:|
> |0.1|0.412|0.03726|0.4796|
> |0.5|0.406|0.04227|0.4895|
> |1|0.403|0.05166|0.4914|
> |2|0.400|0.04751|0.4955|
> |5|0.401|0.03199|0.4497|
>
> |$\mathcal{L}_\text{Comp}$Weight-0%MissingRate|ETTh1|Exchange|Illness|
> |--|:--:|:--:|:--:|
> |10|0.214|0.01915|0.1713|
> |1|0.208|0.01876|0.1702|
> |1E-02|0.199|0.01861|0.1533|
> |1E-06|0.198|0.01859|0.1457|
> |1E-10|0.199|0.01859|0.1458|
>
> |$\mathcal{L}_\text{Comp}$Weight-20%MissingRate|ETTh1|Exchange|Illness|
> |--|:--:|:--:|:--:|
> |10|0.260|0.05088|0.2936|
> |1|0.251|0.04707|0.2802|
> |1E-02|0.238|0.02997|0.2536|
> |1E-06|0.237|0.02873|0.2306|
> |1E-10|0.237|0.02756|0.2349|
>
> |$\mathcal{L}_\text{Comp}$Weight-40%MissingRate|ETTh1|Exchange|Illness|
> |--|:--:|:--:|:--:|
> |10|0.314|0.08059|0.3263|
> |1|0.296|0.08352|0.2942|
> |1E-02|0.274|0.03186|0.2679|
> |1E-06|0.274|0.02528|0.2573|
> |1E-10|0.273|0.02603|0.2583|
>
> |$\mathcal{L}_\text{Comp}$Weight-60%MissingRate|ETTh1|Exchange|Illness|
> |--|:--:|:--:|:--:|
> |10|0.417|0.21306|0.4830|
> |1|0.365|0.13925|0.4377|
> |1E-02|0.341|0.04349|0.3710|
> |1E-06|0.337|0.03634|0.3703|
> |1E-10|0.339|0.04173|0.3625|
>
> |$\mathcal{L}_\text{Comp}$Weight-70%MissingRate|ETTh1|Exchange|Illness|
> |--|:--:|:--:|:--:|
> |10|0.506|0.28510|0.6264|
> |1|0.437|0.22926|0.5119|
> |1E-02|0.406|0.04991|0.4588|
> |1E-06|0.403|0.05166|0.4914|
> |1E-10|0.403|0.04077|0.4949|
>
>
>
>
> |$\mathcal{L}_\text{Consis}$Weight-0%MissingRate|ETTh1|Exchange|Illness|
> |--|:--:|:--:|:--:|
> |0.1|0.201|0.01859|0.1556|
> |0.5|0.199|0.01859|0.1497|
> |1|0.198|0.01859|0.1457|
> |2|0.199|0.01858|0.1477|
> |5|0.200|0.01858|0.1497|
>
>
> |$\mathcal{L}_\text{Consis}$Weight-20%MissingRate|ETTh1|Exchange|Illness|
> |--|:--:|:--:|:--:|
> |0.1|0.237|0.02306|0.2289|
> |0.5|0.236|0.02603|0.2348|
> |1|0.237|0.02873|0.2306|
> |2|0.239|0.03059|0.2344|
> |5|0.240|0.03117|0.2402|
>
>
> |$\mathcal{L}_\text{Consis}$Weight-40%MissingRate|ETTh1|Exchange|Illness|
> |--|:--:|:--:|:--:|
> |0.1|0.274|0.02697|0.2527|
> |0.5|0.274|0.02726|0.2582|
> |1|0.274|0.02528|0.2573|
> |2|0.275|0.03001|0.2801|
> |5|0.278|0.03771|0.2808|
>
>
> |$\mathcal{L}_\text{Consis}$Weight-60%MissingRate|ETTh1|Exchange|Illness|
> |--|:--:|:--:|:--:|
> |0.1|0.340|0.03013|0.3519|
> |0.5|0.339|0.03783|0.3647|
> |1|0.337|0.03634|0.3703|
> |2|0.340|0.03516|0.3414|
> |5|0.344|0.03756|0.3524|
>
> |$\mathcal{L}_\text{Consis}$Weight-70%MissingRate|ETTh1|Exchange|Illness|
> |--|:--:|:--:|:--:|
> |0.1|0.400|0.03408|0.4587|
> |0.5|0.401|0.03678|0.4919|
> |1|0.403|0.05166|0.4914|
> |2|0.404|0.04267|0.5009|
> |5|0.410|0.03573|0.5203|

---

> > ### Author Response · Authors · 2025-11-23
> > **Rebuttal by Authors**
> >
> > > **W13:** The code cannot be run directly. It would be better if you can add more detailed instructions for using the code.
> >
> >
> > We apologize for the inconvenience regarding the usability of the codebase. To ensure full reproducibility, we commit to updating the GitHub repository to include the additional baselines and datasets introduced during this rebuttal phase. Furthermore, we will provide a comprehensive `README.md`, a precise `requirements.txt` for environment configuration, and a unified `run.sh` script to facilitate one-click reproduction of all reported experimental results.

---

> > ### Comment · Reviewer_2Cic · 2025-11-26
> >
> > Thank you for your response.
> >
> > The rebuttal period is three weeks, and as far as I know, this should be sufficient time.
> >
> > Regarding Figure 1, is it possible not to use the asymmetric structure? DLinear is a relatively weak model, and highlighting our motivation by comparing against such a low-performing model is not very convincing.

---

> > > ### Author Response · Authors · 2025-11-29
> > > **Response to Reviewer 2Cic**
> > >
> > > Thank you for the suggestion.
> > >
> > > In the revised manuscript, we have updated the **Introduction** to include an analysis of **TimeXer**. Since TimeXer is a very strong baseline in our experiments, this validates that our motivation still holds for the powerful model.

---

### Official Review · Reviewer_ek3n · 2025-10-31

**Soundness:** 3
**Presentation:** 3
**Contribution:** 2
**Rating:** 4
**Confidence:** 3

**Summary:**

This paper presents CRIB, a novel method for Multivariate Time Series Forecasting with Missing Values, demonstrating state-of-the-art performance across four diverse datasets in an extensive experimental evaluation. The work addresses a critically important real-world challenge. However, several significant limitations temper these strengths. The practical relevance is undermined as experiments use artificially induced missing data on complete, highly periodic datasets, failing to test on real-world, non-stationary, or high-dimensional data. The problem setup's generalizability is weakened by an imputation strategy that ignores realistic missing mechanisms (e.g., MNAR). Furthermore, the core innovation of unified-univariate attention lacks theoretical grounding, and claims of superiority over other models remain unconvincing without deeper analysis into why they fail. Consequently, while CRIB shows promising results, its validity and applicability to genuine real-world scenarios are questionable.

**Strengths:**

S1. The paper addresses the increasingly important problem of Multivariate Time Series Forecasting with Missing values, a critical challenge in real-world applications such as healthcare, environmental monitoring, and industrial IoT, where data completeness cannot be assumed.

S2. The experimental section is extensive and methodologically sound, covering four diverse datasets

S3. The proposed method, CRIB, achieves state-of-the-art performance across all datasets and metrics

**Weaknesses:**

W1. The paper claims to address real-world scenarios where ground-truth values are unavailable. However, the two-stage imputation strategy—training TimesNet on 10% missing data and applying it to higher missing rates—does not reflect real-world missing mechanisms (e.g., MNAR due to sensor failure or human behavior). The absence of discussion on missing mechanisms (MCAR/MAR/MNAR) weakens the practical motivation and limits the generalizability of the problem setup.

W2. The paper asserts superiority over existing methods but fails to explain why models like SAITS or BiTGraph degrade under high missing rates. A deeper analysis—e.g., error decomposition, attention pattern visualization, or gradient sensitivity—would strengthen the motivation. Without such analysis, the claim that "existing methods fail" remains superficial and unconvincing.

W3. The unified-univariate attention is a core innovation, yet the paper offers no theoretical analysis of its properties—e.g., representational capacity, gradient flow, or robustness to missing data. Why is this structure more suitable for MTSF-M than standard self-attention or cross-variate attention? Without such grounding, the method appears heuristic rather than principled.

W4. Experiments are conducted on only four datasets: PEMS-BAY, Metr-LA, ETTh1, and Electricity. These are all highly periodic, low-noise, relatively low-dimensional and . The lack of testing on non-stationary, high-dimensional (e.g., >1000 variates), or irregularly sampled datasets limits the generalizability of the results and raises concerns about overfitting to specific data characteristics

W5. The experimental evaluation is conducted on only four datasets. While these are commonly used in time series forecasting, they share highly similar characteristics: strong periodicity, low noise, regular sampling, and relatively low dimensionality (tens to hundreds of variables). Crucially, none of these datasets contain real missing values. Instead, the authors artificially inject missing data under controlled patterns (point, block, column), which significantly weakens the practical relevance of the study.

**Questions:**

Please see Weaknesses

---

> ### Author Response · Authors · 2025-11-23
> **Rebuttal by Authors**
>
> We sincerely thank Reviewer ek3n for the thoughtful and constructive reviews of our manuscript! Based on your questions and recommendations, we give point-by-point responses to your comments and describe the revisions we made to address them.
>
> ---
>
> > **W1:** The paper claims to address real-world scenarios where ground-truth values are unavailable. However, the two-stage imputation strategy—training TimesNet on 10% missing data and applying it to higher missing rates—does not reflect real-world missing mechanisms (e.g., MNAR due to sensor failure or human behavior). The absence of discussion on missing mechanisms (MCAR/MAR/MNAR) weakens the practical motivation and limits the generalizability of the problem setup.
>
>
> Our experimental setup is designed to **simulate practical scenarios where inference missing rates are unknown or exceed those seen during training, and ground truth for missing values is unavailable** (Section 4.1). While we primarily focus on random missingness, **this formulation is sufficient to reveal the inherent limitations of the "impute-then-predict" paradigm**. We employ Point, Block, and Column missing patterns to systematically simulate different types of distribution shifts.
>
> We acknowledge that robustness against complex MAR and MNAR mechanisms represents a distinct challenge, and we will explicitly discuss this as an important direction for future work in the Limitations section.
>
>
>
> > **W2:** The paper asserts superiority over existing methods but fails to explain why models like SAITS or BiTGraph degrade under high missing rates. A deeper analysis—e.g., error decomposition, attention pattern visualization, or gradient sensitivity—would strengthen the motivation. Without such analysis, the claim that "existing methods fail" remains superficial and unconvincing.
>
> We address the performance degradation of existing methods through the empirical analysis presented in Figure 1 and the Introduction (two core observations). Our observations identify the root cause: in the absence of ground truth supervision, imputation modules generate values that significantly deviate from the true data distribution. **This unsupervised process induces a distribution shift and disrupts inherent variate correlations**, as evidenced by the chaotic correlation maps. **These erroneous imputations act as noisy inputs for the subsequent forecasting module, directly causing error propagation and resulting in the observed performance decline.**
>
>
>
> > **W3:** The unified-univariate attention is a core innovation, yet the paper offers no theoretical analysis of its properties—e.g., representational capacity, gradient flow, or robustness to missing data. Why is this structure more suitable for MTSF-M than standard self-attention or cross-variate attention? Without such grounding, the method appears heuristic rather than principled.
>
>
> We clarify that the design of the unified-univariate attention (Section 3.2) addresses the limitations of rigid structural priors in sparse data regimes. By eliminating the boundaries between variates and time steps, our method treats all tokens uniformly. In scenarios with scattered missing values, predefined spatial-temporal structures often fail to capture valid dependencies. In contrast, our flattened approach maximizes flexibility, allowing the model to adaptively extract informative signals from any available observed patches, regardless of their original dimension.
>
> **This advantage is empirically validated in Figure 6 (Appendix F.2)**, where we further compare it with a direct IB-based vanilla Transformer. While standard separated attention mechanisms produce degenerate maps biased towards local linearity, **our unified attention successfully captures meaningful global dependencies and both intra-variate and inter-variate correlations.**

---

> ### Author Response · Authors · 2025-11-23
> **Rebuttal by Authors**
>
> > **W4:** Experiments are conducted on only four datasets: PEMS-BAY, Metr-LA, ETTh1, and Electricity. These are all highly periodic, low-noise, relatively low-dimensional and . The lack of testing on non-stationary, high-dimensional (e.g., >1000 variates), or irregularly sampled datasets limits the generalizability of the results and raises concerns about overfitting to specific data characteristics.
> >
> > **W5:** The experimental evaluation is conducted on only four datasets. While these are commonly used in time series forecasting, they share highly similar characteristics: strong periodicity, low noise, regular sampling, and relatively low dimensionality (tens to hundreds of variables). Crucially, none of these datasets contain real missing values. Instead, the authors artificially inject missing data under controlled patterns (point, block, column), which significantly weakens the practical relevance of the study.
>
>
>
> To address the concern regarding generalization, we have expanded our experimental evaluation with additional datasets and baselines. While we acknowledge the importance of testing on extremely high-dimensional data, we limit our current scope to datasets with fewer than 1000 variates due to computational constraints.
>
> The detailed properties of these additional datasets and the corresponding performance benchmarks are presented below, demonstrating the robustness of our method across diverse data characteristics.
>
> **Extensive experiments on these datasets confirm that CRIB consistently achieves state-of-the-art performance, exhibiting superior robustness across varying missing rates compared to both direct prediction and imputation-based baselines.**
>
> |DatasetsInformation|ETTh1|ETTh2|ETTm1|ETTm2|Electricity|PEMS-BAY|Metr-LA|BeijingAir|Weather|Exchange|
> |--|:--:|:--:|:--:|:--:|:--:|:--:|:--:|:--:|:--:|:--:|
> |TimeSteps|17420|17420|69680|69680|26304|52116|34272|36000|52696|7588|
> |Variates|7|7|7|7|321|325|207|7|21|8|
>
> |ETTh1|0.0|0.0|0.2|0.2|0.4|0.4|0.6|0.6|0.7|0.7|
> |--|--|--|--|--|--|--|--|--|--|--|
> ||MAE|MSE|MAE|MSE|MAE|MSE|MAE|MSE|MAE|MSE|
> |CRIB|**0.195**|**0.163**|**0.238**|**0.211**|**0.274**|**0.273**|**0.342**|**0.427**|**0.411**|**0.624**|
> |SegRNN|0.209|0.167|0.247|0.215|0.297|0.307|0.380|0.506|0.455|0.737|
> |WPMixer|0.201|0.165|0.243|0.217|0.290|0.302|0.376|0.504|0.458|0.755|
> |iTransformer|0.211|0.172|0.250|0.219|0.338|0.316|0.385|0.494|0.453|0.731|
> |PatchTST|0.206|0.166|0.245|0.219|0.306|0.292|0.367|0.473|0.431|0.693|
> |DLinear|0.204|0.170|0.270|0.245|0.331|0.361|0.443|0.629|0.544|0.927|
> |PAttn|0.199|0.164|0.249|0.227|0.329|0.309|0.375|0.487|0.437|0.718|
> |CSDI|0.217|0.203|0.305|0.325|0.404|0.503|0.567|0.880|0.717|1.311|
> |NeuralCDE|0.291|0.236|0.340|0.315|0.408|0.460|0.522|0.761|0.629|1.101|
> |ImputeFormer|0.213|0.182|0.260|0.265|0.312|0.369|0.399|0.630|0.488|0.938|
> |TimesNet|0.225|0.178|0.256|0.234|0.297|0.302|0.373|0.457|0.443|0.627|
>
>
> |ETTh2|0.0|0.0|0.2|0.2|0.4|0.4|0.6|0.6|0.7|0.7|
> |--|--|--|--|--|--|--|--|--|--|--|
> ||MAE|MSE|MAE|MSE|MAE|MSE|MAE|MSE|MAE|MSE|
> |CRIB|**0.094**|**0.023**|**0.107**|**0.028**|**0.131**|**0.039**|**0.154**|**0.055**|**0.172**|**0.069**|
> |SegRNN|0.096|0.024|0.115|0.031|0.148|0.047|0.183|0.072|0.205|0.093|
> |WPMixer|0.095|0.024|0.116|0.030|0.147|0.045|0.202|0.086|0.218|0.104|
> |iTransformer|0.098|0.025|0.115|0.033|0.298|0.180|0.270|0.170|0.218|0.111|
> |PatchTST|0.096|0.024|0.109|0.030|0.215|0.125|0.216|0.121|0.186|0.083|
> |DLinear|0.098|0.025|0.142|0.040|0.181|0.065|0.253|0.135|0.325|0.228|
> |PAttn|0.096|0.025|0.116|0.033|0.289|0.173|0.243|0.133|0.196|0.085|
> |CSDI|0.113|0.033|0.255|0.192|0.404|0.399|0.618|0.840|0.798|1.303|
> |NeuralCDE|0.197|0.074|0.244|0.105|0.276|0.134|0.347|0.210|0.413|0.303|
> |ImputeFormer|0.101|0.026|0.122|0.034|0.140|0.049|0.173|0.061|0.174|0.071|
> |TimesNet|0.097|0.025|0.121|0.033|0.153|0.048|0.209|0.092|0.246|0.132|
>
>
> |Weather|0.0|0.0|0.2|0.2|0.4|0.4|0.6|0.6|0.7|0.7|
> |--|--|--|--|--|--|--|--|--|--|--|
> ||MAE|MSE|MAE|MSE|MAE|MSE|MAE|MSE|MAE|MSE|
> |CRIB|**0.028**|**0.016**|**0.038**|**0.025**|**0.050**|**0.033**|**0.062**|**0.057**|**0.070**|**0.075**|
> |SegRNN|0.031|0.021|0.042|0.028|0.050|0.037|0.066|0.061|0.076|0.081|
> |WPMixer|0.030|0.019|0.050|0.025|0.074|0.036|0.102|0.062|0.121|0.084|
> |iTransformer|0.030|0.018|0.044|0.031|0.152|0.167|0.175|0.223|0.101|0.138|
> |PatchTST|0.031|0.019|0.048|0.027|0.150|0.156|0.169|0.169|0.118|0.128|
> |DLinear|0.034|0.023|0.099|0.045|0.135|0.078|0.197|0.178|0.254|0.318|
> |PAttn|0.031|0.020|0.046|0.032|0.157|0.140|0.168|0.163|0.100|0.120|
> |CSDI|0.051|0.042|0.173|0.220|0.300|0.500|0.466|1.099|0.586|1.707|
> |NeuralCDE|0.051|0.029|0.073|0.036|0.095|0.060|0.150|0.150|0.204|0.284|
> |ImputeFormer|0.035|0.020|0.041|0.028|0.051|0.033|0.066|0.064|0.076|0.083|
> |TimesNet|0.028|0.016|0.062|0.033|0.088|0.047|0.128|0.089|0.166|0.154|

---

> ### Author Response · Authors · 2025-11-23
> **Rebuttal by Authors**
>
> |Exchange|0.0|0.0|0.2|0.2|0.4|0.4|0.6|0.6|0.7|0.7|
> |--|--|--|--|--|--|--|--|--|--|--|
> ||MAE|MSE|MAE|MSE|MAE|MSE|MAE|MSE|MAE|MSE|
> |CRIB|**0.0184**|**0.0009**|**0.0217**|**0.0016**|**0.0253**|**0.0015**|**0.0363**|**0.0031**|**0.0517**|**0.0058**|
> |SegRNN|0.0185|0.0010|0.0292|0.0031|0.0629|0.0129|0.1165|0.0363|0.1418|0.0490|
> |WPMixer|0.0187|0.0010|0.0324|0.0024|0.0783|0.0127|0.1364|0.0405|0.1836|0.0697|
> |iTransformer|0.0190|0.0010|0.0288|0.0024|0.1899|0.0770|0.2439|0.1060|0.2865|0.1413|
> |PatchTST|0.0186|0.0010|0.0273|0.0019|0.1416|0.0884|0.1978|0.1129|0.1321|0.0581|
> |DLinear|0.0205|0.0011|0.0795|0.0126|0.1469|0.0429|0.2599|0.1394|0.3778|0.3011|
> |PAttn|0.0187|0.0010|0.0227|0.0016|0.1408|0.1047|0.1283|0.0831|0.0718|0.0169|
> |CSDI|0.0264|0.0017|0.1992|0.2089|0.4358|0.5718|0.7911|1.3714|1.0766|2.2850|
> |NeuralCDE|0.3223|0.1724|0.2969|0.1592|0.3326|0.2146|0.4334|0.3814|0.5412|0.5912|
> |ImputeFormer|0.0344|0.0031|0.0364|0.0029|0.0431|0.0034|0.0608|0.0071|0.0846|0.0128|
> |TimesNet|0.0195|0.0011|0.0458|0.0041|0.0786|0.0130|0.1357|0.0404|0.1991|0.0881|
>
>
> |ETTm1|0.0|0.0|0.2|0.2|0.4|0.4|0.6|0.6|0.7|0.7|
> |--|--|--|--|--|--|--|--|--|--|--|
> ||MAE|MSE|MAE|MSE|MAE|MSE|MAE|MSE|MAE|MSE|
> |CRIB|**0.2000**|**0.2008**|**0.2368**|**0.2586**|**0.2734**|**0.3382**|**0.3414**|**0.5200**|**0.4013**|**0.6956**|
> |SegRNN|0.2265|0.2302|0.2616|0.3068|0.3005|0.4007|0.3678|0.5960|0.4266|0.8006|
> |WPMixer|0.2405|0.2741|0.2900|0.3593|0.3308|0.4645|0.4171|0.7328|0.4863|0.9928|
> |iTransformer|0.2309|0.2514|0.2909|0.3751|0.3593|0.4851|0.4365|0.7580|0.5034|1.0663|
> |PatchTST|0.2379|0.2648|0.2779|0.3486|0.3456|0.4671|0.4091|0.6769|0.4743|0.9325|
> |DLinear|0.2772|0.3504|0.3711|0.5499|0.4418|0.7412|0.5551|1.1067|0.6508|1.4721|
> |PAttn|0.2423|0.2782|0.2971|0.3864|0.3592|0.4801|0.4191|0.7022|0.4816|0.9837|
> |CSDI|0.4759|0.8854|0.5198|0.9345|0.5787|1.0499|0.7038|1.3801|0.8287|1.7733|
> |NeuralCDE|0.3331|0.4218|0.3809|0.5177|0.4391|0.6643|0.5457|0.9573|0.6278|1.2710|
> |ImputeFormer|0.2999|0.2682|0.3271|0.3282|0.3571|0.4749|0.4315|0.6978|0.4845|0.9035|
> |TimesNet|0.2100|0.2018|0.2584|0.2992|0.3040|0.4093|0.3993|0.6525|0.4828|0.8988|
>
>
> |ETTm2|0.0|0.0|0.2|0.2|0.4|0.4|0.6|0.6|0.7|0.7|
> |--|--|--|--|--|--|--|--|--|--|--|
> ||MAE|MSE|MAE|MSE|MAE|MSE|MAE|MSE|MAE|MSE|
> |CRIB|**0.0746**|**0.0152**|**0.0885**|**0.0203**|**0.1037**|**0.0258**|**0.1266**|**0.0384**|**0.1467**|**0.0520**|
> |SegRNN|0.0802|0.0175|0.0956|0.0231|0.1159|0.0305|0.1432|0.0477|0.1650|0.0619|
> |WPMixer|0.0821|0.0182|0.1059|0.0266|0.1355|0.0394|0.1790|0.0685|0.1899|0.0808|
> |iTransformer|0.0781|0.0161|0.1020|0.0284|0.2673|0.1688|0.2651|0.1671|0.1905|0.0861|
> |PatchTST|0.0824|0.0182|0.1005|0.0251|0.2041|0.1210|0.2425|0.1423|0.1803|0.0779|
> |DLinear|0.0918|0.0217|0.1405|0.0391|0.1743|0.0601|0.2412|0.1215|0.3104|0.2056|
> |PAttn|0.0828|0.0187|0.1024|0.0278|0.2441|0.1490|0.2398|0.1448|0.1728|0.0717|
> |CSDI|0.1313|0.0410|0.2687|0.1976|0.4131|0.4003|0.6221|0.8293|0.8024|1.2968|
> |NeuralCDE|0.1327|0.0363|0.1760|0.0591|0.1984|0.0765|0.2479|0.1194|0.3130|0.1857|
> |ImputeFormer|0.0813|0.0160|0.0888|0.0222|0.1088|0.0267|0.1360|0.0424|0.1602|0.0569|
> |TimesNet|0.0782|0.0165|0.1087|0.0260|0.1417|0.0406|0.1986|0.0811|0.2317|0.1142|
>
> |BejingAir|0.0|0.0|0.2|0.2|0.4|0.4|0.6|0.6|0.7|0.7|
> |--|--|--|--|--|--|--|--|--|--|--|
> ||MAE|MSE|MAE|MSE|MAE|MSE|MAE|MSE|MAE|MSE|
> |CRIB|**0.2526**|**0.2929**|**0.2758**|**0.3317**|**0.3106**|**0.4120**|**0.3728**|**0.5862**|**0.4337**|0.8010|
> |SegRNN|0.2552|0.3026|0.2849|0.3467|0.3188|0.4285|0.3860|0.5939|0.4412|**0.7595**|
> |WPMixer|0.2571|0.3085|0.2896|0.3651|0.3312|0.4601|0.4011|0.6503|0.4632|0.8730|
> |iTransformer|0.2606|0.3156|0.2923|0.3655|0.3545|0.4820|0.4124|0.6573|0.4691|0.8874|
> |PatchTST|0.2609|0.3008|0.2866|0.3421|0.3278|0.4305|0.3976|0.6458|0.4613|0.8540|
> |DLinear|0.2707|0.3263|0.3204|0.3853|0.3620|0.4760|0.4362|0.6799|0.5022|0.9358|
> |PAttn|0.2601|0.3102|0.2949|0.3603|0.3480|0.4689|0.4142|0.6624|0.4673|0.8745|
> |CSDI|0.3431|0.5141|0.4035|0.6010|0.4714|0.7584|0.5930|1.1378|0.7057|1.5761|
> |NeuralCDE|0.3167|0.3779|0.3478|0.4196|0.3950|0.5474|0.4750|0.7803|0.5456|1.0151|
> |ImputeFormer|0.2533|0.3189|0.2770|0.3496|0.3297|0.4645|0.4006|0.6110|0.4549|0.8327|
> |TimesNet|0.2583|0.2939|0.2922|0.3488|0.3322|0.4422|0.4017|0.6275|0.4789|0.9234|

---

### Official Review · Reviewer_K95D · 2025-11-01

**Soundness:** 2
**Presentation:** 3
**Contribution:** 2
**Rating:** 2
**Confidence:** 4

**Summary:**

This paper proposes that imputing missing values may be inaccurate, which can affect the accuracy of time simulation and thereby harm the precision of future predictions. To mitigate the errors brought by imputation, the paper further proposes a novel framework called CRIB, constructed based on the information bottleneck principle. CRIB combines a unified variable attention mechanism with a consistency regularization scheme to learn robust representations, filter out noise introduced by missing values, and meanwhile preserve the fundamental predictive signals. The experimental results of the paper show that the proposed method outperforms existing classic baselines.

**Strengths:**

This paper proposes that imputing missing values may be inaccurate, which can affect the accuracy of time simulation and thereby harm the precision of future predictions. To mitigate the errors brought by imputation, the paper further proposes a novel framework called CRIB, constructed based on the information bottleneck principle. CRIB combines a unified variable attention mechanism with a consistency regularization scheme to learn robust representations, filter out noise introduced by missing values, and meanwhile preserve the fundamental predictive signals. The experimental results of the paper show that the proposed method outperforms existing classic baselines.

**Weaknesses:**

1. The experiments in this paper are unreasonable. From the source code, this paper fills the missing value part with 0, which is actually imputing the data with a value of 0, severely disrupting the original temporal patterns. In the subsequent forward and backward propagation processes, the filled 0 values are involved in the calculations. This will seriously affect the model's optimization and lead to training failure. This may also be why the baselines in Figure 1 perform poorly, as they are trained under conditions where the data patterns are severely disrupted (real values are randomly replaced with 0). It may be that even simple non-model cubic function interpolation would perform better than the models trained in Figure 1. Missing value parts should not be involved in calculations. They should be handled using masks during the network computation process, or using a process like neural CDE for processing, instead of filling the missing parts with zero at the input stage.

2. Due to the unreasonable handling of missing values, many of the modules proposed in the paper are unreliable. For example, the KL divergence under IB theory and the consistency constraints under noise are closely related to setting missing values to 0 for participation in forward and backward propagation, but this is meaningless. For example, random noise will replace the 0 value positions with more meaningful values, and the effectiveness may be because the 0 values are replaced.

3. Although the logic of using INFORMATION BOTTLENECK GUIDANCE in the paper is reasonable, the approach is confusing. Calculating the Gaussian posterior based on features and making it close to the standard Gaussian prior with the KL divergence seems to have no supervisory information. Why can this minimize mutual information and filter out irrelevant information?

4. Some parts of the paper are vaguely described. W/o IB means removing the compactness and informativeness guidance of IB. The former is Eq. 9, but what is the latter? The two are mixed in the ablation study, and the effect of removing Eq. 9 alone is not shown.

5. The paper lacks key baselines. For example, baselines for handling irregular and missing data, such as Neural ODE or CDE.

6. The paper flattens all variable patches to calculate attention, which is not a very efficient or reasonable approach. When the number of variables is large, e.g., a traffic dataset, this method will become infeasible due to the overly long patch sequence.

7. The experiments in the paper are not solid. The main experiments are only conducted on four datasets, and the ablation studies are only performed on one dataset.

**Questions:**

Although the logic of using INFORMATION BOTTLENECK GUIDANCE in the paper is reasonable, the approach is confusing. Calculating the Gaussian posterior based on features and making it close to the standard Gaussian prior with the KL divergence seems to have no supervisory information. Why can this minimize mutual information and filter out irrelevant information?

**Details Of Ethics Concerns:**

None.

---

> ### Author Response · Authors · 2025-11-23
>
> We sincerely thank Reviewer K95D for the thoughtful and constructive reviews of our manuscript! Based on your questions and recommendations, we give point-by-point responses to your comments and describe the revisions we made to address them.
>
> ---
>
>
> > **W1:** The experiments in this paper are unreasonable. From the source code, this paper fills the missing value part with 0, which is actually imputing the data with a value of 0, severely disrupting the original temporal patterns. In the subsequent forward and backward propagation processes, the filled 0 values are involved in the calculations. This will seriously affect the model's optimization and lead to training failure. This may also be why the baselines in Figure 1 perform poorly, as they are trained under conditions where the data patterns are severely disrupted (real values are randomly replaced with 0). It may be that even simple non-model cubic function interpolation would perform better than the models trained in Figure 1. Missing value parts should not be involved in calculations. They should be handled using masks during the network computation process, or using a process like neural CDE for processing, instead of filling the missing parts with zero at the input stage.
>
>
> We appreciate the reviewer's question regarding missing value handling. We clarify that our implementation follows standard deep learning practices and does not disrupt model optimization.
>
> 1. **Zero-filling is Mean Imputation**: We normalize all input data to have zero mean and unit variance. **Consequently, filling missing values with 0 is mathematically equivalent to mean imputation**. This is a standard preprocessing technique. The model treats these zeros as the mean value of the distribution, which serves as a neutral and stable input for optimization.
>
> 2. **Empirical Validation**: Contrary to the concern that zero-filling causes training failure, our results demonstrate its effectiveness. As stated in **Observation 2** in section 4.2, using this zero-filling strategy significantly outperforms complex "imputation-then-prediction" pipelines (both two-stage and end-to-end methods). This indicates that zero-filling is a robust baseline strategy. **The performance degradation observed in other methods stems from the noise introduced by unsupervised imputation modules, not from the zero-filling initialization itself.**
>
>
>
> > **W2:** Due to the unreasonable handling of missing values, many of the modules proposed in the paper are unreliable. For example, the KL divergence under IB theory and the consistency constraints under noise are closely related to setting missing values to 0 for participation in forward and backward propagation, but this is meaningless. For example, random noise will replace the 0 value positions with more meaningful values, and the effectiveness may be because the 0 values are replaced.
>
>
> Thanks for your comments. We clarify two key points regarding our methodology and experimental results:
>
> 1. **Clarification on Data Augmentation**: The reviewer suggests that random noise replaces zero-filled values with meaningful information. This is incorrect. In our augmentation strategy, we mask an additional subset of observed values by setting them to zero. This process effectively reduces the information available to the model, **making the reconstruction task more challenging rather than easier**. The augmentation does not introduce new information into the missing parts.
>
> 2. **Necessity of KL Divergence and Consistency**:
>    - KL loss is an approximation of minimizing the mutual information of 𝑋o and 𝑍, which works with 𝐿pred to help the cal{L}_\text{pred}$ to help model understand what is really task-related information, thus filtering noise brought by the missing values.
>    - We empirically demonstrate that the KL divergence and consistency constraints are not meaningless but essential. A naive application of the Information Bottleneck is insufficient. As visualized in Figure 6 (Appendix F.2), a direct IB-based Transformer produces degenerate attention maps, biasing the model towards local linearity and ignoring global dependencies.
>    - Additionally, the ablation study in Table 2 (Section 4.3) shows that removing the consistency loss significantly increases variance.
>
>
> **These results confirm that the proposed KL and consistency terms are critical for stabilizing the optimization and handling noise effectively.**

---

> ### Author Response · Authors · 2025-11-23
> **Rebuttal by Authors**
>
> > **W3:** Although the logic of using INFORMATION BOTTLENECK GUIDANCE in the paper is reasonable, the approach is confusing. Calculating the Gaussian posterior based on features and making it close to the standard Gaussian prior with the KL divergence seems to have no supervisory information. Why can this minimize mutual information and filter out irrelevant information?
>
>
> Our implementation follows the standard Variational Information Bottleneck framework, where the absence of supervisory signals in the KL divergence term is the intended design of the Compactness objective $\mathcal{L}\_\text{Comp}$. By minimizing the KL divergence $D_{KL}(p\_{\theta}(z|x^{o})||q(z))$ between the posterior and a standard Gaussian prior, we force the latent representation $Z$ to approximate an uninformative distribution. This constraint explicitly penalizes $Z$ for retaining excessive information about the input $X^\text{o}$, thereby filtering out non-essential components such as noise introduced by missing values.
>
> The necessary supervision is provided solely by the Prediction objective ($\mathcal{L}\_\text{Pred}$), which ensures that only task-relevant information is preserved. This trade-off between $\mathcal{L}\_\text{Comp}， \mathcal{L}\_\text{Pred}$ effectively minimizes mutual information while maximizing predictive performance.
>
>
>
> > **W4:** Some parts of the paper are vaguely described. W/o IB means removing the compactness and informativeness guidance of IB. The former is Eq. 9, but what is the latter? The two are mixed in the ablation study, and the effect of removing Eq. 9 alone is not shown.
> >
> > **W7:** The experiments in the paper are not solid. The main experiments are only conducted on four datasets, and the ablation studies are only performed on one dataset.
>
>
> We clarify that the informativeness guidance corresponds to the prediction objective defined in Eq. 10. In the original "W/o IB" setting, we removed the entire Information Bottleneck framework, which simultaneously excludes both the compactness (Eq. 9) and informativeness (Eq. 10) terms.
>
> To address the concern regarding the specific impact of Eq. 9, we conduct additional ablation studies to decouple these components. We evaluate the individual contributions of each loss function across four datasets, and the detailed results are presented below.
>
> **The ablation analysis across four datasets confirms the necessity of each component, as removing the Consistency, Regularization, or Prediction objectives consistently leads to performance degradation, validating their collective role in ensuring robust and accurate forecasting.**
>
>
> |ETTh1|0.2|0.2|0.4|0.4|0.6|0.6|0.7|0.7|
> |--|:--:|:--:|:--:|:--:|:--:|:--:|:--:|:--:|
> ||MAE|MSE|MAE|MSE|MAE|MSE|MAE|MSE|
> |w/oConsis|0.237|0.206|0.274|0.270|0.338|0.413|0.402|0.584|
> |w/oReg|0.236|0.206|0.274|0.270|0.339|0.410|0.400|0.579|
> |w/oPred|0.432|0.559|0.505|0.698|0.655|1.066|0.796|1.486|
> |Entire|0.220|0.171|0.251|0.249|0.267|0.296|0.288|0.361|
>
> |Elec|0.2|0.2|0.4|0.4|0.6|0.6|0.7|0.7|
> |--|:--:|:--:|:--:|:--:|:--:|:--:|:--:|:--:|
> ||MAE|MSE|MAE|MSE|MAE|MSE|MAE|MSE|
> |w/oConsis|0.0201|0.0196|0.0267|0.0338|0.0346|0.0569|0.0415|0.0931|
> |w/oReg|0.0186|0.0175|0.0244|0.0286|0.0322|0.0544|0.0399|0.0980|
> |w/oPred|0.0881|0.5454|0.1243|0.9424|0.1818|1.8995|0.2283|2.9058|
> |Entire|0.0150|0.0120|0.0230|0.0280|0.0300|0.0470|0.0380|0.0910|
>
> |Metr|0.2|0.2|0.4|0.4|0.6|0.6|0.7|0.7|
> |--|:--:|:--:|:--:|:--:|:--:|:--:|:--:|:--:|
> ||MAE|MSE|MAE|MSE|MAE|MSE|MAE|MSE|
> |w/oConsis|0.271|0.332|0.257|0.275|0.271|0.306|0.314|0.366|
> |w/oReg|0.253|0.307|0.251|0.275|0.266|0.307|0.311|0.364|
> |w/oPred|0.442|0.418|0.624|0.529|0.985|1.177|1.276|1.959|
> |Entire|0.248|0.271|0.249|0.272|0.265|0.305|0.309|0.356|
>
> |PEMS|0.2|0.2|0.4|0.4|0.6|0.6|0.7|0.7|
> |--|:--:|:--:|:--:|:--:|:--:|:--:|:--:|:--:|
> ||MAE|MSE|MAE|MSE|MAE|MSE|MAE|MSE|
> |w/oConsis|0.0959|0.0470|0.0882|0.0389|0.0976|0.0465|0.1296|0.0596|
> |w/oReg|0.0942|0.0453|0.0869|0.0373|0.0964|0.0456|0.1190|0.0591|
> |w/oPred|0.4006|0.2405|0.5618|0.3631|1.0832|1.2727|2.1480|4.8111|
> |Entire|0.0830|0.0340|0.0850|0.0350|0.0930|0.0430|0.1100|0.0580|

---

> ### Author Response · Authors · 2025-11-23
> **Rebuttal by Authors**
>
> > **W5:** The paper lacks key baselines. For example, baselines for handling irregular and missing data, such as Neural ODE or CDE.
> >
> > **W6:** The paper flattens all variable patches to calculate attention, which is not a very efficient or reasonable approach. When the number of variables is large, e.g., a traffic dataset, this method will become infeasible due to the overly long patch sequence.
> >
> > **W7:** The experiments in the paper are not solid. The main experiments are only conducted on four datasets, and the ablation studies are only performed on one dataset.
>
>
> To address concerns regarding baselines and scalability, we have expanded our experimental evaluation. We now include NeuralCDE as a baseline, and we have large-scale traffic datasets (PEMS and Metr) to demonstrate robustness in high-dimensional settings. Regarding the efficiency of the attention mechanism, we clarify that our patching operation is applied strictly along the temporal dimension. Consequently, the sequence length is determined by the time window and does not scale prohibitively with the number of variables. This design ensures that the computational complexity remains manageable even for datasets with a large number of variables. The updated experimental results are presented below. **Extensive experiments on these datasets confirm that CRIB consistently achieves state-of-the-art performance, exhibiting superior robustness across varying missing rates compared to both direct prediction and imputation-based baselines.**
>
> |DatasetsInformation|ETTh1|ETTh2|ETTm1|ETTm2|Electricity|PEMS-BAY|Metr-LA|BeijingAir|Weather|Exchange|
> |--|:--:|:--:|:--:|:--:|:--:|:--:|:--:|:--:|:--:|:--:|
> |TimeSteps|17420|17420|69680|69680|26304|52116|34272|36000|52696|7588|
> |Variates|7|7|7|7|321|325|207|7|21|8|
>
>
> |ETTh1|0.0|0.0|0.2|0.2|0.4|0.4|0.6|0.6|0.7|0.7|
> |--|--|--|--|--|--|--|--|--|--|--|
> ||MAE|MSE|MAE|MSE|MAE|MSE|MAE|MSE|MAE|MSE|
> |CRIB|**0.195**|**0.163**|**0.238**|**0.211**|**0.274**|**0.273**|**0.342**|**0.427**|**0.411**|**0.624**|
> |SegRNN|0.209|0.167|0.247|0.215|0.297|0.307|0.380|0.506|0.455|0.737|
> |WPMixer|0.201|0.165|0.243|0.217|0.290|0.302|0.376|0.504|0.458|0.755|
> |iTransformer|0.211|0.172|0.250|0.219|0.338|0.316|0.385|0.494|0.453|0.731|
> |PatchTST|0.206|0.166|0.245|0.219|0.306|0.292|0.367|0.473|0.431|0.693|
> |DLinear|0.204|0.170|0.270|0.245|0.331|0.361|0.443|0.629|0.544|0.927|
> |PAttn|0.199|0.164|0.249|0.227|0.329|0.309|0.375|0.487|0.437|0.718|
> |CSDI|0.217|0.203|0.305|0.325|0.404|0.503|0.567|0.880|0.717|1.311|
> |NeuralCDE|0.291|0.236|0.340|0.315|0.408|0.460|0.522|0.761|0.629|1.101|
> |ImputeFormer|0.213|0.182|0.260|0.265|0.312|0.369|0.399|0.630|0.488|0.938|
> |TimesNet|0.225|0.178|0.256|0.234|0.297|0.302|0.373|0.457|0.443|0.627|
>
>
> |ETTh2|0.0|0.0|0.2|0.2|0.4|0.4|0.6|0.6|0.7|0.7|
> |--|--|--|--|--|--|--|--|--|--|--|
> ||MAE|MSE|MAE|MSE|MAE|MSE|MAE|MSE|MAE|MSE|
> |CRIB|**0.094**|**0.023**|**0.107**|**0.028**|**0.131**|**0.039**|**0.154**|**0.055**|**0.172**|**0.069**|
> |SegRNN|0.096|0.024|0.115|0.031|0.148|0.047|0.183|0.072|0.205|0.093|
> |WPMixer|0.095|0.024|0.116|0.030|0.147|0.045|0.202|0.086|0.218|0.104|
> |iTransformer|0.098|0.025|0.115|0.033|0.298|0.180|0.270|0.170|0.218|0.111|
> |PatchTST|0.096|0.024|0.109|0.030|0.215|0.125|0.216|0.121|0.186|0.083|
> |DLinear|0.098|0.025|0.142|0.040|0.181|0.065|0.253|0.135|0.325|0.228|
> |PAttn|0.096|0.025|0.116|0.033|0.289|0.173|0.243|0.133|0.196|0.085|
> |CSDI|0.113|0.033|0.255|0.192|0.404|0.399|0.618|0.840|0.798|1.303|
> |NeuralCDE|0.197|0.074|0.244|0.105|0.276|0.134|0.347|0.210|0.413|0.303|
> |ImputeFormer|0.101|0.026|0.122|0.034|0.140|0.049|0.173|0.061|0.174|0.071|
> |TimesNet|0.097|0.025|0.121|0.033|0.153|0.048|0.209|0.092|0.246|0.132|
>
>
> |Weather|0.0|0.0|0.2|0.2|0.4|0.4|0.6|0.6|0.7|0.7|
> |--|--|--|--|--|--|--|--|--|--|--|
> ||MAE|MSE|MAE|MSE|MAE|MSE|MAE|MSE|MAE|MSE|
> |CRIB|**0.028**|**0.016**|**0.038**|**0.025**|**0.050**|**0.033**|**0.062**|**0.057**|**0.070**|**0.075**|
> |SegRNN|0.031|0.021|0.042|0.028|0.050|0.037|0.066|0.061|0.076|0.081|
> |WPMixer|0.030|0.019|0.050|0.025|0.074|0.036|0.102|0.062|0.121|0.084|
> |iTransformer|0.030|0.018|0.044|0.031|0.152|0.167|0.175|0.223|0.101|0.138|
> |PatchTST|0.031|0.019|0.048|0.027|0.150|0.156|0.169|0.169|0.118|0.128|
> |DLinear|0.034|0.023|0.099|0.045|0.135|0.078|0.197|0.178|0.254|0.318|
> |PAttn|0.031|0.020|0.046|0.032|0.157|0.140|0.168|0.163|0.100|0.120|
> |CSDI|0.051|0.042|0.173|0.220|0.300|0.500|0.466|1.099|0.586|1.707|
> |NeuralCDE|0.051|0.029|0.073|0.036|0.095|0.060|0.150|0.150|0.204|0.284|
> |ImputeFormer|0.035|0.020|0.041|0.028|0.051|0.033|0.066|0.064|0.076|0.083|
> |TimesNet|0.028|0.016|0.062|0.033|0.088|0.047|0.128|0.089|0.166|0.154|

---

> ### Author Response · Authors · 2025-11-23
> **Rebuttal by Authors**
>
> |Exchange|0.0|0.0|0.2|0.2|0.4|0.4|0.6|0.6|0.7|0.7|
> |--|--|--|--|--|--|--|--|--|--|--|
> ||MAE|MSE|MAE|MSE|MAE|MSE|MAE|MSE|MAE|MSE|
> |CRIB|**0.0184**|**0.0009**|**0.0217**|**0.0016**|**0.0253**|**0.0015**|**0.0363**|**0.0031**|**0.0517**|**0.0058**|
> |SegRNN|0.0185|0.0010|0.0292|0.0031|0.0629|0.0129|0.1165|0.0363|0.1418|0.0490|
> |WPMixer|0.0187|0.0010|0.0324|0.0024|0.0783|0.0127|0.1364|0.0405|0.1836|0.0697|
> |iTransformer|0.0190|0.0010|0.0288|0.0024|0.1899|0.0770|0.2439|0.1060|0.2865|0.1413|
> |PatchTST|0.0186|0.0010|0.0273|0.0019|0.1416|0.0884|0.1978|0.1129|0.1321|0.0581|
> |DLinear|0.0205|0.0011|0.0795|0.0126|0.1469|0.0429|0.2599|0.1394|0.3778|0.3011|
> |PAttn|0.0187|0.0010|0.0227|0.0016|0.1408|0.1047|0.1283|0.0831|0.0718|0.0169|
> |CSDI|0.0264|0.0017|0.1992|0.2089|0.4358|0.5718|0.7911|1.3714|1.0766|2.2850|
> |NeuralCDE|0.3223|0.1724|0.2969|0.1592|0.3326|0.2146|0.4334|0.3814|0.5412|0.5912|
> |ImputeFormer|0.0344|0.0031|0.0364|0.0029|0.0431|0.0034|0.0608|0.0071|0.0846|0.0128|
> |TimesNet|0.0195|0.0011|0.0458|0.0041|0.0786|0.0130|0.1357|0.0404|0.1991|0.0881|
>
>
> |ETTm1|0.0|0.0|0.2|0.2|0.4|0.4|0.6|0.6|0.7|0.7|
> |--|--|--|--|--|--|--|--|--|--|--|
> ||MAE|MSE|MAE|MSE|MAE|MSE|MAE|MSE|MAE|MSE|
> |CRIB|**0.2000**|**0.2008**|**0.2368**|**0.2586**|**0.2734**|**0.3382**|**0.3414**|**0.5200**|**0.4013**|**0.6956**|
> |SegRNN|0.2265|0.2302|0.2616|0.3068|0.3005|0.4007|0.3678|0.5960|0.4266|0.8006|
> |WPMixer|0.2405|0.2741|0.2900|0.3593|0.3308|0.4645|0.4171|0.7328|0.4863|0.9928|
> |iTransformer|0.2309|0.2514|0.2909|0.3751|0.3593|0.4851|0.4365|0.7580|0.5034|1.0663|
> |PatchTST|0.2379|0.2648|0.2779|0.3486|0.3456|0.4671|0.4091|0.6769|0.4743|0.9325|
> |DLinear|0.2772|0.3504|0.3711|0.5499|0.4418|0.7412|0.5551|1.1067|0.6508|1.4721|
> |PAttn|0.2423|0.2782|0.2971|0.3864|0.3592|0.4801|0.4191|0.7022|0.4816|0.9837|
> |CSDI|0.4759|0.8854|0.5198|0.9345|0.5787|1.0499|0.7038|1.3801|0.8287|1.7733|
> |NeuralCDE|0.3331|0.4218|0.3809|0.5177|0.4391|0.6643|0.5457|0.9573|0.6278|1.2710|
> |ImputeFormer|0.2999|0.2682|0.3271|0.3282|0.3571|0.4749|0.4315|0.6978|0.4845|0.9035|
> |TimesNet|0.2100|0.2018|0.2584|0.2992|0.3040|0.4093|0.3993|0.6525|0.4828|0.8988|
>
>
> |ETTm2|0.0|0.0|0.2|0.2|0.4|0.4|0.6|0.6|0.7|0.7|
> |--|--|--|--|--|--|--|--|--|--|--|
> ||MAE|MSE|MAE|MSE|MAE|MSE|MAE|MSE|MAE|MSE|
> |CRIB|**0.0746**|**0.0152**|**0.0885**|**0.0203**|**0.1037**|**0.0258**|**0.1266**|**0.0384**|**0.1467**|**0.0520**|
> |SegRNN|0.0802|0.0175|0.0956|0.0231|0.1159|0.0305|0.1432|0.0477|0.1650|0.0619|
> |WPMixer|0.0821|0.0182|0.1059|0.0266|0.1355|0.0394|0.1790|0.0685|0.1899|0.0808|
> |iTransformer|0.0781|0.0161|0.1020|0.0284|0.2673|0.1688|0.2651|0.1671|0.1905|0.0861|
> |PatchTST|0.0824|0.0182|0.1005|0.0251|0.2041|0.1210|0.2425|0.1423|0.1803|0.0779|
> |DLinear|0.0918|0.0217|0.1405|0.0391|0.1743|0.0601|0.2412|0.1215|0.3104|0.2056|
> |PAttn|0.0828|0.0187|0.1024|0.0278|0.2441|0.1490|0.2398|0.1448|0.1728|0.0717|
> |CSDI|0.1313|0.0410|0.2687|0.1976|0.4131|0.4003|0.6221|0.8293|0.8024|1.2968|
> |NeuralCDE|0.1327|0.0363|0.1760|0.0591|0.1984|0.0765|0.2479|0.1194|0.3130|0.1857|
> |ImputeFormer|0.0813|0.0160|0.0888|0.0222|0.1088|0.0267|0.1360|0.0424|0.1602|0.0569|
> |TimesNet|0.0782|0.0165|0.1087|0.0260|0.1417|0.0406|0.1986|0.0811|0.2317|0.1142|
>
> |BejingAir|0.0|0.0|0.2|0.2|0.4|0.4|0.6|0.6|0.7|0.7|
> |--|--|--|--|--|--|--|--|--|--|--|
> ||MAE|MSE|MAE|MSE|MAE|MSE|MAE|MSE|MAE|MSE|
> |CRIB|**0.2526**|**0.2929**|**0.2758**|**0.3317**|**0.3106**|**0.4120**|**0.3728**|**0.5862**|**0.4337**|0.8010|
> |SegRNN|0.2552|0.3026|0.2849|0.3467|0.3188|0.4285|0.3860|0.5939|0.4412|**0.7595**|
> |WPMixer|0.2571|0.3085|0.2896|0.3651|0.3312|0.4601|0.4011|0.6503|0.4632|0.8730|
> |iTransformer|0.2606|0.3156|0.2923|0.3655|0.3545|0.4820|0.4124|0.6573|0.4691|0.8874|
> |PatchTST|0.2609|0.3008|0.2866|0.3421|0.3278|0.4305|0.3976|0.6458|0.4613|0.8540|
> |DLinear|0.2707|0.3263|0.3204|0.3853|0.3620|0.4760|0.4362|0.6799|0.5022|0.9358|
> |PAttn|0.2601|0.3102|0.2949|0.3603|0.3480|0.4689|0.4142|0.6624|0.4673|0.8745|
> |CSDI|0.3431|0.5141|0.4035|0.6010|0.4714|0.7584|0.5930|1.1378|0.7057|1.5761|
> |NeuralCDE|0.3167|0.3779|0.3478|0.4196|0.3950|0.5474|0.4750|0.7803|0.5456|1.0151|
> |ImputeFormer|0.2533|0.3189|0.2770|0.3496|0.3297|0.4645|0.4006|0.6110|0.4549|0.8327|
> |TimesNet|0.2583|0.2939|0.2922|0.3488|0.3322|0.4422|0.4017|0.6275|0.4789|0.9234|

---

### Official Review · Reviewer_g59r · 2025-11-01

**Soundness:** 2
**Presentation:** 2
**Contribution:** 2
**Rating:** 4
**Confidence:** 2

**Summary:**

This paper focuses on the critical task of Multivariate Time Series Forecasting with Missing Values (MTSF-M), where missing values are prevalent in real-world scenarios. It first points out the flaw of the current mainstream "imputation-then-prediction" framework: since there is no ground truth for missing values, unsupervised imputation is prone to errors, which corrupts the underlying data distribution and degrades forecasting accuracy—a conclusion validated by the authors’ systematic empirical study. To address this issue, the paper proposes a paradigm shift: abandoning independent imputation and directly forecasting from partially observed time series.

**Strengths:**

The paper identifies a critical limitation of the mainstream "imputation-then-prediction" framework in MTSF-M—unsupervised imputation’s lack of ground truth leading to data distribution corruption and forecasting accuracy degradation—and validates this via systematic empirical studies, which provides a clear and necessary critique of existing methods. Additionally, the proposed paradigm shift, direct forecasting from partially observed data, and the CRIB framework  demonstrate theoretical innovation.

**Weaknesses:**

The paper’s central claim—shifting from “imputation-then-prediction” to direct forecasting via an IB-based framework—is conceptually relevant but lacks sufficient novelty to meet ICLR standards. The critique of imputation-induced distribution corruption is not fully original. Prior works have already questioned the imputation paradigm and proposed direct prediction methods, yet these are not adequately discussed or contrasted. The proposed “CRIB” framework combines two existing components—Information Bottleneck (IB) and consistency regularization—without demonstrating a non-trivial, innovative integration. The paper fails to explain how CRIB’s design solves unique challenges that prior IB-based or consistency-regularized models cannot address. The paper only compares against “imputation-then-prediction” methods but omits state-of-the-art direct forecasting baselines This one-sided comparison overstates CRIB’s performance.

**Questions:**

The paper only claims CRIB is effective but does not compare it to SOTA MTSF-M models. Could you add a head-to-head comparison with recent methods?
The paper references "four real-world datasets" but provides no context to assess generalizability. Please list the dataset names, domains, time series length, and number of variables for each dataset?

---

> ### Author Response · Authors · 2025-11-23
> **Rebuttal by Authors**
>
> We sincerely thank Reviewer g59r for the thoughtful and constructive reviews of our manuscript! Based on your questions and recommendations, we give point-by-point responses to your comments and describe the revisions we made to address them.
>
> ---
>
> > **W1:** The paper’s central claim—shifting from “imputation-then-prediction” to direct forecasting via an IB-based framework—is conceptually relevant but lacks sufficient novelty to meet ICLR standards. The critique of imputation-induced distribution corruption is not fully original.
> >
> > **W3:** The proposed “CRIB” framework combines two existing components—Information Bottleneck (IB) and consistency regularization—without demonstrating a non-trivial, innovative integration. The paper fails to explain how CRIB’s design solves unique challenges that prior IB-based or consistency-regularized models cannot address.
>
>
> Thanks for your comments. We acknowledge that the concept of direct prediction and IB-based methods exists. **However, our primary contribution is a systematic re-evaluation of the dominant "imputation-then-prediction" paradigm and the proposal of CRIB to address its fundamental limitations.**
>
> 1. **Critique of Imputation.** We empirically demonstrate that imputation without ground truth is detrimental. As visualized in Figure 1, unsupervised imputation fails to recover the true data distribution and merely reinforces the biased patterns of partial observations.
> 2. **Evidence of Degradation.** We present counter-intuitive evidence that explicit imputation actively degrades prediction accuracy. For example, equipping the predictor **TimeXer** with the SOTA imputer **TimesNet** worsens the MAE from **0.125** to **0.135**.
> 3. **Limitations of Naive IB.** We also identify that a naive application of the Information Bottleneck is insufficient. Figure 6 (Appendix F.2) shows that a direct IB-based Transformer yields degenerate attention maps (collapsing into local straight lines) and neglects global dependencies. Furthermore, results in Table 2 (Section 4.3) indicate that CRIB without the consistency loss exhibits significantly higher variance.
> 4. **The CRIB Solution.** To address these issues, CRIB adopts a direct prediction paradigm. By integrating IB with consistency regularization and Unified-Attention, CRIB effectively filters the noise inherent in missing values, thereby bypassing the error propagation caused by erroneous imputation.
> 5. We also add the qualitative comparison between the **average** final prediction performance on missing rate of 20%, 40%, 60% and 70% of our method CRIB, BiTGraph, DLinear, TimesNet combined with DLinear, TimeXer, and TimesNet combined with TimeXer.
>
> | PEMS-BAY |   CRIB    | BiTGraph | DLinear | TimesNet + DLinear | TimeXer | TimesNet + TimeXer |
> | :------- | :-------: | :------: | :-----: | :----------------: | :-----: | :----------------: |
> | MAE      | **0.093** |  0.413   |  0.156  |       0.148        | `0.125` |       0.135        |
> | MSE      | **0.043** |  0.788   |  0.087  |       0.081        | `0.051` |       0.073        |

---

> ### Author Response · Authors · 2025-11-23
> **Rebuttal by Authors**
>
> > **W2:** Prior works have already questioned the imputation paradigm and proposed direct prediction methods, yet these are not adequately discussed or contrasted.
> >
> > **W4:** The paper only compares against “imputation-then-prediction” methods but omits state-of-the-art direct forecasting baselines. This one-sided comparison overstates CRIB’s performance.
> >
> > **Q1:** The paper only claims CRIB is effective but does not compare it to SOTA MTSF-M models. Could you add a head-to-head comparison with recent methods?
> >
> > **Q2:** The paper references "four real-world datasets" but provides no context to assess generalizability. Please list the dataset names, domains, time series length, and number of variables for each dataset?
>
>
> We discuss existing MTSF-M methods in the Related Work section. These include previous two-stage methods and recent end-to-end models like BiTGraph and GinAR. **Since end-to-end models perform imputation in the embedding space, we classify them into the "imputation-then-prediction" framework rather than direct prediction.**
>
> Moreover, we point out a crucial limitation that there is no ground truth for missing values in the real world. Our Figure 1 empirically demonstrates that unsupervised imputation distorts data distribution and variate correlations.
>
> Furthermore, our baselines cover state-of-the-art forecasting models, including Transformer-based (PatchTST, iTransformer), MLP-based (DLinear, WPMixer), and RNN-based (SegRNN) architectures. We also include BiTGraph for comparison.
>
> Our baselines are new, including ImputeFormer (2024 KDD), BiTGraph (ICLR 2024), SAITS (ESWA 2023), SegRNN(Arxiv 2023), WPMixer (AAAI 2025), iTransformer (ICLR 2024), PatchTST (ICLR 2023), DLinear (AAAI 2023), TimeXer (NeurIPS 2024), and PAttn (NeurIPS 2024). We exclude GinAR and S4M due to the unavailability of the complete code.
>
> During the rebuttal, we have done additional forecasting results and provided them below (More datasets and baselines). **Extensive experiments on these datasets confirm that CRIB consistently achieves state-of-the-art performance, exhibiting superior robustness across varying missing rates compared to both direct prediction and imputation-based baselines.**
>
> |DatasetsInformation|ETTh1|ETTh2|ETTm1|ETTm2|Electricity|PEMS-BAY|Metr-LA|BeijingAir|Weather|Exchange|
> |-----|:---:|:---:|:---:|:---:|:---------:|:------:|:-----:|:---------:|:-----:|:------:|
> |TimeSteps|17420|17420|69680|69680|26304|52116|34272|36000|52696|7588|
> |Variates|7|7|7|7|321|325|207|7|21|8|
>
> |ETTh1|0.0|0.0|0.2|0.2|0.4|0.4|0.6|0.6|0.7|0.7|
> |--|--|--|--|--|--|--|--|--|--|--|
> ||MAE|MSE|MAE|MSE|MAE|MSE|MAE|MSE|MAE|MSE|
> |CRIB|**0.195**|**0.163**|**0.238**|**0.211**|**0.274**|**0.273**|**0.342**|**0.427**|**0.411**|**0.624**|
> |SegRNN|0.209|0.167|0.247|0.215|0.297|0.307|0.380|0.506|0.455|0.737|
> |WPMixer|0.201|0.165|0.243|0.217|0.290|0.302|0.376|0.504|0.458|0.755|
> |iTransformer|0.211|0.172|0.250|0.219|0.338|0.316|0.385|0.494|0.453|0.731|
> |PatchTST|0.206|0.166|0.245|0.219|0.306|0.292|0.367|0.473|0.431|0.693|
> |DLinear|0.204|0.170|0.270|0.245|0.331|0.361|0.443|0.629|0.544|0.927|
> |PAttn|0.199|0.164|0.249|0.227|0.329|0.309|0.375|0.487|0.437|0.718|
> |CSDI|0.217|0.203|0.305|0.325|0.404|0.503|0.567|0.880|0.717|1.311|
> |NeuralCDE|0.291|0.236|0.340|0.315|0.408|0.460|0.522|0.761|0.629|1.101|
> |ImputeFormer|0.213|0.182|0.260|0.265|0.312|0.369|0.399|0.630|0.488|0.938|
> |TimesNet|0.225|0.178|0.256|0.234|0.297|0.302|0.373|0.457|0.443|0.627|
>
>
> |ETTh2|0.0|0.0|0.2|0.2|0.4|0.4|0.6|0.6|0.7|0.7|
> |--|--|--|--|--|--|--|--|--|--|--|
> ||MAE|MSE|MAE|MSE|MAE|MSE|MAE|MSE|MAE|MSE|
> |CRIB|**0.094**|**0.023**|**0.107**|**0.028**|**0.131**|**0.039**|**0.154**|**0.055**|**0.172**|**0.069**|
> |SegRNN|0.096|0.024|0.115|0.031|0.148|0.047|0.183|0.072|0.205|0.093|
> |WPMixer|0.095|0.024|0.116|0.030|0.147|0.045|0.202|0.086|0.218|0.104|
> |iTransformer|0.098|0.025|0.115|0.033|0.298|0.180|0.270|0.170|0.218|0.111|
> |PatchTST|0.096|0.024|0.109|0.030|0.215|0.125|0.216|0.121|0.186|0.083|
> |DLinear|0.098|0.025|0.142|0.040|0.181|0.065|0.253|0.135|0.325|0.228|
> |PAttn|0.096|0.025|0.116|0.033|0.289|0.173|0.243|0.133|0.196|0.085|
> |CSDI|0.113|0.033|0.255|0.192|0.404|0.399|0.618|0.840|0.798|1.303|
> |NeuralCDE|0.197|0.074|0.244|0.105|0.276|0.134|0.347|0.210|0.413|0.303|
> |ImputeFormer|0.101|0.026|0.122|0.034|0.140|0.049|0.173|0.061|0.174|0.071|
> |TimesNet|0.097|0.025|0.121|0.033|0.153|0.048|0.209|0.092|0.246|0.132|

---

> ### Author Response · Authors · 2025-11-23
> **Rebuttal by Authors**
>
> |Weather|0.0|0.0|0.2|0.2|0.4|0.4|0.6|0.6|0.7|0.7|
> |--|--|--|--|--|--|--|--|--|--|--|
> ||MAE|MSE|MAE|MSE|MAE|MSE|MAE|MSE|MAE|MSE|
> |CRIB|**0.028**|**0.016**|**0.038**|**0.025**|**0.050**|**0.033**|**0.062**|**0.057**|**0.070**|**0.075**|
> |SegRNN|0.031|0.021|0.042|0.028|0.050|0.037|0.066|0.061|0.076|0.081|
> |WPMixer|0.030|0.019|0.050|0.025|0.074|0.036|0.102|0.062|0.121|0.084|
> |iTransformer|0.030|0.018|0.044|0.031|0.152|0.167|0.175|0.223|0.101|0.138|
> |PatchTST|0.031|0.019|0.048|0.027|0.150|0.156|0.169|0.169|0.118|0.128|
> |DLinear|0.034|0.023|0.099|0.045|0.135|0.078|0.197|0.178|0.254|0.318|
> |PAttn|0.031|0.020|0.046|0.032|0.157|0.140|0.168|0.163|0.100|0.120|
> |CSDI|0.051|0.042|0.173|0.220|0.300|0.500|0.466|1.099|0.586|1.707|
> |NeuralCDE|0.051|0.029|0.073|0.036|0.095|0.060|0.150|0.150|0.204|0.284|
> |ImputeFormer|0.035|0.020|0.041|0.028|0.051|0.033|0.066|0.064|0.076|0.083|
> |TimesNet|0.028|0.016|0.062|0.033|0.088|0.047|0.128|0.089|0.166|0.154|
>
>
>
> |Exchange|0.0|0.0|0.2|0.2|0.4|0.4|0.6|0.6|0.7|0.7|
> |--|--|--|--|--|--|--|--|--|--|--|
> ||MAE|MSE|MAE|MSE|MAE|MSE|MAE|MSE|MAE|MSE|
> |CRIB|**0.0184**|**0.0009**|**0.0217**|**0.0016**|**0.0253**|**0.0015**|**0.0363**|**0.0031**|**0.0517**|**0.0058**|
> |SegRNN|0.0185|0.0010|0.0292|0.0031|0.0629|0.0129|0.1165|0.0363|0.1418|0.0490|
> |WPMixer|0.0187|0.0010|0.0324|0.0024|0.0783|0.0127|0.1364|0.0405|0.1836|0.0697|
> |iTransformer|0.0190|0.0010|0.0288|0.0024|0.1899|0.0770|0.2439|0.1060|0.2865|0.1413|
> |PatchTST|0.0186|0.0010|0.0273|0.0019|0.1416|0.0884|0.1978|0.1129|0.1321|0.0581|
> |DLinear|0.0205|0.0011|0.0795|0.0126|0.1469|0.0429|0.2599|0.1394|0.3778|0.3011|
> |PAttn|0.0187|0.0010|0.0227|0.0016|0.1408|0.1047|0.1283|0.0831|0.0718|0.0169|
> |CSDI|0.0264|0.0017|0.1992|0.2089|0.4358|0.5718|0.7911|1.3714|1.0766|2.2850|
> |NeuralCDE|0.3223|0.1724|0.2969|0.1592|0.3326|0.2146|0.4334|0.3814|0.5412|0.5912|
> |ImputeFormer|0.0344|0.0031|0.0364|0.0029|0.0431|0.0034|0.0608|0.0071|0.0846|0.0128|
> |TimesNet|0.0195|0.0011|0.0458|0.0041|0.0786|0.0130|0.1357|0.0404|0.1991|0.0881|
>
>
> |ETTm1|0.0|0.0|0.2|0.2|0.4|0.4|0.6|0.6|0.7|0.7|
> |--|--|--|--|--|--|--|--|--|--|--|
> ||MAE|MSE|MAE|MSE|MAE|MSE|MAE|MSE|MAE|MSE|
> |CRIB|**0.2000**|**0.2008**|**0.2368**|**0.2586**|**0.2734**|**0.3382**|**0.3414**|**0.5200**|**0.4013**|**0.6956**|
> |SegRNN|0.2265|0.2302|0.2616|0.3068|0.3005|0.4007|0.3678|0.5960|0.4266|0.8006|
> |WPMixer|0.2405|0.2741|0.2900|0.3593|0.3308|0.4645|0.4171|0.7328|0.4863|0.9928|
> |iTransformer|0.2309|0.2514|0.2909|0.3751|0.3593|0.4851|0.4365|0.7580|0.5034|1.0663|
> |PatchTST|0.2379|0.2648|0.2779|0.3486|0.3456|0.4671|0.4091|0.6769|0.4743|0.9325|
> |DLinear|0.2772|0.3504|0.3711|0.5499|0.4418|0.7412|0.5551|1.1067|0.6508|1.4721|
> |PAttn|0.2423|0.2782|0.2971|0.3864|0.3592|0.4801|0.4191|0.7022|0.4816|0.9837|
> |CSDI|0.4759|0.8854|0.5198|0.9345|0.5787|1.0499|0.7038|1.3801|0.8287|1.7733|
> |NeuralCDE|0.3331|0.4218|0.3809|0.5177|0.4391|0.6643|0.5457|0.9573|0.6278|1.2710|
> |ImputeFormer|0.2999|0.2682|0.3271|0.3282|0.3571|0.4749|0.4315|0.6978|0.4845|0.9035|
> |TimesNet|0.2100|0.2018|0.2584|0.2992|0.3040|0.4093|0.3993|0.6525|0.4828|0.8988|

---

> ### Author Response · Authors · 2025-11-24
> **Rebuttal by Author**
>
> |ETTm2|0.0|0.0|0.2|0.2|0.4|0.4|0.6|0.6|0.7|0.7|
> |--|--|--|--|--|--|--|--|--|--|--|
> ||MAE|MSE|MAE|MSE|MAE|MSE|MAE|MSE|MAE|MSE|
> |CRIB|**0.0746**|**0.0152**|**0.0885**|**0.0203**|**0.1037**|**0.0258**|**0.1266**|**0.0384**|**0.1467**|**0.0520**|
> |SegRNN|0.0802|0.0175|0.0956|0.0231|0.1159|0.0305|0.1432|0.0477|0.1650|0.0619|
> |WPMixer|0.0821|0.0182|0.1059|0.0266|0.1355|0.0394|0.1790|0.0685|0.1899|0.0808|
> |iTransformer|0.0781|0.0161|0.1020|0.0284|0.2673|0.1688|0.2651|0.1671|0.1905|0.0861|
> |PatchTST|0.0824|0.0182|0.1005|0.0251|0.2041|0.1210|0.2425|0.1423|0.1803|0.0779|
> |DLinear|0.0918|0.0217|0.1405|0.0391|0.1743|0.0601|0.2412|0.1215|0.3104|0.2056|
> |PAttn|0.0828|0.0187|0.1024|0.0278|0.2441|0.1490|0.2398|0.1448|0.1728|0.0717|
> |CSDI|0.1313|0.0410|0.2687|0.1976|0.4131|0.4003|0.6221|0.8293|0.8024|1.2968|
> |NeuralCDE|0.1327|0.0363|0.1760|0.0591|0.1984|0.0765|0.2479|0.1194|0.3130|0.1857|
> |ImputeFormer|0.0813|0.0160|0.0888|0.0222|0.1088|0.0267|0.1360|0.0424|0.1602|0.0569|
> |TimesNet|0.0782|0.0165|0.1087|0.0260|0.1417|0.0406|0.1986|0.0811|0.2317|0.1142|
>
> |BejingAir|0.0|0.0|0.2|0.2|0.4|0.4|0.6|0.6|0.7|0.7|
> |--|--|--|--|--|--|--|--|--|--|--|
> ||MAE|MSE|MAE|MSE|MAE|MSE|MAE|MSE|MAE|MSE|
> |CRIB|**0.2526**|**0.2929**|**0.2758**|**0.3317**|**0.3106**|**0.4120**|**0.3728**|**0.5862**|**0.4337**|0.8010|
> |SegRNN|0.2552|0.3026|0.2849|0.3467|0.3188|0.4285|0.3860|0.5939|0.4412|**0.7595**|
> |WPMixer|0.2571|0.3085|0.2896|0.3651|0.3312|0.4601|0.4011|0.6503|0.4632|0.8730|
> |iTransformer|0.2606|0.3156|0.2923|0.3655|0.3545|0.4820|0.4124|0.6573|0.4691|0.8874|
> |PatchTST|0.2609|0.3008|0.2866|0.3421|0.3278|0.4305|0.3976|0.6458|0.4613|0.8540|
> |DLinear|0.2707|0.3263|0.3204|0.3853|0.3620|0.4760|0.4362|0.6799|0.5022|0.9358|
> |PAttn|0.2601|0.3102|0.2949|0.3603|0.3480|0.4689|0.4142|0.6624|0.4673|0.8745|
> |CSDI|0.3431|0.5141|0.4035|0.6010|0.4714|0.7584|0.5930|1.1378|0.7057|1.5761|
> |NeuralCDE|0.3167|0.3779|0.3478|0.4196|0.3950|0.5474|0.4750|0.7803|0.5456|1.0151|
> |ImputeFormer|0.2533|0.3189|0.2770|0.3496|0.3297|0.4645|0.4006|0.6110|0.4549|0.8327|
> |TimesNet|0.2583|0.2939|0.2922|0.3488|0.3322|0.4422|0.4017|0.6275|0.4789|0.9234|

---

### Official Review · Reviewer_bbCN · 2025-11-03

**Soundness:** 2
**Presentation:** 1
**Contribution:** 2
**Rating:** 6
**Confidence:** 3

**Summary:**

This paper introduces a new paradigm for time-series forecasting with missing data based on the Information Bottleneck Principle. They argue that current forecasting with missing data tends to use imputation modules to recover the missing data, then do the forecasting. This is claimed to be flawed because of the lack of ground truth for missing data, which can corrupt the observed data and hurt the performance. As an alternative approach, they (i) propose an architecture that handles the sparse nature of the observed data through patching embedding and unified-variate attention, and (ii) include the consistency regularization loss additionally to the usual reconstruction and compactness losses, along with data augmentation to learn invariance to missing patterns. They show that their scheme provides strong MAE and MSE performance on four datasets against twelve baselines with four missing rates settings.

**Strengths:**

The paper achieves strong performance against the baselines (Table 1), and multiple ablations that show the importance of the different modules (Table 2 and Figure 4), including the Consistency loss and the Unified-Variate Attention. In particular, Figure 4 (a) shows the importance of (i) IB (ii) Consistency loss, and (iii) Uni-Attn, and Figure 4 (b) shows the robustness to hyperparameters. Also, Table 2 shows that CRST-IB can perform significantly better than w/o for well-chosen consistency regularization weights.

**Weaknesses:**

There is too much information in Figure 1: it is hard to read. You should simplify it and put your method on the right. It would also be nice to add more qualitative comparison with the competitors. Although it is said in the Related Work section, it should be made clearer in the paper that Compactness and Informativeness losses already exist and are not part of the novelty. This needs to be clearer both in 3.4.1, 3.4.2, and precise references of previous work in the proofs in Appendix B.

Also, the justification of the consistency loss, e.g with "The core intuition is that the model’s prediction should be invariant to the missingness." is not clear. Can you better explain that?

Issues/typos:

L.42: $\\beta \in \\mathbb{R}^{+}_{*}$

Eq.3: Shouldn't it be $\sin(t/10000^{2m/P})$ instead (and same for cos(...)) ? Otherwise, the embedding doesn’t depend on $m$ when the parity is fixed.

L.277-282:

Sign issue in Eq.10

$\\mathbb{E}\_{p(z, y)}[\log q_\\theta(y \mid z)]= \textcolor{red}{-} \mathbb{E}\_{p(z, y)}[\frac{1}{2 \\sigma^2}\||y-\\hat{y}\||^2+\frac{T}{2} \\log \left(2 \\pi \\sigma^2\right)]$

Figures 3 and Figure 4a: It should be better to have consistency in color for the method (e.g., yellow).

**Questions:**

See weaknesses, also:

What are the main limitations of the method ? Is that the amount of hyperparameters ? How did you choose these ($\\alpha, \\beta, \\gamma$, 10% in the additional random masking, variance of the added Gassian noise) and what are the chosen hyperparams for the Figure 4a ? Limitations should be acknowledged in a separated section.

Are the competitors comparable in terms of computational cost ? Did you retrain the baselines from scratch ?

It would be nice to add the model trained with clean data as well (missing rate = 0) to get a bound on the achievable performance.

---

> ### Author Response · Authors · 2025-11-23
> **Rebuttal by Authors**
>
> We sincerely thank Reviewer bbCN for the thoughtful and constructive reviews of our manuscript! Based on your questions and recommendations, we give point-by-point responses to your comments and describe the revisions we made to address them.
>
> ---
>
> > **W1:** There is too much information in Figure 1: it is hard to read. You should simplify it and put your method on the right. It would also be nice to add more qualitative comparison with the competitors.
>
>
>
> We acknowledge the feedback regarding Figure 1. We will simplify Figure 1 as suggested and place our method on the right for easier comparison.
>
> Figure 1 aims to illustrate the "imputation-then-prediction" paradigm analysis on PEMS-BAY, including a **two-stage method** (combining TimesNet and DLinear) and an **end-to-end method** (BiTGraph). Based on Figure 1, we conclude that ***Improper imputation can corrupt the observed data***, and ***Flawed imputation, in turn, leads to poor prediction performance***.
>
> We also add the qualitative comparison between the **average** final prediction performance on missing rate of 20%, 40%, 60% and 70% of our method CRIB, BiTGraph, DLinear, TimesNet combined with DLinear, TimeXer, and TimesNet combined with TimeXer.
>
> | PEMS-BAY |   CRIB    | BiTGraph | DLinear | TimesNet + DLinear | TimeXer | TimesNet + TimeXer |
> | :------- | :-------: | :------: | :-----: | :----------------: | :-----: | :----------------: |
> | MAE      | **0.093** |  0.413   |  0.156  |       0.148        | `0.125` |       0.135        |
> | MSE      | **0.043** |  0.788   |  0.087  |       0.081        | `0.051` |       0.073        |
>
>
>
>
> >
> > **W2:** Although it is said in the Related Work section, it should be made clearer in the paper that Compactness and Informativeness losses already exist and are not part of the novelty. This needs to be clearer both in 3.4.1, 3.4.2, and precise references of previous work in the proofs in Appendix B.
>
>
> In the next revision, we will explicitly cite the foundational IB works in Section 3.4 and Appendix B to clarify the theoretical origins. **However, our core contribution is not the IB theory itself, but a systematic re-evaluation of the "imputation-then-prediction" paradigm and the proposal of CRIB to resolve its fundamental limitations:**
>
> 1. **Critique of Imputation:** We demonstrate that imputation without ground truth is often detrimental. Our visualization (Figure 1) reveals that imputation methods fail to recover the true data distribution and merely reinforce the biased patterns of partial observations.
> 2. **Evidence of Degradation:** We provide counter-intuitive empirical evidence that imputation actively harms prediction accuracy. For instance, equipping the predictor **TimeXer** with the SOTA imputer **TimesNet** degrades the MAE from **0.125** to **0.135**.
> 3. **Limitations of Vanilla IB:** We further identify that naively applying IB is insufficient. As shown in Fig. 6 (Appendix F.2), a direct IB-based Transformer yields degenerate attention maps (focusing on local straight lines) and neglects global dependencies. Furthermore, results in Table 2 (Section 4.3) indicate that CRIB without the consistency loss exhibits significantly higher variance.
> 4. **The CRIB Solution:** To address these issues, CRIB adopts a "direct prediction" paradigm. By integrating IB with consistency regularization and Unified-Attention, CRIB effectively filters noise inherent in missing values and captures global information, bypassing the error propagation caused by erroneous imputation.
>
>
>
>
> > **W3:** Also, the justification of the consistency loss, e.g with "The core intuition is that the model’s prediction should be invariant to the missingness." is not clear. Can you better explain that?
>
>
> In Section 3.5, we introduce consistency regularization based on the hypothesis that **latent representations should be invariant to missing patterns**. By minimizing the divergence between representations of the original observed data and its augmented counterparts (simulating diverse missingness), we enforce the model to learn robust features that generalize beyond specific observation artifacts.
>
> Crucially, **we identify that a naive application of the Information Bottleneck principle is insufficient**. As visualized in Figure 6 (Appendix F.2), a direct IB-based Transformer produces degenerate attention maps that collapse into local patterns, thereby neglecting essential global dependencies.

---

> ### Author Response · Authors · 2025-11-23
> **Rebuttal by Authors**
>
> > **W4:**
> >
> > Typo
> >
> > 1. Line142: $\beta \in \mathbb{R}^+_*$
> > 2. Eq3: Shouldn't it be $\sin(t/10000^{2m/P})$ instead (and same for cos(...)) ? Otherwise, the embedding doesn’t depend on $m$ when the parity is fixed.
> > 3. Line277-282: $\ge \mathbb{E}\_{p(z,y)}[\log q\_{\theta}(y|z)] =\textcolor{red}{-}\mathbb{E}\_{p(z,y)}\left[ \frac{1}{2\sigma^2} \|y - \widehat{y}\|^2 + \frac{T}{2} \log(2\pi \sigma^2) \right],$
> > 4. Figures 3 and Figure 4a: It should be better to have consistency in color for the method (e.g., yellow).
>
>
> Thank you for your careful review. We correct all typos and formula symbol errors in Line 142, Eq 3, and Lines 277-282 in the new version. Regarding Eq. 3, it should indeed be a function dependent on $m$ to ensure different encodings for different dimensions.
>
>
>
>
> > **Q1:** what are the main limitations of the method ? Is that the amount of hyperparameters ? How did you choose these ($\alpha, \beta, \gamma$, 10% in the additional random masking, variance of the added Gassian noise) and what are the chosen hyperparams for the Figure 4a ? Limitations should be acknowledged in a separated section.
>
>
> **Limitations.** While CRIB remains robust under high missing rates, it is sensitive to the magnitude of regularization. As discussed in Section 4.3 (Obs. 5), excessively high weights for the KL divergence term result in over-regularization, which hinders the model from capturing complex variate correlations.
>
> **Hyperparameter Selection.** We analyze hyperparameter sensitivity in Figure 4(b) and conduct additional experiments. **Empirical results indicate that the optimal settings are consistent across different datasets and missing rates.** In our default configuration, we set the weights for $\mathcal{L}\_\text{comp}$, $\mathcal{L}\_\text{pred}$, and $\mathcal{L}\_\text{consis}$ to $10^{-6}$, $1$, and $1$, respectively.
>
>
>
> | $\mathcal{L}_\text{Pred}$ Weight - 0% Missing Rate | ETTh1 | Exchange | Illness |
> | -------------------------------------------------- | :---: | :------: | :-----: |
> | 0.1                                                | 0.201 | 0.01858  | 0.1495  |
> | 0.5                                                | 0.198 | 0.01858  | 0.1474  |
> | 1                                                  | 0.198 | 0.01859  | 0.1457  |
> | 2                                                  | 0.199 | 0.01859  | 0.1500  |
> | 5                                                  | 0.200 | 0.01862  | 0.1533  |
>
>
> | $\mathcal{L}_\text{Pred}$ Weight - 20% Missing Rate | ETTh1 | Exchange | Illness |
> | --------------------------------------------------- | :---: | :------: | :-----: |
> | 0.1                                                 | 0.243 | 0.03319  | 0.2466  |
> | 0.5                                                 | 0.238 | 0.03005  | 0.2341  |
> | 1                                                   | 0.237 | 0.02873  | 0.2306  |
> | 2                                                   | 0.236 | 0.02614  | 0.2383  |
> | 5                                                   | 0.237 | 0.02457  | 0.2337  |
>
>
> | $\mathcal{L}_\text{Pred}$ Weight - 40% Missing Rate | ETTh1 | Exchange | Illness |
> | --------------------------------------------------- | :---: | :------: | :-----: |
> | 0.1                                                 | 0.281 | 0.04386  | 0.2741  |
> | 0.5                                                 | 0.273 | 0.02943  | 0.2828  |
> | 1                                                   | 0.274 | 0.02528  | 0.2573  |
> | 2                                                   | 0.273 | 0.02614  | 0.2660  |
> | 5                                                   | 0.273 | 0.02577  | 0.2709  |
>
>
> | $\mathcal{L}_\text{Pred}$ Weight - 60% Missing Rate | ETTh1 | Exchange | Illness |
> | --------------------------------------------------- | :---: | :------: | :-----: |
> | 0.1                                                 | 0.348 | 0.04130  | 0.3631  |
> | 0.5                                                 | 0.340 | 0.03621  | 0.3535  |
> | 1                                                   | 0.337 | 0.03634  | 0.3703  |
> | 2                                                   | 0.339 | 0.03634  | 0.3870  |
> | 5                                                   | 0.338 | 0.02874  | 0.3509  |
>
>
> | $\mathcal{L}_\text{Pred}$ Weight - 70% Missing Rate | ETTh1 | Exchange | Illness |
> | --------------------------------------------------- | :---: | :------: | :-----: |
> | 0.1                                                 | 0.412 | 0.03726  | 0.4796  |
> | 0.5                                                 | 0.406 | 0.04227  | 0.4895  |
> | 1                                                   | 0.403 | 0.05166  | 0.4914  |
> | 2                                                   | 0.400 | 0.04751  | 0.4955  |
> | 5                                                   | 0.401 | 0.03199  | 0.4497  |

---

> > ### Author Response · Authors · 2025-11-23
> > **Rebuttal by Authors**
> >
> > | $\mathcal{L}_\text{Comp}$ Weight - 0% Missing Rate | ETTh1 | Exchange | Illness |
> > | ----- | :---: | :------: | :-----: |
> > | 10 | 0.214 | 0.01915  | 0.1713  |
> > | 1  | 0.208 | 0.01876  | 0.1702  |
> > | 1E-02                                              | 0.199 | 0.01861  | 0.1533  |
> > | 1E-06                                              | 0.198 | 0.01859  | 0.1457  |
> > | 1E-10                                              | 0.199 | 0.01859  | 0.1458  |
> >
> > | $\mathcal{L}_\text{Comp}$ Weight - 20% Missing Rate | ETTh1 | Exchange | Illness |
> > | ------ | :---: | :------: | :-----: |
> > | 10  | 0.260 | 0.05088  | 0.2936  |
> > | 1   | 0.251 | 0.04707  | 0.2802  |
> > | 1E-02                                               | 0.238 | 0.02997  | 0.2536  |
> > | 1E-06                                               | 0.237 | 0.02873  | 0.2306  |
> > | 1E-10                                               | 0.237 | 0.02756  | 0.2349  |
> >
> > | $\mathcal{L}_\text{Comp}$ Weight - 40% Missing Rate | ETTh1 | Exchange | Illness |
> > | ------ | :---: | :------: | :-----: |
> > | 10  | 0.314 | 0.08059  | 0.3263  |
> > | 1   | 0.296 | 0.08352  | 0.2942  |
> > | 1E-02                                               | 0.274 | 0.03186  | 0.2679  |
> > | 1E-06                                               | 0.274 | 0.02528  | 0.2573  |
> > | 1E-10                                               | 0.273 | 0.02603  | 0.2583  |
> >
> > | $\mathcal{L}_\text{Comp}$ Weight - 60% Missing Rate | ETTh1 | Exchange | Illness |
> > | ------ | :---: | :------: | :-----: |
> > | 10  | 0.417 | 0.21306  | 0.4830  |
> > | 1   | 0.365 | 0.13925  | 0.4377  |
> > | 1E-02                                               | 0.341 | 0.04349  | 0.3710  |
> > | 1E-06                                               | 0.337 | 0.03634  | 0.3703  |
> > | 1E-10                                               | 0.339 | 0.04173  | 0.3625  |
> >
> > | $\mathcal{L}_\text{Comp}$ Weight - 70% Missing Rate | ETTh1 | Exchange | Illness |
> > | ------ | :---: | :------: | :-----: |
> > | 10  | 0.506 | 0.28510  | 0.6264  |
> > | 1   | 0.437 | 0.22926  | 0.5119  |
> > | 1E-02                                               | 0.406 | 0.04991  | 0.4588  |
> > | 1E-06                                               | 0.403 | 0.05166  | 0.4914  |
> > | 1E-10                                               | 0.403 | 0.04077  | 0.4949  |
> >
> >
> >
> >
> > | $\mathcal{L}_\text{Consis}$ Weight - 0% Missing Rate | ETTh1 | Exchange | Illness |
> > | ------- | :---: | :------: | :-----: |
> > | 0.1  | 0.201 | 0.01859  | 0.1556  |
> > | 0.5  | 0.199 | 0.01859  | 0.1497  |
> > | 1    | 0.198 | 0.01859  | 0.1457  |
> > | 2    | 0.199 | 0.01858  | 0.1477  |
> > | 5    | 0.200 | 0.01858  | 0.1497  |
> >
> >
> > | $\mathcal{L}_\text{Consis}$ Weight - 20% Missing Rate | ETTh1 | Exchange | Illness |
> > | -------- | :---: | :------: | :-----: |
> > | 0.1   | 0.237 | 0.02306  | 0.2289  |
> > | 0.5   | 0.236 | 0.02603  | 0.2348  |
> > | 1     | 0.237 | 0.02873  | 0.2306  |
> > | 2     | 0.239 | 0.03059  | 0.2344  |
> > | 5     | 0.240 | 0.03117  | 0.2402  |
> >
> >
> > | $\mathcal{L}_\text{Consis}$ Weight - 40% Missing Rate | ETTh1 | Exchange | Illness |
> > | -------- | :---: | :------: | :-----: |
> > | 0.1   | 0.274 | 0.02697  | 0.2527  |
> > | 0.5   | 0.274 | 0.02726  | 0.2582  |
> > | 1     | 0.274 | 0.02528  | 0.2573  |
> > | 2     | 0.275 | 0.03001  | 0.2801  |
> > | 5     | 0.278 | 0.03771  | 0.2808  |
> >
> >
> > | $\mathcal{L}_\text{Consis}$ Weight - 60% Missing Rate | ETTh1 | Exchange | Illness |
> > | -------- | :---: | :------: | :-----: |
> > | 0.1   | 0.340 | 0.03013  | 0.3519  |
> > | 0.5   | 0.339 | 0.03783  | 0.3647  |
> > | 1     | 0.337 | 0.03634  | 0.3703  |
> > | 2     | 0.340 | 0.03516  | 0.3414  |
> > | 5     | 0.344 | 0.03756  | 0.3524  |
> >
> > | $\mathcal{L}_\text{Consis}$ Weight - 70% Missing Rate | ETTh1 | Exchange | Illness |
> > | -------- | :---: | :------: | :-----: |
> > | 0.1   | 0.400 | 0.03408  | 0.4587  |
> > | 0.5   | 0.401 | 0.03678  | 0.4919  |
> > | 1     | 0.403 | 0.05166  | 0.4914  |
> > | 2     | 0.404 | 0.04267  | 0.5009  |
> > | 5     | 0.410 | 0.03573  | 0.5203  |

---

> ### Author Response · Authors · 2025-11-23
> **Rebuttal by Authors**
>
> > **Q2:** Are the competitors comparable in terms of computational cost ? Did you retrain the baselines from scratch ?
>
> **Answer**
>
> Thanks for your suggestions. To ensure fair comparisons, we train all baseline models **from scratch** using identical dataset splits and experimental protocols, as detailed in Section 4.1. We evaluate computational efficiency by reporting the memory footprint and parameter counts of every model on ETTh1 as follows.
>
> **The results demonstrate that CRIB maintains a computational cost comparable to efficient Transformer baselines (e.g., PatchTST) while being significantly more lightweight than complex architectures such as CSDI and TimesNet.**
>
> | Model| Total Parameters | Memory Cost (MB) |
> | ------------- | :--------------: | :--------------: |
> | CRIB |37450 |148.03|
> | DLinear | 1200 |18.99 |
> | PAttn|15640 |55.13 |
> | SegRNN  | 8056 |30.17 |
> | Transformer|57063 |189.38|
> | iTransformer  |39768 |154.18|
> | PatchTST|41528 |179.80|
> | TSMixer | 6837 |21.16 |
> | WPMixer |44370 |50.16 |
> | CSDI |239649|  1269.72|
> | NeuralCDE  |37767 |41.40 |
> | ImputerFormer |264193|  1488.22|
> | TimesNet|863895|  1427.44|
>
>
>
>
> >**Q3:** It would be nice to add the model trained with clean data as well (missing rate = 0) to get a bound on the achievable performance.
>
> **Answer**
>
> Thanks for your advice. We have made additional experiments with missing rate=0 and added more datasets and baselines as follows.
>
> **Extensive experiments on these datasets confirm that CRIB consistently achieves state-of-the-art performance, exhibiting superior robustness across varying missing rates compared to both direct prediction and imputation-based baselines.**
>
> | Datasets Information | ETTh1 | ETTh2 | ETTm1 | ETTm2 | Electricity | PEMS-BAY | Metr-LA | Beijing Air | Weather | Exchange |
> | -------------------- | :---: | :---: | :---: | :---: | :---------: | :------: | :-----: | :---------: | :-----: | :------: |
> | Time Steps  | 17420 | 17420 | 69680 | 69680 | 26304 |  52116|  34272  | 36000 |  52696  |7588|
> | Variates |7|7|7|7|  321  |325 |207|7|21 | 8  |
>
> |ETTh1|0.0|0.0|0.2|0.2|0.4|0.4|0.6|0.6|0.7|0.7|
> |--|--|--|--|--|--|--|--|--|--|--|
> ||MAE|MSE|MAE|MSE|MAE|MSE|MAE|MSE|MAE|MSE|
> |CRIB|**0.195**|**0.163**|**0.238**|**0.211**|**0.274**|**0.273**|**0.342**|**0.427**|**0.411**|**0.624**|
> |SegRNN|0.209|0.167|0.247|0.215|0.297|0.307|0.380|0.506|0.455|0.737|
> |WPMixer|0.201|0.165|0.243|0.217|0.290|0.302|0.376|0.504|0.458|0.755|
> |iTransformer|0.211|0.172|0.250|0.219|0.338|0.316|0.385|0.494|0.453|0.731|
> |PatchTST|0.206|0.166|0.245|0.219|0.306|0.292|0.367|0.473|0.431|0.693|
> |DLinear|0.204|0.170|0.270|0.245|0.331|0.361|0.443|0.629|0.544|0.927|
> |PAttn|0.199|0.164|0.249|0.227|0.329|0.309|0.375|0.487|0.437|0.718|
> |CSDI|0.217|0.203|0.305|0.325|0.404|0.503|0.567|0.880|0.717|1.311|
> |NeuralCDE|0.291|0.236|0.340|0.315|0.408|0.460|0.522|0.761|0.629|1.101|
> |ImputeFormer|0.213|0.182|0.260|0.265|0.312|0.369|0.399|0.630|0.488|0.938|
> |TimesNet|0.225|0.178|0.256|0.234|0.297|0.302|0.373|0.457|0.443|0.627|
>
>
> |ETTh2|0.0|0.0|0.2|0.2|0.4|0.4|0.6|0.6|0.7|0.7|
> |--|--|--|--|--|--|--|--|--|--|--|
> ||MAE|MSE|MAE|MSE|MAE|MSE|MAE|MSE|MAE|MSE|
> |CRIB|**0.094**|**0.023**|**0.107**|**0.028**|**0.131**|**0.039**|**0.154**|**0.055**|**0.172**|**0.069**|
> |SegRNN|0.096|0.024|0.115|0.031|0.148|0.047|0.183|0.072|0.205|0.093|
> |WPMixer|0.095|0.024|0.116|0.030|0.147|0.045|0.202|0.086|0.218|0.104|
> |iTransformer|0.098|0.025|0.115|0.033|0.298|0.180|0.270|0.170|0.218|0.111|
> |PatchTST|0.096|0.024|0.109|0.030|0.215|0.125|0.216|0.121|0.186|0.083|
> |DLinear|0.098|0.025|0.142|0.040|0.181|0.065|0.253|0.135|0.325|0.228|
> |PAttn|0.096|0.025|0.116|0.033|0.289|0.173|0.243|0.133|0.196|0.085|
> |CSDI|0.113|0.033|0.255|0.192|0.404|0.399|0.618|0.840|0.798|1.303|
> |NeuralCDE|0.197|0.074|0.244|0.105|0.276|0.134|0.347|0.210|0.413|0.303|
> |ImputeFormer|0.101|0.026|0.122|0.034|0.140|0.049|0.173|0.061|0.174|0.071|
> |TimesNet|0.097|0.025|0.121|0.033|0.153|0.048|0.209|0.092|0.246|0.132|

---

> ### Author Response · Authors · 2025-11-23
> **Rebuttal by Authors**
>
> |Weather|0.0|0.0|0.2|0.2|0.4|0.4|0.6|0.6|0.7|0.7|
> |--|--|--|--|--|--|--|--|--|--|--|
> ||MAE|MSE|MAE|MSE|MAE|MSE|MAE|MSE|MAE|MSE|
> |CRIB|**0.028**|**0.016**|**0.038**|**0.025**|**0.050**|**0.033**|**0.062**|**0.057**|**0.070**|**0.075**|
> |SegRNN|0.031|0.021|0.042|0.028|0.050|0.037|0.066|0.061|0.076|0.081|
> |WPMixer|0.030|0.019|0.050|0.025|0.074|0.036|0.102|0.062|0.121|0.084|
> |iTransformer|0.030|0.018|0.044|0.031|0.152|0.167|0.175|0.223|0.101|0.138|
> |PatchTST|0.031|0.019|0.048|0.027|0.150|0.156|0.169|0.169|0.118|0.128|
> |DLinear|0.034|0.023|0.099|0.045|0.135|0.078|0.197|0.178|0.254|0.318|
> |PAttn|0.031|0.020|0.046|0.032|0.157|0.140|0.168|0.163|0.100|0.120|
> |CSDI|0.051|0.042|0.173|0.220|0.300|0.500|0.466|1.099|0.586|1.707|
> |NeuralCDE|0.051|0.029|0.073|0.036|0.095|0.060|0.150|0.150|0.204|0.284|
> |ImputeFormer|0.035|0.020|0.041|0.028|0.051|0.033|0.066|0.064|0.076|0.083|
> |TimesNet|0.028|0.016|0.062|0.033|0.088|0.047|0.128|0.089|0.166|0.154|
>
>
>
> |Exchange|0.0|0.0|0.2|0.2|0.4|0.4|0.6|0.6|0.7|0.7|
> |--|--|--|--|--|--|--|--|--|--|--|
> ||MAE|MSE|MAE|MSE|MAE|MSE|MAE|MSE|MAE|MSE|
> |CRIB|**0.0184**|**0.0009**|**0.0217**|**0.0016**|**0.0253**|**0.0015**|**0.0363**|**0.0031**|**0.0517**|**0.0058**|
> |SegRNN|0.0185|0.0010|0.0292|0.0031|0.0629|0.0129|0.1165|0.0363|0.1418|0.0490|
> |WPMixer|0.0187|0.0010|0.0324|0.0024|0.0783|0.0127|0.1364|0.0405|0.1836|0.0697|
> |iTransformer|0.0190|0.0010|0.0288|0.0024|0.1899|0.0770|0.2439|0.1060|0.2865|0.1413|
> |PatchTST|0.0186|0.0010|0.0273|0.0019|0.1416|0.0884|0.1978|0.1129|0.1321|0.0581|
> |DLinear|0.0205|0.0011|0.0795|0.0126|0.1469|0.0429|0.2599|0.1394|0.3778|0.3011|
> |PAttn|0.0187|0.0010|0.0227|0.0016|0.1408|0.1047|0.1283|0.0831|0.0718|0.0169|
> |CSDI|0.0264|0.0017|0.1992|0.2089|0.4358|0.5718|0.7911|1.3714|1.0766|2.2850|
> |NeuralCDE|0.3223|0.1724|0.2969|0.1592|0.3326|0.2146|0.4334|0.3814|0.5412|0.5912|
> |ImputeFormer|0.0344|0.0031|0.0364|0.0029|0.0431|0.0034|0.0608|0.0071|0.0846|0.0128|
> |TimesNet|0.0195|0.0011|0.0458|0.0041|0.0786|0.0130|0.1357|0.0404|0.1991|0.0881|
>
>
> |ETTm1|0.0|0.0|0.2|0.2|0.4|0.4|0.6|0.6|0.7|0.7|
> |--|--|--|--|--|--|--|--|--|--|--|
> ||MAE|MSE|MAE|MSE|MAE|MSE|MAE|MSE|MAE|MSE|
> |CRIB|**0.2000**|**0.2008**|**0.2368**|**0.2586**|**0.2734**|**0.3382**|**0.3414**|**0.5200**|**0.4013**|**0.6956**|
> |SegRNN|0.2265|0.2302|0.2616|0.3068|0.3005|0.4007|0.3678|0.5960|0.4266|0.8006|
> |WPMixer|0.2405|0.2741|0.2900|0.3593|0.3308|0.4645|0.4171|0.7328|0.4863|0.9928|
> |iTransformer|0.2309|0.2514|0.2909|0.3751|0.3593|0.4851|0.4365|0.7580|0.5034|1.0663|
> |PatchTST|0.2379|0.2648|0.2779|0.3486|0.3456|0.4671|0.4091|0.6769|0.4743|0.9325|
> |DLinear|0.2772|0.3504|0.3711|0.5499|0.4418|0.7412|0.5551|1.1067|0.6508|1.4721|
> |PAttn|0.2423|0.2782|0.2971|0.3864|0.3592|0.4801|0.4191|0.7022|0.4816|0.9837|
> |CSDI|0.4759|0.8854|0.5198|0.9345|0.5787|1.0499|0.7038|1.3801|0.8287|1.7733|
> |NeuralCDE|0.3331|0.4218|0.3809|0.5177|0.4391|0.6643|0.5457|0.9573|0.6278|1.2710|
> |ImputeFormer|0.2999|0.2682|0.3271|0.3282|0.3571|0.4749|0.4315|0.6978|0.4845|0.9035|
> |TimesNet|0.2100|0.2018|0.2584|0.2992|0.3040|0.4093|0.3993|0.6525|0.4828|0.8988|

---

> ### Author Response · Authors · 2025-11-23
> **Rebuttal by Authors**
>
> |ETTm2|0.0|0.0|0.2|0.2|0.4|0.4|0.6|0.6|0.7|0.7|
> |--|--|--|--|--|--|--|--|--|--|--|
> ||MAE|MSE|MAE|MSE|MAE|MSE|MAE|MSE|MAE|MSE|
> |CRIB|**0.0746**|**0.0152**|**0.0885**|**0.0203**|**0.1037**|**0.0258**|**0.1266**|**0.0384**|**0.1467**|**0.0520**|
> |SegRNN|0.0802|0.0175|0.0956|0.0231|0.1159|0.0305|0.1432|0.0477|0.1650|0.0619|
> |WPMixer|0.0821|0.0182|0.1059|0.0266|0.1355|0.0394|0.1790|0.0685|0.1899|0.0808|
> |iTransformer|0.0781|0.0161|0.1020|0.0284|0.2673|0.1688|0.2651|0.1671|0.1905|0.0861|
> |PatchTST|0.0824|0.0182|0.1005|0.0251|0.2041|0.1210|0.2425|0.1423|0.1803|0.0779|
> |DLinear|0.0918|0.0217|0.1405|0.0391|0.1743|0.0601|0.2412|0.1215|0.3104|0.2056|
> |PAttn|0.0828|0.0187|0.1024|0.0278|0.2441|0.1490|0.2398|0.1448|0.1728|0.0717|
> |CSDI|0.1313|0.0410|0.2687|0.1976|0.4131|0.4003|0.6221|0.8293|0.8024|1.2968|
> |NeuralCDE|0.1327|0.0363|0.1760|0.0591|0.1984|0.0765|0.2479|0.1194|0.3130|0.1857|
> |ImputeFormer|0.0813|0.0160|0.0888|0.0222|0.1088|0.0267|0.1360|0.0424|0.1602|0.0569|
> |TimesNet|0.0782|0.0165|0.1087|0.0260|0.1417|0.0406|0.1986|0.0811|0.2317|0.1142|
>
> |BejingAir|0.0|0.0|0.2|0.2|0.4|0.4|0.6|0.6|0.7|0.7|
> |--|--|--|--|--|--|--|--|--|--|--|
> ||MAE|MSE|MAE|MSE|MAE|MSE|MAE|MSE|MAE|MSE|
> |CRIB|**0.2526**|**0.2929**|**0.2758**|**0.3317**|**0.3106**|**0.4120**|**0.3728**|**0.5862**|**0.4337**|0.8010|
> |SegRNN|0.2552|0.3026|0.2849|0.3467|0.3188|0.4285|0.3860|0.5939|0.4412|**0.7595**|
> |WPMixer|0.2571|0.3085|0.2896|0.3651|0.3312|0.4601|0.4011|0.6503|0.4632|0.8730|
> |iTransformer|0.2606|0.3156|0.2923|0.3655|0.3545|0.4820|0.4124|0.6573|0.4691|0.8874|
> |PatchTST|0.2609|0.3008|0.2866|0.3421|0.3278|0.4305|0.3976|0.6458|0.4613|0.8540|
> |DLinear|0.2707|0.3263|0.3204|0.3853|0.3620|0.4760|0.4362|0.6799|0.5022|0.9358|
> |PAttn|0.2601|0.3102|0.2949|0.3603|0.3480|0.4689|0.4142|0.6624|0.4673|0.8745|
> |CSDI|0.3431|0.5141|0.4035|0.6010|0.4714|0.7584|0.5930|1.1378|0.7057|1.5761|
> |NeuralCDE|0.3167|0.3779|0.3478|0.4196|0.3950|0.5474|0.4750|0.7803|0.5456|1.0151|
> |ImputeFormer|0.2533|0.3189|0.2770|0.3496|0.3297|0.4645|0.4006|0.6110|0.4549|0.8327|
> |TimesNet|0.2583|0.2939|0.2922|0.3488|0.3322|0.4422|0.4017|0.6275|0.4789|0.9234|

---

> > ### Comment · Reviewer_bbCN · 2025-11-24
> >
> > Thank you for your answer and your work during the rebuttal.
> >
> > What is the metric you are reporting for the ablation on the loss weights of Q1 ? MAE ? Are the results of the original configuration supposed to be in Table 4 of the Appendix (e.g., for ETTh1) ?

---

> > > ### Author Response · Authors · 2025-11-24
> > > **Response to bbCN's Questions**
> > >
> > > **Thanks for your time and response.**
> > >
> > >
> > > **Metric:** Yes, the metric reported in the ablation study on loss weights is indeed MAE.
> > >
> > > **Original Configuration:** Yes, you are correct. The "original configuration" refers to the CRIB model using the optimal hyperparameters (the fixed weights used for the main experiments).
> > >
> > > **Note on Reproducibility:** We would also like to mention that we have re-run all experiments (both baselines and our model) during the rebuttal period to ensure the solidity of our results. Consequently, there may be slight numerical fluctuations in the updated tables compared to the original submission, but the overall performance trends and conclusions remain consistent.

---

### Author Response · Authors · 2025-11-29
**New version of paper**

Dear Area Chair and Reviewers,

We have uploaded the revised manuscript. This version incorporates the additional experimental results (new baselines and datasets) and clarifications promised in our rebuttal responses in the **final part** of our paper.

Key updates include:
1. Integration of **4** new baselines (CSDI, ImputeFormer, etc.) and **6** new datasets.
2. Clarified theoretical motivations and method descriptions.
3. Improved figures and corrected typos.

Thank you again for your constructive feedback.

Best regards,

The Authors

---

### Author Response · Authors · 2025-12-01
**General Response to All Reviewers and Area Chair**

We thank all reviewers for their insightful and constructive feedback. We have carefully addressed all concerns and significantly strengthened our manuscript with extensive new experiments and clarifications.

**Below is a summary of our major updates of the paper:**

1. **Clarification of Motivation and Novelty**

   We have reinforced our critique of the "imputation-then-prediction" paradigm.

   - **Detrimental Imputation:** We provided new empirical evidence (e.g., **TimesNet + TimeXer**) showing that unsupervised imputation often degrades performance compared to direct prediction, as it reinforces biased patterns from partial observations (see **Figure 1** and **Table 5**).

   - **Beyond Naive IB:** We clarified that a naive application of the Information Bottleneck (IB) is insufficient for sparse time series. We demonstrated that without our proposed **Consistency Regularization** and **Unified-Variate Attention**, standard IB models yield degenerate, locally-biased attention maps.

2. **Extensive Expansion of Baselines and Datasets**

   To ensure a comprehensive evaluation, we have expanded our experiments significantly:

   - **New Baselines:** We added comparisons against **CSDI**, **ImputeFormer**, **NeuralCDE**, and **TimesNet** (Direct) across the board.

   - **More Datasets:** We expanded our evaluation to **10 real-world datasets** (adding Weather, Exchange, BeijingAir, etc.).

   - **Results:** As shown in the updated **Table 4 & 7**, CRIB consistently achieves state-of-the-art performance across varying missing rates.

3. **Validation on Natural Missingness (AQI Dataset)**

   Addressing the suggestion to test on real-world missing data, we conducted experiments on the AQI dataset with **naturally occurring missing values.**

   - **Key Finding:** We observed that imputing natural missing values (using a pre-trained TimesNet) **degraded** prediction performance compared to direct prediction (see **Table 8** in the updated paper). This strongly validates our core hypothesis that direct prediction is superior when ground truth for missing values is unavailable.

4. **Computational Efficiency and Robustness**

   - **Training Cost:** We provided a detailed analysis of parameter counts and memory usage, demonstrating that CRIB remains computationally efficient, comparable to lightweight Transformer baselines like PatchTST, and far more efficient than diffusion-based methods like CSDI.

   - **Ablation & Sensitivity:** We conducted additional ablation and sensitivity studies across multiple datasets, confirming the necessity of all loss components ($\mathcal{L}\_{\text{Consis}}$, $\mathcal{L}\_{\text{Comp}}$, $\mathcal{L}\_{\text{Pred}}$) and the stability of our hyperparameters.

We believe these revisions solidly address the reviewers' concerns regarding motivation, generalizability, and experimental rigor.

---

### Meta-Review · Area_Chair_CEke · 2026-01-05

**Summary:**

## Summary

This paper introduces a new paradigm for time-series forecasting with missing data based on the Information Bottleneck Principle, demonstrating state-of-the-art performance across four diverse datasets in an extensive experimental evaluation. The work addresses a critically important real-world challenge. However, several significant limitations temper these strengths. The practical relevance is undermined as experiments use artificially induced missing data on complete, highly periodic datasets, failing to test on real-world, non-stationary, or high-dimensional data.



## Overall Score

bbCN: 6 (keep positive score)

g59r:  4 (no response)

K95D: 2 (no response)

ek3n: 4 (no response)

2Cic:  4 (maintain)

nmeJ: 4 (maintain)

## Addressed concerns

* Limited comparison with the competitors (bbCN)
* More details about the proposed point in intro (bbCN)
* Typos and errors (bbCN)
* Limited discussion on experiment results and settings (bbCN)

* Missing baselines and experiments (g59r, K95D, ek3n, 2Cic, nmeJ)

## Remaining concerns

* Limited novelty (bbCN, g59r, 2Cic, nmeJ)
* Unreasonable experiment design (K95D)
* Missing some explaination on why models degrade under high missing rates (ek3n)
* Missing theoretical analysis (ek3n, nmeJ)
* Limited ablation design (2Cic)
* Reproducibility  (2Cic)

## Conclusion:

Overall, during the review phase, most of the reviewers (g59r, K95D, ek3n,  2Cic and nmeJ) weren't fully satisfied with the experimental design and the novelty of the paper. They raised concerns about the limited reproducibility, weak baselines and missing  theoretical analysis. Reviewer bbCN, on the other hand, leaned toward acceptance. During the discussion, the authors did a good job addressing the reviewers' questions and concerns partly, but most of the reviewers didn't follow up afterward or raise scores.

This paper has potential to be stronger with some improvements to the experimental design, novelty and writing clarity.

**Reviewer Concerns:**

Refer to Summary

**Reviewer Scores:**

Refer to Summary

---

### Decision · Program_Chairs · 2026-01-26

Reject